# Late Paleolithic whale bone tools reveal human and whale ecology in the Bay of Biscay

Krista McGrath[1,18], Laura G. van der Sluis [2,3,4,18], Alexandre Lefebvre [5,6], Anne Charpentier [7], Ana S. L. Rodrigues [7], Esteban Álvarez-Fernández [8], François Baleux[9], Eduardo Berganza[10], François-Xavier Chauvière [11], Morgane Dachary[9,12], Elsa Duarte Matías[13], Claire Houmard[14], Ana B. Marín-Arroyo[5], Marco de la Rasilla Vives[13], Jesus Tapia[10], François Thil[15], Olivier Tombret[2], Leire Torres-Iglesias [5,16], Camilla Speller [17], Antoine Zazzo [2,19] & Jean-Marc Pétillon[9,19] ✉

Reconstructing how prehistoric humans used the products obtained from large cetaceans is challenging, but key to understand the history of early human coastal adaptations. Here we report the multiproxy analysis (ZooMS, radiocarbon, stable isotopes) of worked objects made of whale bone, and unworked whale bone fragments, found at Upper Paleolithic sites (Magdalenian) around the Bay of Biscay. Taxonomic identification using ZooMS reveals at least five species of large whales, expanding the range of known taxa whose products were utilized by humans in this period. Radiocarbon places the use of whale products ca. 20–14 ka cal BP, with a maximum diffusion and diversity at 17.5–16 ka cal BP, making it the oldest evidence of whale-bone working to our knowledge. $\delta^{13}$C and $\delta^{15}$N stable isotope values reflect taxon-specific differences in foraging behavior. The diversity and chronology of these cetacean populations attest to the richness of the marine ecosystem of the Bay of Biscay in the late Paleolithic, broadening our understanding of coastal adaptations at that time.

Whales are the largest living animals on Earth and the current populations of many species are a mere fraction of their abundance in the past[1]. Before intensive whaling depleted the populations of most species, whales were a valuable source of food and other resources (e.g., oil, bone, baleen[2]). They were thus a key part of subsistence for many coastal human groups worldwide, including hunter-gatherers and Neolithic farmers, with acquisition methods that included scavenging freshly beached animals, opportunistic killing and organized whaling[2–8]. However, reconstructing the beginning of whale utilization is challenging because prehistoric coastal sites are an especially fragile part of the archeological record, many of them having been lost to marine erosion or flooded by the last marine transgressions[9]. In most cases, the only available

evidence is indirect, in the form of materials of coastal origin transported by people into inland sites. In Europe, a number of sites attributed to the Middle and Upper Magdalenian culture (ca. 19-14 ka cal BP, or 19-14 millennia before present, at the end of MIS2) in southwestern France and in Atlantic Iberia have yielded an invaluable record of this type of evidence[10]: whale barnacles attesting to the transport of whale skin, blubber and meat[11]; unworked whale bones transported and processed at the habitation site[12]; worked whale teeth[13]; and, especially, more than 150 tools and projectile heads made of whale bone presumably of Atlantic origin, mostly found scattered from Asturias to the central part of the northern Pyrenean range[14]. This record, mostly identified within the last ten years, represents (to our knowledge) the oldest evidence of a regular

utilization of whale products by humans, for dietary needs and raw materials[10].

Understanding this Magdalenian utilization of whale products is of crucial importance, not only because it opens a window into early human interactions with whales, but more broadly because of the light it sheds onto the history of human coastal adaptations. Indeed, even though the earliest evidence of the use of coastal resources in Europe dates back to the Middle Paleolithic (older than 40 ka cal BP, and up to ca. 150 ka BP[15–17]), and similar evidence exists for the Early Upper Paleolithic[18], the richness and diversity of evidence related to seashore exploitation increases dramatically in association with the Magdalenian culture. Despite a sea level rise of ca. 120 m since that period making the Magdalenian shoreline inaccessible to today's archeologists, depictions of marine animals, remains of seals, dolphins, marine fish and birds, the use of marine mollusks as food and the use of their shells as raw material for ornaments, are all attested with unprecedented frequency and wide diffusion in the Middle and Upper Magdalenian, especially in Iberia and in the Pyrenees[10,19,20]. This combined evidence points to a stronger link with the seashore compared to the earlier part of the European Upper Paleolithic, suggesting the inception of a settled coastal economy in certain parts of Europe at this time. Understanding the role that whale products played in this process is thus key to the ongoing investigations into the existence, chronology and organization of hunter-gatherer early coastal adaptations, a subject that is today considered central in prehistoric research[21–23]. However, the information provided by the whale remains found in Magdalenian assemblages is limited for several reasons. First, it is unknown to which whale species they correspond. Indeed, these remains were identified only through visual macroscopic criteria and most of them are small, fragmented and/or worked, thus precluding any precise anatomical or taxonomic identification beyond the cetacean category. Furthermore, although these morphological criteria are robust, they have a margin of uncertainty, meaning that some of the remains are classified only as "likely" made of cetacean bone[14,24]. Since different whale species have different ecologies and feeding behavior, this generic (and sometimes uncertain) identification as cetacean gives only low resolution insight into the range of interactions that foragers could have had with these animals, and deprives us of the environmental information that the presence of certain particular whale species could indicate[25]. Finally, there is substantial uncertainty regarding the dating of these whale remains. Because of their typology, their stratigraphic context, and the characteristics of the archeological assemblages they were found in, they can be attributed to the Magdalenian period in the broad sense of the term. But since the majority are from ancient excavations with poor chrono-stratigraphic resolution, they can only be collectively ascribed to a broad time span of several millennia, precluding a precise reconstruction of the chronology, rhythm and evolution of the utilization of whale products.

Here we analyze a large sample of worked bone objects ($n = 83$) from 26 Magdalenian cave and rockshelter sites in the Cantabrian region and southwestern France, all visually identified as made of whale bone (Supplementary Data 1 and 2, Supplementary Figs. 1 through 64). We also analyze 90 unworked bone fragments from the single assemblage of Santa Catalina cave (Biscay), also ascribed to whale on a visual basis, found among the faunal remains of the site's Upper Magdalenian occupation and showing traces of anthropic processing (notably percussion notches: Supplementary Figs. 65 through 90). We sampled these for taxonomic identification through collagen peptide mass fingerprinting (ZooMS, or Zooarcheology by Mass Spectrometry[26,27]). Thirty-seven of the worked objects and 31 of the unworked bone fragments were also sampled for radiocarbon dating using the compact AMS ECHoMICADAS, and for carbon and nitrogen stable isotope analysis ($\delta^{13}$C and $\delta^{15}$N).

In this work, we assess the range of whale taxa whose products were utilized by the Magdalenian foragers and the chronology of this utilization, showing that these groups availed themselves of the carcasses of at least five species starting ca. 20-19 ka cal BP, with a peak time ca. 17.5-16 ka cal BP. The results of the stable isotope analysis, placed in the context of patterns of isotopic niche partitioning among contemporary whales, contribute to reconstructing whales' relative foraging behavior and past ocean ecology, showing that the distribution of the stable isotope ratios in Paleolithic whales is structured by taxa and overlaps to some extent with those of their modern counterparts, suggesting broadly similar feeding strategies and analogies with today's arctic water communities. Even though the Paleolithic seashore itself is no longer accessible, and the range of taxa identified here might not reflect the full range of species present in the Bay of Biscay at that period, the analysis of these whale bones brought inland by the hunter-gatherers opens a unique window into whale ecology and the marine environments in the northeastern Atlantic at that period, and on the timing and nature of their utilization by human groups.

## Results
### Taxonomic identification using ZooMS
Of the 173 bone specimens (83 worked objects and 90 bone fragments) analyzed with ZooMS, all but four yielded a taxonomic identification, demonstrating the power of this approach to identify taxa on highly transformed and/or fragmented remains of late Pleistocene age. Of the 83 worked objects, 71 were confirmed as cetaceans, while 8 were identified as large terrestrial mammals and 4 did not yield a ZooMS identification, indicating that the macroscopic visual attribution was correct in 90% of the cases (71/79). The visual misidentification of 8 objects made of bone from large terrestrial mammals was due to their thoroughly porous aspect, normally a diagnostic feature of whale bones, but also present in some anatomical elements of certain terrestrial species (in this case mammoth, rhinoceros, reindeer and equids), and that can be misleading when dealing with small, fragmented objects. Of the 90 unworked bone fragments, the attribution as cetacean was confirmed for 60 bones (67%), with the other 30 being identified mostly as large land mammals, but also one seal. The higher error rate for Santa Catalina (33% vs. 10%) is a consequence of the fact that the visual selection of putative whale-bone fragments was more inclusive at this site (see "Methods" below).

Overall, ZooMS analyses of 131 cetacean specimens reveal the presence of at least six cetacean taxa in the northeastern Atlantic during the Magdalenian (Fig. 1 and Supplementary Data 3): fin whale, *Balaenoptera physalus* ($n = 65$); sperm whale, *Physeter macrocephalus* ($n = 32$); gray whale, *Eschrichtius robustus* ($n = 11$); blue whale, *Balaenoptera musculus* ($n = 2$); one species of porpoise (harbor porpoise or Dall's porpoise, *Phocoenidae*, $n = 1$); and at least one species of Balaenid whale (Balaenidae), with 13 samples that can be attributed either to the North Atlantic right whale, *Eubalaena glacialis*, or to the bowhead whale, *Balaena mysticetus* (two species that are indistinguishable using ZooMS), both present in the North Atlantic. The remaining 7 samples yielded ZooMS spectra that could not be attributed to a precise cetacean taxon. Among the six cetacean taxa, only the sperm whale had previously been unambiguously documented in the Magdalenian record, through the presence of two carved teeth and several depictions on other portable objects from the Bay of Biscay region[28]. The other taxa—fin whale, gray whale, blue whale, right and/or bowhead whale and porpoise—had (to our knowledge) previously not been identified in this archeological context.

### Radiocarbon dating
Of the 37 worked objects sampled for radiocarbon dating, 5 failed as a result of poor collagen preservation, while the remaining 32 yielded results (Fig. 2, and see "Methods" below for calibration and caveats). The two earliest dates are from the Cantabrian sites of Rascaño and El Juyo, ca. 20.2-19.6 and 19.6-19 ka cal BP, respectively. The two artifacts

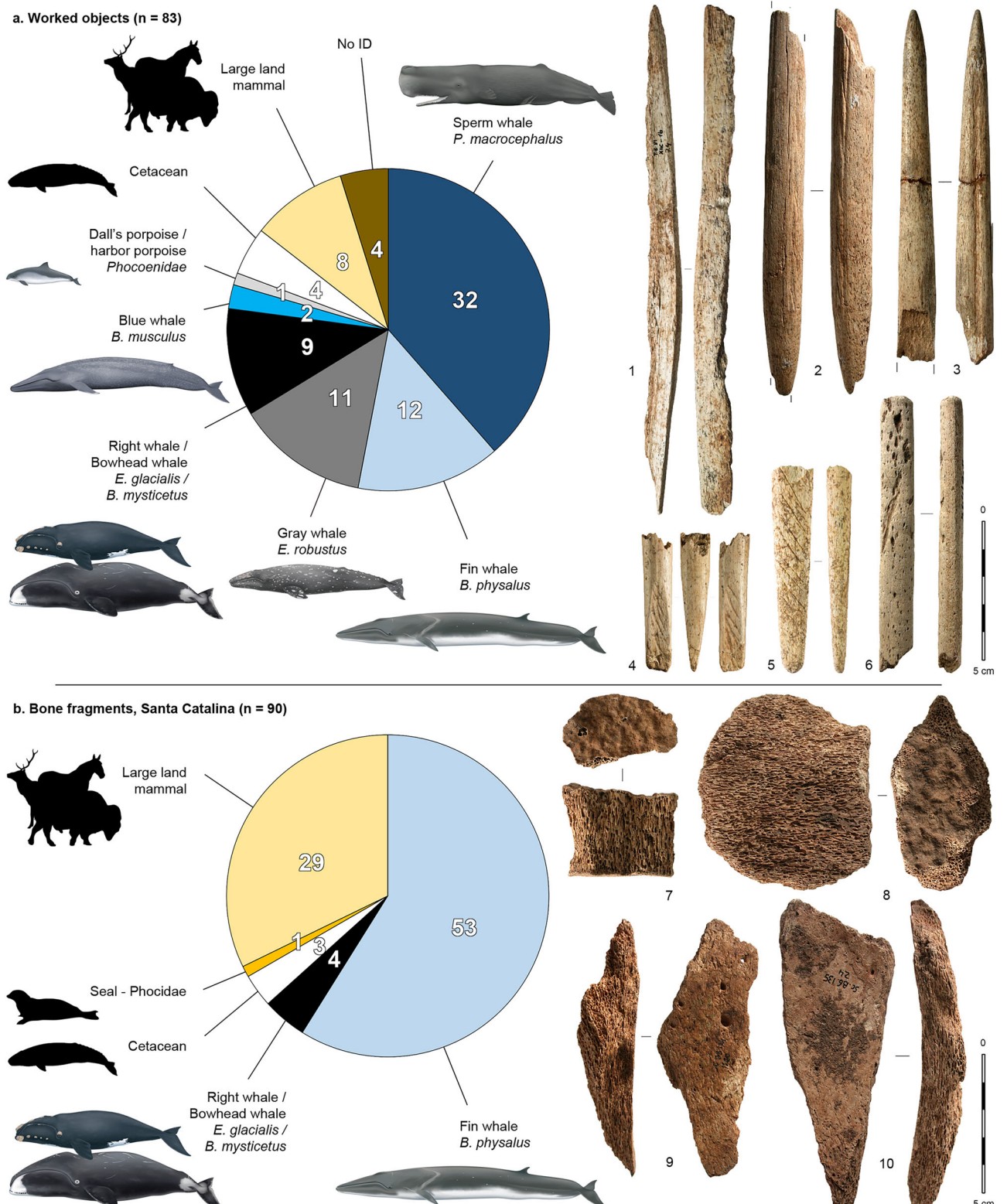

**Fig. 1 | Taxonomic identification of the 173 bone specimens analyzed using ZooMS, and examples of the main categories of elements. a** worked objects; **b** unworked bone fragments. 1: blank, Tito Bustillo, sperm whale (#Hum1); 2: projectile point with massive base, Isturitz, blue whale (#15); 3: projectile point, Brassempouy, fin whale (#822); 4: possible foreshaft, Las Caldas, sperm whale (#982); 5: projectile point with massive base, Ermittia, gray whale (#968); 6: unidentified object, Saint-Michel, sperm whale (#799); 7–10: unworked fragments of fin whale bone, Santa Catalina (#SC B6 109 1085, SC B8 144 439, SC B8 124 1180, SC B6 135 24). Whale drawings courtesy of Uko Gorter. Black silhouettes from phylopic.org, CC0 1.0 license. Pictures by AL and JMP. See Supplementary Data 1 for source data.

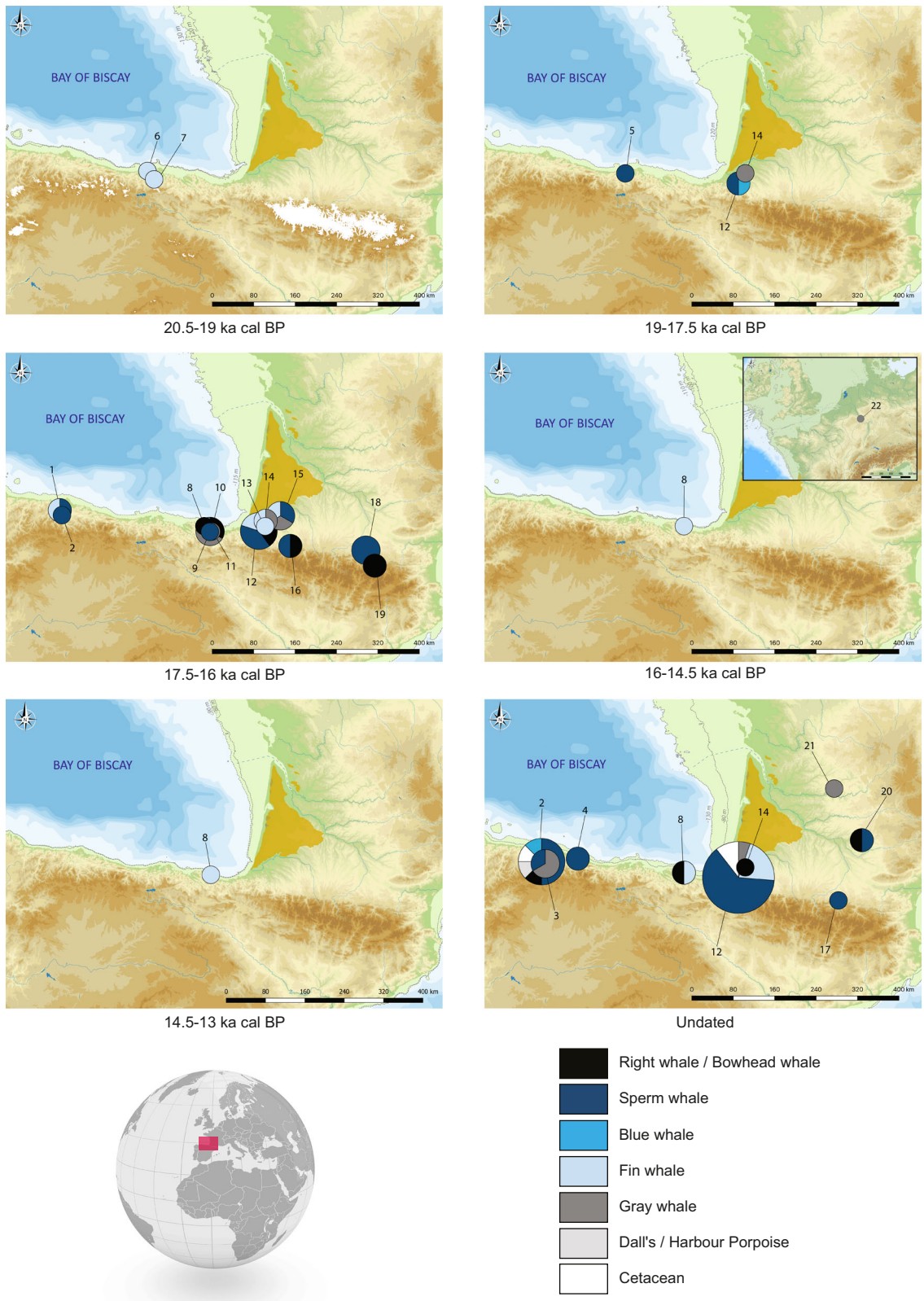

**Fig. 2 | Chronological and geographical distributions of all cetacean bone remains taxonomically identified from Atlantic Magdalenian sites (except the site of Andernach, Rhineland).** All circles correspond to worked objects except site 8, which is the assemblage of unworked fragments of whale bone from Santa Catalina. The size of the circles is proportional to the numbers of remains per site, except in Santa Catalina, where all the unworked remains are considered as a single one so as not to overwrite the representation. Bathymetric data are from Natural Earth (www.naturalearthdata.com). The Sables des Landes region appears in yellow. 1: La Paloma, 2: Las Caldas, 3: La Viña, 4: Tito Bustillo, 5: El Pendo, 6: El Juyo, 7: El Rascaño, 8: Santa Catalina, 9: Iruroin, 10: Ermittia, 11: Urtiaga, 12: Isturitz, 13: Bourrouilla, 14: Duruthy, 15: Brassempouy, 16: Saint-Michel, 17: Tuc d'Audoubert, 18: Mas d'Azil, 19: La Vache, 20: Courbet, 21: La Madeleine, 22: Andernach. Maps by AL. See Supplementary Data 1 for source data.

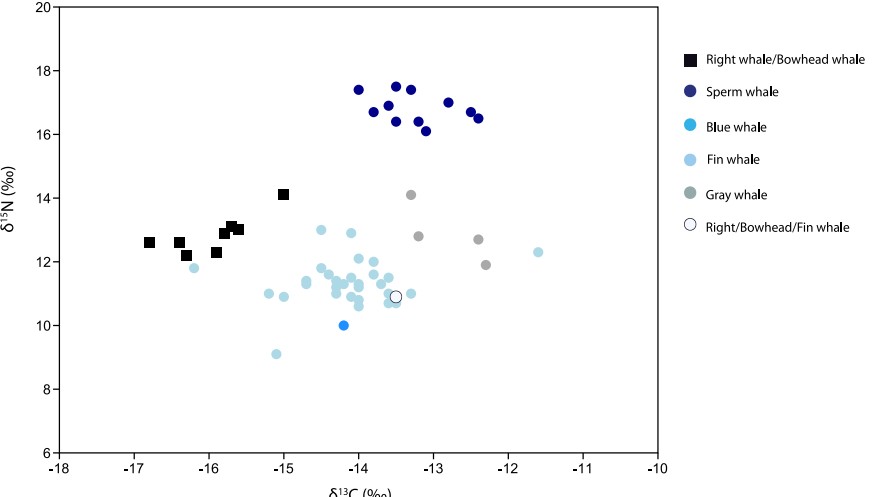

**Fig. 3 | Carbon (δ13C) and nitrogen (δ15N) stable isotope ratios in the 55 samples of whale bone.** The open circle indicates the position of the one sample where ZooMS failed to distinguish between right/bowhead/fin whale. See Supplementary Data 4, tab "PaleoCet material", for source data, and Supplementary Data 1 for more details on samples.

were found among assemblages attributed to the Cantabrian Lower Magdalenian (Rascaño IVb and El Juyo 8), and our results confirm this archeological association.

Only four worked objects, from three sites in the western Pyrenees and Cantabria (Duruthy, Isturitz and El Pendo), yielded dates in the 19-17.5 ka cal BP time range. The chronological distribution of the dates then shows a sharp rise to the 17.5-16 ka cal BP time range, with 26 objects from 12 sites (from west to east, 2 from La Paloma, 1 from Las Caldas, 3 from Ermittia, 1 from Urtiaga, 1 from Iruroin, 5 from Isturitz, 1 from Bourrouilla, 2 from Duruthy, 3 from Brassempouy, 2 from Saint-Michel, 3 from Mas d'Azil and 2 from La Vache), accompanied by a wide geographic extension from Asturias (Las Caldas) to the central Pyrenees (La Vache, Le Mas d'Azil) and a maximal diversity

of taxa used. The distribution of the dates ends abruptly after ca. 16 ka cal BP.

At Santa Catalina, from the 31 fragments of unworked whale bone sampled for radiocarbon dating, all but one yielded reliable results (Fig. 2). The two fragments attributed to the North Atlantic right whale or bowhead whale (one of which was dated three times) yielded identical 14C dates and might come from a single individual that died ca. 16.5-16 ka cal BP. The 28 other dates are all from fragments of fin whale bone and, with one exception, cluster between ca. 16-14.5 ka cal BP (thus after the 17.5-16 ka cal BP time range that yielded most of the dates on worked objects). The number of acquisition events represented by these 27 dates remains undetermined, and the degree of fragmentation of the bones precludes the calculation of an anatomical Minimum Number of Individuals; but the distribution of the dates appears compatible with a minimum of two individuals, one ca. 15.5 and one ca. 15 ka cal BP. A single date ca. 13.6–13.2 ka cal BP is the only possible evidence of a later episode of bone acquisition on a fin whale individual.

### Stable isotope analysis

Carbon (δ13C) and nitrogen (δ15N) stable isotope ratios were successfully measured in 55 whale bone samples and ranged from −16.8‰ to −11.6‰ in δ13C ratios and from 9.1‰ to 17.5‰ in δ15N ratios (Fig. 3). In total, 8 samples failed collagen extraction, 2 samples failed IRMS analysis and for 7 samples the atomic C:N ratio was too high (ranging between 3.77 and 6.28) or the N peak was below 700 mV (Supplementary Data 1).

The distribution of the stable isotope ratios is structured by taxa: fin whale samples are clustered at low δ15N values and (with one exception) intermediate values of δ13C; the single blue whale sample is nested among the fin whales; right/bowhead whale samples show low δ13C ratios and intermediate values of δ15N; gray whales display high δ13C ratios and intermediate values of δ15N; and sperm whales show elevated values both of δ15N values and δ13C (Table 1). The one sample where ZooMS failed to distinguish between right/bowhead and fin whales—due to the overall poor quality of the spectra and specifically the absence of a clear peptide marker at P2_α2 292—is nested within the fin whale samples, strongly suggesting it might have corresponded to a fin whale (Fig. 3). We found no evidence that these patterns of stable isotopic signatures across taxa are driven by differences in the radiocarbon age of samples: there was no significant relationship between

**Table 1 | Carbon (δ13C) and nitrogen (δ15N) stable isotope averages, standard deviations and ranges for each taxon in the 55 samples of whale bone**

|  | δ13C (‰) | δ15N (‰) |
|---|---|---|
| **Fin whales** |  |  |
| average | −14.1 | 11.3 |
| stdev. | 0.76 | 0.72 |
| min | −16.2 | 9.1 |
| max | −11.6 | 13.0 |
| **Gray whales** |  |  |
| average | −12.8 | 12.9 |
| stdev. | 0.51 | 0.93 |
| min | −13.3 | 11.9 |
| max | −12.3 | 14.1 |
| **Right/bowhead whales** |  |  |
| average | −16.0 | 12.9 |
| stdev. | 0.55 | 0.60 |
| min | −16.8 | 12.2 |
| max | −15.0 | 14.1 |
| **Sperm whales** |  |  |
| average | −13.2 | 16.8 |
| stdev. | 0.51 | 0.47 |
| min | −14.0 | 16.1 |
| max | −12.4 | 17.5 |

conventional [14]C age and the $\delta^{13}$C values (see Methods below, and Supplementary Fig. 98).

A comparison between the stable isotopic signatures of Paleolithic whale samples with modern counterparts shows in most cases an overlap (at least partial) between the two. The large majority of modern samples used for this comparison originate from the same oceanic basin (Atlantic) as our samples, except for gray whale (Pacific) (Supplementary Data 4). Still, on average we found higher $\delta^{13}$C and $\delta^{15}$N ratios in ancient fin whales, higher $\delta^{15}$N ratios in ancient sperm whales, and higher $\delta^{13}$C ratios in bowhead/right whales and in gray whales when compared with modern whales (Supplementary Discussion, and Supplementary Figs. 99 through 102).

## Discussion

### Whale bone chronological and geographical distribution
A previous study of whale-bone objects from the Cantabrian coast suggested that, according to contextual evidence (both archeological and radiocarbon), the utilization of whale bone might have appeared in this region and in this archeological context ca. 18–17.5 ka cal BP[14]. Our results confirm this archeological association and the earlier evidence of whale-bone working the Cantabrian region, and push back the chronology of its inception by at least one millennium (with the caveat of radiocarbon dating uncertainty; see Methods below). The whale bones used for tool manufacture do not necessarily come from fresh carcasses, but they are unlikely to have been more than a few decades old, given the hazards of bone preservation in the open air on the seashore[29], and given that the low density of cetacean bones[30] suggests that they will be particularly susceptible to density-mediated attrition[31]. Accordingly, at our level of chronological resolution, the date of the material is essentially the same as the date of the bone working activity. Therefore, according to our results, the manufacturing of implements made of whale bone began ca. 20-19 ka cal BP on the southern shores of the Bay of Biscay. Since the visual identification of whale-bone objects from more ancient (Gravettian) archeological contexts remains uncertain[32,33], these 20-19 ka cal BP dates represent to our knowledge the oldest evidence for the working of whale bone by hunter-gatherers. They place the inception of this industry at the beginning of the Late Glacial, when the sea level began to rise again after the LGM lowstand[34]. Between ca. 17.5 and 16 ka cal BP, the amount of archeological evidence increases sharply and the geographic distribution of the objects expands, encompassing an area between Asturias and the central Pyrenees, perhaps with an extension to the north and east of the Aquitaine Basin (given their archeological context, the objects found at La Madeleine, Dordogne, and Courbet, Tarn, might indeed date to the same period[35–37]). This period, which archeologically corresponds to the Middle Magdalenian in the Cantabrian region and the Late Middle Magdalenian in southwest France[38], thus appears as the peak time for the production and diffusion of objects made of whale bone. No chronological trend appears regarding the type of objects manufactured or the taxa used, and this period in general can be considered as representing the maximum diffusion and diversity of this whale-bone industry. This is in accordance with other categories of archeological evidence showing the existence of active exchange/diffusion networks between the Cantabrian coast and the central Pyrenees at that time (lithic raw materials, art, and types of tools[14]).

Most of the objects made of whale bone are weapon elements (projectile points and foreshafts) typologically similar to the antler points that make up a large part of the Magdalenian hunting equipment[39]. From what can be observed on the finished objects, and on the few pieces of manufacturing waste documented[37], the manufacturing techniques used to work whale bone into objects do not differ from those used for working antler. The choice of using whale bone to make one part of the weapon tips might be linked to the dimensions of the raw material, that make it possible to manufacture very long implements, and perhaps the specific mechanical properties of whale bone as compared to antler and to terrestrial bone[14,24,40].

The chronological distribution of the whale-bone objects ends abruptly after ca. 16 ka cal BP. The only worked object that might be attributed to a later date is the whale-bone foreshaft from the Upper Magdalenian site of Andernach, in Rhineland—an object that was found to be made of gray whale bone in this study, and that can be attributed to ca. 16–15.5 ka cal BP according to the AMS dates from the context in which it was discovered[40]. Compared to the rest of the samples, this object is thus both the most geographically outlying and the most recent. Only tentative explanations can be put forth to explain this rarefaction. The period ca. 16–14 ka cal BP corresponds archeologically to the Upper Magdalenian—a cultural phase that, like the preceding one, yielded evidence for the exploitation of a number of coastal resources (such as mollusks[10]). The acquisition of fin whale carcasses at Santa Catalina after 16 ka cal BP, and the presence of a foreshaft made of gray whale bone at Andernach with contextual dates younger than 16 ka cal BP, both show that hunter-gatherer groups still had access to whale bone at that period. Therefore, the nearly complete lack of evidence for whale-bone working after 16 ka cal BP is not linked to an abandonment of coastal resources, or to a lack of suitable raw material or of technical skill. It could thus be either a purely cultural and technical choice (a disregard for whale bone as a raw material for tool making), or a question of archeological visibility linked to the interruption of the whale-bone diffusion networks. Indeed, since the whale-bone objects were probably manufactured on the seashore[14,24], all objects of this type found at inland sites are imports. If the exchange and circulation dynamics that are responsible for the diffusion of these objects ceased to function, the whale-bone objects likely remained at the coastal sites, which are now submerged. There is currently no way to discriminate between those two hypotheses.

### Cetacean diversity and ecology
The taxa identified in this study provide a picture of the rich biodiversity of cetaceans in the northeastern Atlantic during the Magdalenian period that may have been accessible to hunter-gatherers. With its highly productive waters and complex bathymetry, the Bay of Biscay remains today an area of high diversity and abundance of cetacean species[41], but it may have been even more so in the past, given that multiple species were subsequently subject to intensive exploitation[1]. In addition, the climate in the Bay of Biscay was substantially cooler during the Magdalenian period, with sea ice likely present at least seasonally[42], which means that the cetacean community may have more closely resembled that of today's arctic waters. Given the role large cetaceans play in marine ecosystem functionality, these differences in cetacean composition and abundance may have translated into non-negligible effects on local primary productivity, with cascading effects across the broader ecosystem[43–45].

Fin whales, sperm whales, blue whales and harbor porpoises are still present in the Bay of Biscay today[41]. Given their wide distribution, including cold waters[46], it is not surprising that they were also present during the Magdalenian period, but they may have been more abundant prior to human exploitation[1]. Our results show that at least one balaenid species was found in the Bay of Biscay at the time, although the imprecision of the ZooMS ID does not allow for a distinction between right and bowhead whales; DNA analysis would be required to verify the species identities for these specimens (analyses which fell outside the scope of this study). Both species were heavily exploited in the western North Atlantic, with right whales now extirpated[47] and bowheads nearly so[48]. Right whales were present in the Bay of Biscay until relatively recent times, where a coastal calving ground formed the basis of a Basque whaling industry from the 11th–18th centuries[47]. In the cooler Magdalenian waters[42], it is more likely that right whales, if present, were using the area as summer feeding grounds. The sea-ice bound bowhead whales could have been a more likely species, with

stable isotope data also providing some support in this direction (Supplementary Fig. 101). It is not impossible that both species could have been present, in different seasons.

The case of gray whales is more specific. Gray whales today have a strictly North Pacific distribution: they do not seem to have survived in the North Atlantic after the eighteenth century, having likely disappeared due to whaling despite the circumstances of their disappearance remaining poorly understood[7,49]. Previous evidence of their past presence in the North Atlantic dates either to the Holocene (10 to 0.25 ka cal BP) or to the middle of the Marine Isotope Stage 3 (MIS3), before 40 ka cal BP; with the absence of gray whales in the intervening period attributed to a drastic reduction or even extirpation of Atlantic gray whale[50–52]. Furthermore, molecular analyses revealed little genetic continuity between the late Pleistocene and the Holocene populations: most Atlantic Holocene specimens analyzed by Alter et al.[50] are genetically closer to Holocene Pacific gray whales than to Pleistocene Atlantic specimens. This led these authors to suggest that the majority of the Atlantic Holocene population were the result of a second colonization event from the Pacific, after warming temperatures, sea-level rise, and decreases in sea ice permitted passage through the Bering Strait[50]. The 11 artifacts made of bones from gray whales presented here are evidence of the presence of this species in Atlantic waters at the end of MIS2, thus supporting a continuity between the Pleistocene and the Holocene populations.

The stable isotope composition of marine vertebrates is controlled by their diet and habitat preferences[53,54] and thus the stable isotope dataset presented in this study sheds light onto past whale ecology. In general, the stable isotopic signatures of Paleolithic whales overlap to some extent with those of their modern counterparts, suggesting broadly similar feeding strategies, while existing differences can be ascribed to variation in stable isotope baseline values, feeding ground locations and possibly trophic level (Supplementary Discussion). As in modern whales, ancient fin whales and blue whale samples have relatively low nitrogen stable isotope ratios in comparison with the other whale samples, consistent with these species' reliance on krill[55,56], whereas gray whales display higher $\delta^{13}C$ ratios, characteristic for their feeding behavior in benthic environments[57], while sperm whales (the only toothed whale species in this dataset) show the most elevated $\delta^{15}N$ values, reflecting a diet at a higher trophic level including large squid[58] (Fig. 3). The fact that bowhead/right whales (which feed mostly on copepods) have higher $\delta^{15}N$ values than fin whales (again, also found in modern whales) is less straightforward to explain, but likely explained by feeding in different water masses, highlighting the complexity of interpreting isotopic signatures (Supplementary Discussion). This said, in most cases the overlap in stable isotope signatures between modern and ancient whale samples is only partial, with ancient whales tending to exhibit higher $\delta^{15}N$ and/or $\delta^{13}C$ values. These differences may reflect a shift in whale feeding preferences (e.g., if modern whales now target prey at lower trophic levels or warmer waters), large scale temporal changes in the stable isotope baseline[59], environmental change, or a combination. Indeed, previous studies found rapid changes in whale feeding behavior (and in corresponding stable isotope signatures) as a result of environmental changes: for example, Jory et al.[60] found that a reduction in biomass of artic krill coincided with a dietary niche widening in fin whales in Canadian waters, with 60% of individuals changing their foraging strategy from specialist to generalist feeders in order to reduce intraspecific competition. Given the magnitude of the environmental change from the Late Pleistocene to the present, the changes in whale foraging strategies may well have been even stronger.

The stable isotope analysis complements the ZooMS results in showing a clear differentiation between whale species, and furthermore allowed us to identify as likely fin whale one specimen for which identification through ZooMS was imprecise (Fig. 3). It also suggests a higher likelihood that the Balaenidae samples correspond to bowhead

whales rather than right whales (Supplementary Fig. 101) although this is less certain. While the arctic conditions in the region would favor the bowhead hypothesis, the region could similarly have functioned as a northerly feeding ground for right whales. These results highlight the added value of combining multiple analytical methods to maximize the information obtained from archeological records of human-whale interactions[2,53].

## Whale products acquisition

With the exception of the object from Andernach, all our specimens come from karstic environments (cave and rockshelter sites) with good preservation of osseous material, including small elements. The relative abundance of the different whale taxa in our sample is thus unlikely to be affected by taphonomic factors such as differential preservation of bone tissues. However, these proportions cannot be used to infer the relative frequency of the whale taxa in the past, because they are biased by the anthropic choices of the Magdalenian hunter-gatherers and by our sampling strategy. For example, the scarcity of the harbor porpoise remains might reflect a lower value of the bones of this smaller species for tool manufacture. Conversely, the large proportion of fin whales in our sample is biased by our decision to systematically analyze the whale bones from Santa Catalina, where this taxon dominates.

The taxonomic composition of the whale-bone assemblage gives nonetheless an indication of the way hunter-gatherers acquired the whale resources. The worked objects and unworked bone fragments we analyzed are mainly large cetaceans that predominantly forage offshore or in areas with deep water close to shore like the Bay of Biscay—namely fin whale, sperm whale and blue whale. It is extremely unlikely that these species would have been accessible to hunter-gatherers from the European Pleistocene other than through passive acquisition methods, such as the opportunistic acquisition of natural strandings or drift whales. In addition, the list of identified taxa also includes species whose ecology brings them substantially closer to the coast, namely gray whales, right/bowhead whales and harbor porpoises[46]. Bowhead whales and gray whales are the large cetaceans with the longest recorded history of active whaling, going back at least three millennia in arctic ecosystems[53,61]. However, there is no evidence that European Pleistocene hunter-gatherers had the necessary technologies for hunting these species, such as seafaring[62], or multibarbed points that could have been used as harpoons heads (barbed points appear in the local archeological record only after 16 ka cal BP[39]). Overall, then, the archeological evidence points towards an opportunistic acquisition of whale resources in the Bay of Biscay during the Magdalenian period, unlike in more recent periods where this same region became an area of active whaling[7,47].

Whales were likely familiar to coastal communities, and as such may well have played an important role in the Magdalenian cultural world. The Bay of Biscay's highly productive waters make it a cetacean hotspot today[41], and it was likely already the case during the Pleistocene. Along the Spanish coast in particular, water depth increases dramatically just a few kilometers from the coastline, bringing species that are typically found offshore, such as fin whales[63], unusually close. These species could have been observed from high points along the coastline spouting at a distance, and irregularly closer to shore as dead or moribund individuals. The more coastal right/bowhead whales, gray whales and harbor porpoise were likely even better known. Gray whales in particular are the most coastal of the whale species[46], spending summer at low-depth high-latitude feeding grounds, then migrate hugging the coastline to warmer lower-latitude calving areas in sheltered low-depth lagoons. Whereas in today's climate the Bay of Biscay and the Mediterranean would more likely correspond to calving grounds, it is possible that during the cooler Upper Paleolithic climate they would have corresponded to feeding grounds[49,64]—habitat reconstructions indicate that adequate shallow shelf habitat was

available there during the last glaciation[50]. In any case, they would have been conspicuously present at predictable seasons, and they are likely to have been part of the range of animals whose ecology would have been well known to Paleolithic hunter-gatherers.

## Whale products utilization and transportation

The whale remains analyzed in this study attest to two different utilization behaviors. The first one is the use of whale bones as raw material for the manufacture of implements; as suggested in previous studies, the choice of this raw material might have been motivated by the large dimensions of these bones, enabling the manufacture of longer implements, and maybe their specific mechanical properties[14,24]. The two main types of objects manufactured are projectile points and foreshafts, both relating to hunting equipment. Thirty-three projectile points were analyzed with ZooMS, and the results show that the range of species used for their manufacture is varied, with a relative majority of sperm whale (13 of the 31 points whose species could be identified, i.e., 42%; vs. 3-23% for each of the 5 other species identified). For the 11 foreshafts analyzed, the dominance of sperm whale is even more pronounced (8 of 11: 73%). It is uncertain whether this higher frequency compared to the other taxa reflects the relative abundance of sperm whales among stranded individuals or if it testifies to a preference for this species on the part of the Magdalenian carvers. The second hypothesis might be supported by the fact that, among the large cetaceans present in our sample, sperm whales are the only toothed species, displaying a highly characteristic, long, straight, toothed jawbone that may have been considered as a particularly desirable block of raw material. An interest in sperm whale teeth in the Magdalenian is also evidenced by the two carved specimens found at Las Caldas and Mas d'Azil[10]. Nevertheless, since the precise anatomical parts used for tool-making are so far indeterminate, this supposition remains speculative.

The second utilization behavior, evidenced at Santa Catalina, is the transportation of unworked whale bones to habitation sites. The bones were found in the form of fragments, most of which are too small and fragmented to be identified anatomically; still, some of them can be attributed to ribs and to vertebrae (cf. fragments of vertebral disks), evidencing the presence of portions from the trunk, and several display percussion notches indicating anthropic breakage carried out at or outside the site. The transportation of these elements requires explanation, as the species involved (fin whale and right/bowhead whale) are large, and the site was by then 4-5 km from the coast and 70 m up a steep cliff[65]. Indeed, bones of large whales are bulky elements, usually abandoned on the spot where the whale is processed rather than carried as riders attached to the meat and blubber. Their use as implements at the site (ribs for building huts, vertebrae as stools or anvils, etc.) is implausible given their degree of fragmentation, and evidence for their use as raw material for toolmaking is very scarce. Only one fragment shows traces of scraping, and most of the elements are very spongy, while the whale-bone objects known from other sites show a denser osseous tissue indicating the use of denser parts of the skeleton. An alternative explanation is their use as bone fuel for hearths. The use of the bones of terrestrial mammals as fuel is frequently documented in Paleolithic contexts[66], and activities linked to fire, including bone fuel, are intensive in level III of Santa Catalina (presence of charcoal, hearth-like structures, etc.)[67]. However, at Santa Catalina the whale bones we analyzed are not specifically affected by burning, nor concentrated near the hearths; if the collecting of whale bone at this site has to do with fire, it could only be as a reserve for later use. Alternatively, since living animal whale bones are very rich in fat, these large spongy bones might have been brought to collect the oil by letting them drip, or by crushing them[2]. This hypothesis would explain the fragmentation of the bones. It is consistent with the strong interest of Paleolithic groups for fat (e.g., the nearly systematic breakage of the bones for marrow[68]), and with

the fact that bone grease rendering is evidenced in the level III of Santa Catalina on the bones of ungulates[69].

Finally, although our archeological sample includes only bones and bone objects, their presence shows that hunter-gatherers also had access to other whale resources[2]. Whale skin, blubber and meat found numerous dietary and technical uses among populations of the past, and the transportation of (at least) whale skin to the habitat was already evidenced among European hunter-gatherers through the discovery of right whale barnacles at the sites of Las Caldas, Nerja and Cueva Victoria[11]. Furthermore, the identification of several species of mysticetes in our sample proves that Magdalenian hunter-gatherers also had access to baleen plates. The baleen of right and bowhead whales in particular have long been appreciated as a strong and flexible material with many practical uses in several populations[2]. In this case, the osseous material acts as a proxy for other resources with more rapid rates of degradation.

This study provided a chronological and taxonomic characterization of Late Paleolithic whale-bone industry. Its inception on the southern shores of the Bay of Biscay takes place within a more general trend towards the intensification of the exploitation of coastal resources at that period and in that region. The acquisition of whale products seems limited to the opportunistic utilization of beached or drift whales, but likely included the use of many resources besides bones–including baleen, the availability of which is now demonstrated, and whale oil, as possibly evidenced by the presence of unworked and fragmented bones at Santa Catalina. For this reason, stranding events certainly affected the groups' mobility patterns by acting as attractive spots. In this perspective, although whales might not have been the primary impulse for the increased interest in the seashore during the Magdalenian, their presence undoubtedly contributed to this evolution and strengthened it, as was suggested for other regions[70]. Furthermore, the taxonomic diversity and chronological depth of the cetacean populations identified in this study– both previously undocumented for this region at this time period– attest to the richness of the marine and coastal ecosystem of the Bay of Biscay at the end of the Paleolithic, showing how much it represented a favorable milieu for human settlement.

## Methods
### Sample selection, photogrammetry and sampling

The archeological specimens analyzed in this study are curated in 10 museums in France, 7 museums in Spain, 1 museum in the United Kingdom, 1 museum in Germany, and 3 curatorial repositories in France, under the responsibility of the Ministry of Culture, for collections from sites with excavations in progress. The location of each analyzed specimen can be found in Supplementary Data 1. Permission to access, study, borrow or export (when relevant), and sample the specimens was obtained from each institution, curator and/or excavation director following each local procedure (see detailed list in Acknowledgements section).

A total of 185 worked objects visually identified as made of whale bone are known from 31 Magdalenian sites (Supplementary Data 2: 165 objects from previous studies[14,24,40,71], plus 20 recently discovered through excavations or the reassessment of ancient collections). Minimally-invasive ZooMS analysis was tested on a subset of the samples, but proved to be unsuccessful (see below). In order to maintain the morphological integrity of the worked bone assemblage, a destructive ZooMS approach was applied to 83 of the 185 objects (i.e., 45%), with samples selected to ensure a representative selection of sites (n = 26) and of the objects' typology (these 83 objects include 3 ZooMS results previously published[36,37]). In addition, at Santa Catalina, 41 unworked bone fragments had previously (i.e., before this study) been attributed to cetaceans on a visual basis[12]. In this study, the bone assemblage of Santa Catalina was reassessed and sampled to include all fragments that could potentially be whale bone, even if the

morphological identification remained somewhat uncertain. A total of 90 unworked bone fragments from Santa Catalina (the 41 previously identified + 49 new specimens) were subjected to ZooMS analysis, mostly from archeological level III.

Half of the elements whose identification as whale bone was confirmed by ZooMS were subject to additional sampling for radiocarbon dating and stable isotope analysis: 37 of the 71 worked objects (52%), and 31 of the 60 bones from Santa Catalina (52%). The single object identified as made of porpoise bone through ZooMS analysis was a thin fragment of point, and permission was not granted to sample material sufficient for both ZooMS and $^{14}$C dating.

Before any sampling, photogrammetry was used to produce a 3D model of each worked object and each unworked bone fragment in its entirety. This was done for conservation reasons, in order to preserve a digital copy of the specimens before they were morphologically altered by the sampling. Each bone selected for sampling was photographed using a Nikon reflex with 40 or 60 mm lens, a portable light tent with different lights to provide diffuse light, and a turntable with markers for metric projection and angular notations. The Adobe® Photoshop® software was used for post-processing (color adjustment and background masking) and the Agisoft® Metashape® software was used for the photogrammetric process. For each artifact, both a dense cloud (for accuracy studies) and a textured mesh (for exports and measurements) were produced.

Depending on the shape of each element, sampling was done either: by cutting a piece from one of the extremities of the bone (using either a Stanley® mini hacksaw or a Dremel® rotary tool); by drilling with a Proxxon® Colt 2 pocket drill; or by coring using homemade diamond-coated core drills, 4–6 mm in diameter, mounted on a Moviluty® Minyflex® rotary tool. On average, 75.7 mg of bone was extracted from each specimen for ZooMS, with no significant difference between the worked objects and the unworked bone fragments. For radiocarbon dating only, the average mass of the sample was 95.3 mg on the worked objects, and 340.6 mg on the unworked fragments from Santa Catalina.

## ZooMS

ZooMS was first developed as a destructive analytical technique, requiring ~10–30 mg of bone or bone powder for analysis[26]. This project aimed to limit the destructive testing of unique, Magdalenian modified bone objects in order to preserve their morphology. As a result, for the bone objects, a minimally-invasive collagen sampling method was attempted which had been developed for analysis of historic parchments[72], and which had previously proved successful on ca. 500 years old modified bone objects from Quebec, Canada[73]. The 90 unworked cetacean bone samples, and 83 modified bone objects, also underwent a destructive ZooMS protocol to achieve accurate taxonomic identifications (see results above). The bone samples were analyzed using the destructive ZooMS methods listed below, based on a modified protocol as described in Buckley et al.[26].

**Minimally-invasive ZooMS.** The bone objects were gently rubbed with a clean piece of PVC eraser; collagen is transferred from the object to the eraser crumbs through the triboelectric effect[72]. The collected eraser crumbs from each sample were incubated for four hours at 37 °C in 75 μL of 50 mM ammonium bicarbonate solution (NH$_4$HCO$_3$, pH 8.0, AmBic) and 0.4 μg of trypsin. Trypsin activity was terminated using 1 μL of 5% TFA solution, and peptides were purified using C18 ZipTip® pipette tips and eluted with 50 μL of conditioning solution (50% acetonitrile, 0.1% TFA). 1 μL of eluted peptides was then spotted onto a Bruker ground steel target plate, to which 1 μL of matrix (α-cyano-4-hydroxycinnamic acid) was added. Each sample was spotted in triplicate along with calibration standards, and run on a Bruker Ultraflex III MALDI-ToF-MS. Triplicate spectra were averaged and analyzed using mMass software[74] and compared to a database of known collagen peptide masses (Supplementary Data 3)[75–78].

The minimally-invasive sampling technique proved unsuccessful for the majority of the tested samples. This was likely due to a combination of factors, including cetacean bone morphology and sample contamination. Due to the physical requirements of cetaceans, their bones often take on one of two forms depending on the element— either very dense and highly mineralized, or very porous and friable. Both bone conditions are problematic for minimally-invasive sampling methods as they result in reduced availability of surface collagen. Other studies applying the minimally-invasive eraser method also found limited success compared with destructive approaches[73,79,80]; as a result, a destructive and/or double extraction was performed on the samples.

**Destructive ZooMS.** Subsamples of bone or bone powder, sometimes several per specimen and ranging from 20 to 70 mg, were demineralized in 250 μL of 0.6 M HCl (4 °C). The acid was removed and samples were rinsed three times in 200 μL of 50 mM AmBic. Samples were incubated in 100 μL of AmBic for one hour at 65 °C to gelatinize the collagen; then 0.4 μg of trypsin was added to 50 μL of supernatant and incubated overnight at 37 °C. Trypsin activity was terminated using 1 μL of 5% TFA solution, and peptides were purified using C18 ZipTip® pipette tips and eluted with 50 μL of conditioning solution. MALDI-TOF-MS was conducted as described above.

**Destructive ZooMS double extraction.** Several samples analyzed using the minimally-invasive sampling method returned results that suggested sample contamination, likely as a result of a consolidant that had been added to the specimens post-excavation. This contamination particularly affected specimens from the Saint-Périer excavations (during the 1930's) at the site of Isturitz. While there is no mention in the reports for these excavations of the artifacts having been conserved with any type of consolidant, both minimally-invasive and destructive analyses of a number of these specimens consistently returned collagen identifications of cattle (*Bos* genus). Upon further inspection of the spectra, peptide markers that appeared to match those of various cetaceans could also be observed, however at a much lower intensity than the cattle fingerprint (Supplementary Fig. 91). This mixed signal in the collagen spectra led us to believe that consolidant contamination was a likely issue, as collagen-based glues made from cattle (and other animals) were known to have been used in conservation practices during the early 1900's.

A double extraction method was developed in an attempt to remove or reduce the signal from the contaminating consolidant (Supplementary Fig. 91)[81]. In this method, bone or bone powder was immersed in 250 μL of 50 mM ammonium bicarbonate solution (NH$_4$HCO$_3$, pH 8.0, AmBic) and placed on a shaker at room temperature for 3 h. Samples were then centrifuged briefly then incubated at 65 °C for one hour to gelatinize. After incubation, the samples were centrifuged for one minute and the AmBic was transferred to a new eppendorf as the first extraction. Subsequently, 250 μL of 0.6 M HCl (4 °C) was added to each of the original eppendorfs containing the bone chips/powder, then vortexed and left to demineralize at room temperature for four hours. The acid was removed and samples were rinsed three times in 200 μL of 50 mM AmBic. 100 μL of AmBic was then added to the samples, followed by incubation for one hour at 65 °C to gelatinize the collagen for the second extraction. Both the first and second extraction underwent digestion, purification and MALDI-TOF-MS analysis as described above. While this modified protocol resulted in lower peptide recovery, it enabled us to enhance the collagen signal from the actual 'host' animal (in most cases, cetacean collagen peptide markers) while also confirming our suspicions regarding the use of a cattle-based consolidant (Supplementary Fig. 91).

With the exception of 7 samples (Hum-1 to Hum-18) for which collagen extraction was carried out at the EvoAdapta facility and MALDI-TOF-MS analysis at the University of York, all samples were analyzed for ZooMS at the Institute of Environmental Science and Technology (ICTA-UAB) of the Universitat Autònoma de Barcelona.

## Bone collagen extraction protocol

All glassware was cleaned using the following procedure: submerged in Decon 90 at 90 °C and cooled overnight, washed three times with DI water, submerged overnight in 10% HCl, washed three times with Milli-Q. After drying, the glassware was wrapped in aluminum foil to prevent dust contamination during storage and finally baked out in the oven at 450 °C for 5 h.

The bone collagen extraction protocol based on Longin[82] followed the following steps: Bone samples were first demineralized in 0.2 M HCl for several days, with mechanical and visual checks. The acid was renewed several times during this step. The samples were then rinsed three times with Milli-Q, submerged for 20 min in 0.1 M NaOH (with new NaOH being added for another 20 min if discoloration appeared), before being rinsed again three times with Milli-Q, submerged into 0.1 M HCl for 10 min, then rinsed again three times with Milli-Q. In the next steps, the samples were gelatinized in weak (pH 3) HCl at 90 °C until dissolution, filtered with glass filter units (mesh size 10–20 μm), frozen with liquid nitrogen and lyophilized in clean (baked out) vials.

## Radiocarbon dating

As anomalous peaks were observed in FTIR-ATR spectra of several samples, most likely indicating the presence of glue or consolidant, the XAD resin approach was applied to eliminate this type of contamination. This approach was developed by Stafford et al.[83–85], is also described in van der Sluis et al.[86]. Lyophilized collagen samples were dissolved in 1 mL of (sub boiling distilled) 6 M HCl in 10 mL borosilicate tubes with PTFE lines caps before being hydrolyzed at 110 °C for 24 h. The hydrolysate was then passed through preconditioned XAD columns fitted with a filter frit at the bottom, filled with ±100 μl (circa 1 cm) of XAD 2 resin slurry and covered with the top filter frit. The latter was pushed down in order to remove air bubbles. The columns were then washed with 20 mL of 1 M HCl and preconditioned with 10 mL of 6 M HCl. After the sample hydrolysate had passed through, the column was washed with 1 bed volume of 6 M HCl in order to collect any amino acids in the void space and added to the collected sample. Samples were dried in small open beakers on the hotplate in the fume hood and were rinsed with Milli-Q to remove any leftover HCl by evaporation. The samples were then transferred in 200 μl (7-8 drops) of Milli-Q to combustion tubes using glass Pasteur pipettes, frozen and lyophilized.

Samples were connected to the $CO_2$ extraction line in the radiocarbon laboratory of the Muséum national d'Histoire naturelle. After adding 900 mbar pure $O_2$, samples were combusted at 900 °C for a duration of 10 to 20 minutes in the presence of a baked out silver strip (10 mg) to remove contaminants, cleaned on the $CO_2$ extraction line (water trap, NOx oven fitted with copper and silver fiber wool), then volume calculated. The vacuum on the line reached $10^{-6}$ mbars. The $CO_2$ gas sample was transferred to a fully-automated $H_2$ reduction line using iron as a catalyst, where the vacuum reached $10^{-7}$ mbars.

Samples were run alongside standards of oxalic acid and phthalic acid. Graphite targets were pressed and analyzed on the same day with the ECHo-MICADAS at the Laboratoire des Sciences du Climat et de l'Environnement (LSCE) in Gif-sur-Yvette, France. Data reduction was performed using the BATS software (version 47)[87]. The first scans were discarded to eliminate possible contamination of the target with ambient atmosphere between target pressing and AMS measurement. Radiocarbon ages were calculated from F[14]C[88], which is corrected for background and isotopic fractionation. Measurement parameters such

as $^{12}C$ current and $^{13}CH$ current were monitored during $^{14}C$ measurement. Time, current and isobar corrections were made before validation. For each individual run, normalization, correction for fractionation and background corrections were applied by measuring the oxalic acid II NIST standard, its $^{13}C/^{12}C$ ratio and the chemical blanks. The standard deviation of the blanks is generally less than 10% but an overestimated 30% is imposed to the blank value in order to take into account a potential variability of the contaminant which could be added during the sample preparation.

Six samples (Hum4, Hum5, Hum8, Hum9, Hum14 and Hum18) were chemically pretreated and prepared at the Higham Lab, University of Vienna, Austria. Sample pretreatment for these samples follows collagen extraction outlined in Brock et al.[89]. Three samples (Hum9, Hum14 and Hum18) were combusted offline, graphitized using an AGE graphitization unit, and measured at the VERA (Vienna Environmental Research Accelerator) AMS facility, University of Vienna. For details on aspects of target preparation and AMS measurement see Steier et al.[90] and Golser & Kutschera[91]. Three other samples (Hum4, Hum5 and Hum8) were measured at the Keck AMS facility, University of California at Irvine. For details on aspects of target preparation and AMS measurement see Santos et al.[92] and Beverly et al.[93]. These six samples have R-numbers and VERA or UCIAMS numbers instead of MUSE-numbers.

## Collagen quality control

As diagenetic alteration can affect stable isotope values as well as radiocarbon ages, only results from good quality collagen with an atomic C:N ratio between 2.9 and 3.6 ought to be used[94]. However, studies by van Klinken[95] on archeological bones and Guiry and Szpak[96] on modern bones have suggested that values outside the 3.0-3.3 range for mammals may result in distorted isotopic compositions due to contamination by non-collagenous carbon. We compared the values for whale species that agree with different atomic C:N ratio ranges (2.9–3.3 and 3.3–3.6) (Supplementary Fig. 96). In these figures there is considerable overlap between samples with an atomic C:N ratio up to 3.3 and 3.6, only the gray whale samples show lower carbon stable isotope values (circa 1‰), although this could also be due to the small sample size. We believe most of the variation visible here is due to individual variation between the whales, rather than diagenetic alteration. Whales are exceptionally large (marine) mammals, thus resulting in a larger variation in stable isotope ratios in their bone collagen due to their longevity and slow bone turnover time. Samples with atomic C:N ratios between 2.9–3.6 were included in the discussion of the stable isotopes results. Additionally, stable isotope analysis was done on collagen, while $^{14}C$ dating was performed on collagen after XAD resin treatment, which would have removed any, if present, remaining contamination. When examining the $^{14}C$ ages against atomic C:N ratios from a single site (Santa Catalina) (Supplementary Fig. 97), there is no evidence of a correlation that might indicate contamination.

## Calibration and reservoir correction

Dating of marine animals needs to take into account the marine reservoir effect: given that oceans are depleted in $^{14}C$ in relation to the atmosphere, conventional radiocarbon dates for marine organisms appear older than for coeval terrestrial individuals[97]. Some studies use sample pairing between coeval terrestrial and marine taxa to aid interpretations of reservoir ages at archeological sites[98]. This was not possible in our case, because our samples are all from Pleistocene cave and rockshelter sites with complex stratigraphy (in which each excavated layer is the result of an accumulation of multiple occupation episodes) and most of our samples are from ancient excavations with poor stratigraphic resolution. We thus relied on the correction for the reservoir effect implemented for marine samples in OxCal 4.4, which is based on the Marine20 calibration curve[99].

In addition, dating of marine organisms can be further refined by taking into account the fact that the marine reservoir effect varies across oceanic masses, being for example stronger (i.e., leading to seemingly older organisms) at higher latitudes and in deeper waters[100]. This deviation from the standard (globally averaged) marine reservoir effect is expressed as a DeltaR term. In the case of our whale specimens, however, it is not the coastal waters adjacent to the archeological sites that matter, but the water mass (or masses) where whales fed. The whale species identified in this dataset are very diverse in terms of their feeding ground locations and feeding depths. For example, previous studies estimated DeltaR values: between −168 and 504 ± 60 years for sperm whales (that feed in deep waters but can be found in either high latitudes or sub-tropical seas[101]); at 24 ± 58 years for bowhead whales (that forage in polar/sub-polar waters close to the sea surface[102]); and about 350 years for a fin whale (a species that forages at depths up to 400 meters and migrates vast distances[103]). Intra-specific variability in DeltaR values renders it difficult to obtain meaningful values per species. We have not found published estimates of DeltaR for gray whales, but the foraging strategy of this species (which feeds on invertebrates found in shallow sedimentary environments and migrates over very long distances) renders it particularly susceptible to intraspecific variation in DeltaR, especially as it will be dependent on the type of sediment (molluscs feeding on calcareous sediments appear over 2000 years older than suspension-feeding ones[104]).

In addition to these difficulties, DeltaR values published in the literature from modern (Holocene) whales need to be approached with caution when applied to Pleistocene samples. Indeed, the spatial variation in the reservoir effect of water masses likely varied over time, in response to changes in ocean circulation. The foraging grounds of whales are also likely to have been different, given that in the Late Pleistocene, the study region was substantially cooler. For these reasons, we have opted for not attempting to apply a species-specific DeltaR based on information from modern whales, and instead use a single DeltaR as suitable as possible to the period and region of our samples.

Choosing the most suitable reservoir correction required careful consideration of various parameters. Foote et al.[105] used a DeltaR = 0 to correct for the reservoir effect in bowhead whales during the Pleistocene. However, this no longer seems suitable since the update of the Marine20 calibration curve, as there are large fluctuations in pre-Holocene reservoir ages[97]. Monge Soares et al.[106] published a DeltaR = −117 ± 70 yr for the Late Pleistocene based on three samples pairs (shells and bones/charred wood) from the Cantabrian coast. Mangerud et al.[107] showed that there is little difference between the DeltaR from whales and mollusks found at the same location. However, since the update of the Marine20 calibration curve, the DeltaR from Monge Soares et al. needs updating as well. Monge Soares et al. used three paired [14]C ages for the calculation of their Late Pleistocene DeltaR (terrestrial [14]C ages: 15 656 ± 75, 19 710 ± 120 and 23 040 ± 50 yr BP). Considering the ages of our samples, we decided it would be most suitable to use the two youngest paired [14]C ages for an updated DeltaR, because the third sample pair was considerably older. These two paired Late Pleistocene [14]C dates from Monge Soares et al. were used in the online application[108] to calculate updated DeltaR values. These new DeltaR values are statistically the same (0.037 ($\chi$2:0.05 = 3.84)). The weighted mean of these DeltaR and standard deviation are calculated according to: http://calib.org/marine/AverageDeltaR.html, which gives a new Delta R of -448 ± 39 yr. The calibration was done in OxCal 4.4 (Supplementary Figs. 92 through 95, and Supplementary Code 1)[99].

Additionally, using all North Atlantic whales (various species) in the Marine Reservoir Correction Database we get a DeltaR of −168 ± 53, compared with the DeltaR of −117 ± 70 from Monge Soares et al. for the Holocene. This means that the latter slightly overestimate the reservoir age, making samples slightly younger. This may not be the case for the

Late Pleistocene samples but it does strengthen our idea that the material used for paired dating in this region by Monge Soares et al. is not that far off from whales themselves. However, the DeltaR used here is not optimal for two species in this dataset, namely sperm whales due to their large variation in ΔR (between −168 and 504 (± 60) [14]C yr[101]) and gray whales due to feeding in shallow waters where they might ingest calcareous sediment with different [14]C ages[104].

Heaton et al.[100] propose a specific calibration of marine samples from polar regions, essentially calculating a range between a minimum and maximum DeltaR. The minimum is based on the Marine20 curve and the maximum is based on the latitude of the sample and the Large Scale Geostrophic Ocean General Circulation Model (LSG OGCM), which is suggested to use for samples from outside ~40°S–40°N during glacial periods. Our samples are found between of 43–44 °N, although the whales could of course have been feeding at higher latitudes. However, we do not have the exact location.

We state that our results provide evidence of whale bone working from 20ka BP, which is based on two fin whales. If the absolute max DeltaR would be applied, these 2 fin whales would be between 2000-1500-1000 years younger (Fig. 3 in Heaton et al.[100], although there is a substantial variation between 71 °N, 88 °N and 53 degrees °N, and again we do not know where these animals were feeding). It would be more realistic to assume sea ice most likely covered a large part of the North Atlantic waters, extending much further south than it does today. If so, most whales would not have been feeding too far north due to sea ice coverage (unlikely reaching the 71 and 88 °N, except for beluga's and bowheads). Additionally, modern fin whales feed in various locations, have summer and winter feeding grounds, and can also adapt their feeding strategy to minimize interspecific competition, suggesting that the maximum value would not be a realistic assumption. Combining all of these assumptions, it is plausible that the maximum DeltaR was ~800 yr, although its value remains extremely uncertain.

Given that the results we present here reconstruct the chronology of whale acquisition using very broad time slices of 1.5 millennia (Fig. 2), it is unlikely that our conclusions are significantly affected by an uncertainty on DeltaR in the range of a few centuries. It does however mean that caution is needed when interpreting individual radiocarbon dating results. These can be improved in the future as more information becomes available regarding suitable corrections for the marine reservoir effect.

## Isotope ratio mass spectrometry

Bone collagen samples (320–380 μg) were weighed into tin capsules (5 × 8 mm) and analyzed with a Thermo Scientific EA Flash 2000 coupled to a Delta V Advantage isotopic mass spectrometer in the Service de Spectrométrie de Masse Isotopique (SSMIM) in the Muséum national d'Histoire naturelle, Paris, France. Isotopic values of all samples were measured relative to the laboratory standard alanine, which has a reproducibility of 0.3 wt% for N and 0.6 wt% for C. The $\delta^{13}$C and $\delta^{15}$N values are reported relative to the VPDB and AIR, respectively. Three primary standards were used to calibrate the Alanine for $\delta^{15}$N air: IAEA USGS-25 (ammonium sulfate), IAEA N1 (ammonium sulfate) and IAEA N2 (ammonium sulfate), while one primary standard (from IAEA) was used to calibrate the Alanine for $\delta^{13}$C V-PDB: IAEA 600 (caffeine). Analytical precision is ± 0.2‰ for both $\delta^{13}$C and $\delta^{15}$N values. Five samples were run in duplicate and the difference fell between 0-0.3‰ for $\delta^{13}$C ratios, and 0–0.34‰ for $\delta^{15}$N ratios.

Subsamples (0.2–0.3 mg) of extracted bone collagen from six samples (Hum4, Hum5, Hum8, Hum9, Hum14 and Hum18) were taken for the analysis of isotope ratios of carbon and nitrogen by elemental analyzer-isotope ratio mass spectrometry (EA-IRMS; Thermo Scientific EA-Isolink with a Flash 2000 coupled to a Delta V Advantage isotope ratio mass spectrometer) in the Silver Laboratory (Large-Instrument Facility for Advanced Isotope Research) at the Center of Microbiology

and Environmental Systems Science of the University of Vienna. Stable isotope values of all samples are measured relative to the laboratory standard alanine, which has a reproducibility of 0.07 wt% for N and 0.1 wt% for C. $\delta^{13}C$ and $\delta^{15}N$ values are reported relative to the VPDB and AIR standard, respectively, and are measured with an analytical precision of $\pm 0.1‰$ for both $\delta^{13}C$ and $\delta^{15}N$ values. Acceptable stable isotope ratios have an atomic C:N ratio that falls between 2.9–3.6[94].

The difference in $\delta^{13}C$ ratios between the balaenidae, fin and gray whales suggests these whales were feeding in different bodies of water, which in turn may have affected their radiocarbon ages. To investigate this, we plotted the conventional $^{14}C$ age against the $\delta^{13}C$ ratio from these whale bones (Supplementary Fig. 98). We found no correlation between these two variables ($R^2 = 8 \times 10^{-5}$). The ages of the three whale groups studied broadly overlap: balaenidae (right/bowhead whales; $n = 8$) are dated from 16–16.5ka to 17.1–17.8ka cal BP; fin whales ($n = 31$) are dated from 13.2–13.5ka to 19.5–20ka cal BP; gray whales ($n = 4$) are dated from 16.3–16.9ka to 17.4–18ka cal BP.

### Reporting summary
Further information on research design is available in the Nature Portfolio Reporting Summary linked to this article.

## Data availability
All data needed to support the conclusions of the paper are presented in the paper, the Supplementary Information, Supplementary Data, and Supplementary Code 1. The location of each analyzed specimen can be found in Supplementary Data 1. Source Data for Figs. 1 and 2, and for Supplementary Figs. 92 through 95, can be found in Supplementary Data 1. Source Data for Fig. 3 and for Supplementary Figs. 96 through 102 can be found in Supplementary Data 4. Collagen peptide mass spectra have been publicly deposited in the Nakala Repository under following link: https://doi.org/10.34847/nkl.dbbfl6fw.

## Code availability
Code used in OxCal 4.4 can be found in Supplementary Code 1.

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

## Acknowledgements

We thank the people and institutions that granted access and sampling permissions for the material studied: I. Alonso Garcia and M. A. Pedregal Montes (Museo arqueológico de Asturias); F. Bon, L. Bruxelles, M. Jarry and C. Pallier; J. Cook and C. Lucas (British Museum); J. Darricau (site d'Isturitz) and C. Normand; L. Ducamp (Maison de la dame, Brassempouy); P. Fatás Monforte (Museo nacional y centro de investigación de Altamira); G. Fleury (Muséum d'histoire naturelle de Toulouse); A. Fort, R. Nespoulet and M. Lebon (Musée de l'Homme); N. Fourment (Musée national de Préhistoire); S. Fraile Gracia (Museo nacional de ciencias naturales); E. Galán (Museo arqueológico nacional); I. García Camino (Arkeologi museoa in Bilbao); D. Haro-Gabay (abbaye d'Arthous); the musée de Lespugue; V. Merlin-Anglade (Musée d'art et d'archéologie du Périgord); R. Ontañón (Museo de Prehistoria y Arqueología de Cantabria); J. Primault; L. Rodriguez (Musée de Borda); C. Saint-Martin and P. Alard (musée du Mas d'Azil); S. San Jose Santamarta (Gordailua center); C. Schwab (Musée d'archéologie nationale); A. Simonet; M. Street (RGZM) and C. Langley; and É. Tartar. Thanks to L. Agudo Pérez and A. Cicero Cabañas for their help with the ZooMS analyses, to C. Merlet for his help with the material from Duruthy, to P. Reimer for her help on calculating the new DeltaR for the reservoir correction, and to D. Fiorillo (SSMIM, Service de Spectrométrie de Masse Isotopique du Muséum) for analyzing samples on the IRMS. We thank the York Center for Excellence in Mass Spectrometry (University of York) and the Laboratori de Proteòmica CSIC (Universitat Autònoma de Barcelona), for allowing access to their MALDI-TOF-MS. This work contributes to the ICTA-UAB María de Maeztu Program for Units of Excellence of the Spanish Ministry of Science and Innovation (CEX2019-000940-M), and EarlyFoods (SGR-Cat 2021, 00527). This study was funded by projects HumAntler (PCI2021-122053-2 B) (A.L.), Whalebone (HORIZON-MSCA-2021-PF-01-101059605) (A.L.) and PaleoCet (ANR-18-CE27-0018) (J.-M.P., A.Z.). This work is dedicated to É. Campmas and G. Marchand.

## Author contributions

Project conception, funding acquisition, project administration: J.-M.P., A.L., and A.Z. Identification and sampling of the archeological material: J.-M.P., A.L., F.-X.C., and J.T. ZooMS methodology and analysis: K.McG., C.S., and L.T.I. Radiocarbon and stable isotopes methodology and analysis: L.G.S., A.Z., F.T., and O.T. Photogrammetry and 3D models: F.B. Historical literature analysis: A.C., A.S.L.R. Excavation and scientific responsibility of archeological sites, access to archeological collections and samples provision: E.A.F., E.B., F.-X.C., M.D., E.D.M., C.H., A.B.M-A, and M.R.V. Writing: J.-M.P., A.C., A.L., K.McG., A.S.L.R., C.S., L.G.S., and A.Z. All authors revised and approved the manuscript.

## Competing interests

The authors declare no competing interests.

## Additional information

[1]Department of Prehistory and Institute of Environmental Science and Technology (ICTA-UAB), Universitat Autònoma de Barcelona, Bellaterra, Spain. [2]BioArchéologie, Interactions Sociétés Environnements (BioArch), UMR 7209, Muséum national d'Histoire naturelle, CNRS, Paris, France. [3]Department of Evolutionary Anthropology, University of Vienna, Vienna, Austria. [4]Human Evolution and Archaeological Sciences, Vienna, Austria. [5]Grupo I + D + i EVOA-DAPTA, Universidad de Cantabria, Santander, Spain. [6]De la Préhistoire à l'Actuel: Culture, Environnement et Anthropologie (PACEA), UMR 5199, CNRS, Université de Bordeaux, Pessac, France. [7]CEFE, Univ Montpellier, CNRS, EPHE, IRD, Montpellier, France. [8]GIR PREHUSAL, Universidad de Salamanca, Facultad de Geografía e Historia, Departamento de Prehistoria, Historia Antigua y Arqueología, Salamanca, Spain. [9]Travaux et Recherches Archéologiques sur les Cultures, les Espaces et les Sociétés (TRACES) UMR 5608, CNRS, Université Toulouse Jean Jaurès, Toulouse, France. [10]Sociedad de Ciencias Aranzadi, Donostia-San Sebastian, Spain. [11]Office du patrimoine et de l'archéologie du canton de Neuchâtel, section Archéologie, Laténium, Hauterive, Switzerland. [12]Ministère de la Culture, Service Régional de l'Archéologie de Nouvelle-Aquitaine, Limoges, France. [13]Departamento de Historia, Universidad de Oviedo, Oviedo, Spain. [14]Université de Franche-Comté, UMR 6249 Chrono-environnement, Besançon, France. [15]Laboratoire des Sciences du Climat et de l'Environnement (LSCE/IPSL), UMR 8212, CEA-CNRS-UVSQ, Université Paris Saclay, Gif-sur-Yvette, France. [16]Section for Molecular Ecology and Evolution, Globe Institute, University of Copenhagen, Øster Farimagsgade 5, Copenhagen, Denmark. [17]Department of Anthropology, University of British Columbia, Vancouver, Canada. [18]These authors contributed equally: Krista McGrath, Laura G. van der Sluis. [19]These authors jointly supervised this work: Antoine Zazzo, Jean-Marc Pétillon. ✉e-mail: jean-marc.petillon@cnrs.fr

