## [Transparent Peer Review file · Nature Communications]

Late Paleolithic whale bone tools reveal human and whale ecology in the Bay of Biscay

Corresponding Author: Dr Jean-Marc Petillon

Version 0:

Reviewer comments:

Reviewer #1

(Remarks to the Author)

This study taxonomically identifies and radiocarbon dates whale artefacts and ecofacts from sites near to the Bay of Biscay. The marine reservoir correction that the authors implemented date the samples somewhere between 16 - 20 kya. The radiocarbon results are rather striking, providing evidence for some of the earliest known utilisation of whale faunal remains in Europe at this scale. However, there are some important implications of using a blanket reservoir correction for different whale species that need to be considered and the associated caveats discussed before this manuscript should be accepted for publication. Whilst additional methodological information on the stable isotope analysis and improved interpretation of the stable isotope results is also necessary. Despite the major revisions required, I believe the interesting findings from this study deserve consideration for publication, as they contribute to an ever-expanding pool of evidence of early interactions between humans and marine taxa, with implications on our understanding of how humans and coastal habitats have co-existed for millennia far beyond what many researchers have imagined to date.

Radiocarbon dating considerations:

- Differences in the foraging strategies of the different whale species and the implications of this on marine reservoir corrections MUST be considered and discussed, especially as this could bring dates more in-line with previous evidence from this region.
 - o Sperm whales forage in deep waters and existing deltaR values can vary between -168 and 504 +/- 60 years. See Beck et al. 2022. <https://doi.org/10.1017/RDC.2022.79>
 - o Bowhead whales forage in polar/sub-polar waters close to the sea surface, the deltaR for Bowheads has been estimated at 24 +/- 58 years. See Pienkowski et al. 2022. <https://doi.org/10.1111/bor.12606> Similar foraging behaviour is observed for Eubalaena.
 - o Fin whales forage in water depths of 100 – 400ms and migrate vast distances. They are known to forage predominantly in polar and sub-polar waters, but some individuals are known to also forage at lower latitudes, see Buss et al. 2022. <https://doi.org/10.1007/s00227-022-04131-x>
 - o Grey whales forage in shallow sedimentary environments and migrate vast distances. Please consider the implications of this on the reservoir corrections and the Portlandia effect. See Dyke et al. 2013 <https://doi.org/10.1111/j.1502-3885.2012.00256.x>
 - o Harbor porpoises forage locally in rocky coastal environments, although they seasonally also forage further offshore, I would estimate that this species is least likely to be affected by the deltaR choice used here.
- Many studies use sample pairing between terrestrial and marine taxa to aid interpretations of reservoir ages at archaeological sites. See Dury et al. 2021. <https://doi.org/10.1177/09596836211041> - it is not clear to me why other chronological evidence (e.g. terrestrial fauna or shells from the same sites) were not used for comparison. Please explain why only whales were used, it seems an odd choice given all the challenges with using marine mammals for marine reservoir corrections and dating, especially for migratory species that feed in multiple bodies of waters at varied depths.
- Known deltaR corrections for marine taxa foraging in polar waters relative to lower latitudes can vary by up to approximately 800 – 1200 years due to the upwelling of older water in these regions. This will impact the surface waters less than the deeper waters (e.g. a fin whale foraging in polar waters will look older than a bowhead whale). Given these considerations, the results of your findings may differ by over a millennia for a fin whale foraging at higher latitudes. However, we cannot guarantee that fin whales always forage in the same body of water given their migratory foraging behaviours and differences between individuals of the same population (e.g. evidence of foraging at lower latitudes (Buss et

al. 2022) and during winter (Silva et al. 2019). Although this may impact how definitive the dates you have provided here are, I do not think these considerations will impact the overall result from the manuscript, but the uncertainty around the MRE given the taxonomic groups you have studied MUST be incorporated in the text.

Stable isotope considerations:

- The reason for including the SIA results is currently not clear to the reader. Are all these samples from the same time period? Or a few millennia apart. Where these samples dated?
- The carbon isotope ratios suggest the balaenidae, fin whales and gray whales were likely feeding in different bodies of water (if from the same time period), please see comments on radiocarbon considerations above. The relationship between the radiocarbon dates and the SIA carbon data should be considered. The Balaenidae carbon values may potentially indicate younger samples (if these have been dated), is that something you observe? I would try and link the interpretation from the SIA data and the radiocarbon dates more in the discussion.
- Please be careful with wording, relative foraging preferences can be inferred from stable isotopes but exact diet cannot (e.g. the prey chosen) be inferred without other known ecological data/information. Please amend throughout the text.
- Fin whales often forage at a slightly higher trophic level relative to Balaenidae, but here they have lower nitrogen isotopes, why? Is it because the fin whales were feeding at higher latitudes where baseline isotopes differ (lower nitrogen likely?).
- Please include additional information on SIA methodology (see comments in attached manuscript).

Introduction considerations:

You state that understanding the why and how of this Magdalenian whale exploitation is thus of crucial importance. However, I do not feel your approach here sheds light on the why and the how, but does provide important evidence on when and which taxa may have been utilized by humans in this region. I believe the introduction could use some tightening up around the when and which taxa, and why this is an extremely unique and important finding as the main aim of this research. Rather than precluding towards the why and how, without providing much evidence for either of these questions.

Minor corrections (see attached file)

Reviewer #2

(Remarks to the Author)

The authors present a manuscript looking at reconstructing prehistoric exploitation of large cetaceans using ZooMS and isotope analyses of worked bone material.

However, overall it was left unclear as to whether this study is investigating palaeoecology OR human seashore interactions – the latter is much more nuanced, and not well achieved in the presented work... 'Past ocean ecology' to many people would mean a much broader topic than the cetaceans discussed here...

The 'multiproxy' analysis is not targeted at answering one question, but seems to be a collective of different methods, answering different questions, with the only thing in common being that they are contemporary...

Commenting on the manuscript more specifically:

The abstract suggests that the finding of five species of large whales 'significantly' expands the range of known species, but makes little effort to clarify what species and why this is potentially interesting. Likewise, the brief statement on stable isotope results that 'reflect species-specific preferences' is particularly uninformative.

The one clear statement relates to the radiocarbon dates, suggesting that the study involves the oldest known evidence of whalebone working, but lacks clarification on how we can have confidence that it is the activity that is being dated, rather than the potential for re-worked older material.

Introduction:

Line 116-118 – needs citation for whales being key part of subsistence of prehistoric coastal populations, and how this is evidenced – if this statement stems from the sources mentioned in subsequent sentences, then restructure the paragraph accordingly

Line 119 – It seems important to clarify how archaeologists can distinguish between the exploitation methods briefly mentioned – for example, scavenged vs hunted, and the extent of organization of the latter.

Line 121 – This is before moving on to the issue of site loss by nature of many being coastal

Line 128 – unclear how the unworked whale bones have contributed to the evidence of the beginning of whale exploitation – perhaps clarify that this relates to fragmentation patterns with human agency implied, if this is the point being made

Unclear what distinguishes the first two paragraphs in terms of their message, structured awkwardly in that the first goes into details to introduce whale exploitation, and the second more widely at coastal exploitation

Likewise, the start of the third paragraph is awkward and seems as though would be better as a continuation of the previous paragraph

Line 161 – awkward/incomplete sentence (Dietary data and feeding ecology of these whales...)

Line 173 – Unclear why neither of the citations for ZooMS in its first instance refers to the first publication of the method, or its first application to the study of cetaceans

Line 173 – nor is ZooMS defined

Line 175 – statement “This work enabled us to assess the range of whale species...” should be toned down somewhat, given that there are clear biases against the presented range reflecting the ‘true’, full range of whale species present – other than what could be expected for known palaeobiogeographic distributions of such species

Line 190 – ‘while 8 were identified as large terrestrial mammals’ is an interesting point that should be more detailed – i.e., what might be causing these misidentifications

Line 191 – the fact that 90% of ‘macroscopic visual attribution’ to the identification as a ‘whale’ is interesting from a scientific perspective, but not particularly informative to the wider story of past ecology and/or human seashore interaction, whichever of these the study aims to address

Line 193 – The different levels of initial morphological identification were not particularly clear on first reading, so this sentence about positively classified and likely made of, etc., was difficult to follow – should be re-written or better introduced

Line 196 – It at first seems odd that you have more correct ‘identifications’ with the worked material, than you get with the naturally fragmented material

Line 208 onwards – some of this reads as Discussion rather than Results

Line 211 – would be useful to state when the first known appearances of these newly identified taxa, fin, grey, blue, right, and bowhead whales, are to the North Atlantic

Line 219 – this whole section on stable isotope analyses seems to not add anything to the story of ancient human seashore interactions, though perhaps this whole study is more of interest to palaeoecology – if so, would be interesting to clarify this

Line 225-227 – This sentence also reads more like it belongs in the Discussion, but it is unclear if this information is even appropriate – if so, it would benefit a re-structure and re-write – the high number of citations within this paragraph make it clear how out of place it is in the Results

Line 245 – so here it is made clearer that there is chronological support linking the stratigraphy with the radiocarbon dates, which could come across better earlier on

Figure 2 – Although it is clear how these isotope ratios differ between species, as would be expected given differences in diet, it remains unclear what the message is with this study – i.e., if this is to look at what ‘past ocean ecology’ was, it would be more interesting to make comparison with values from extant animals

Line 280 – so if this is more a confirmation of early evidence of cetacean bone working, it becomes more important to give this study novelty by discussing the significance of the new taxa found in this region

The fact that the Discussion starts with radiocarbon dating is suggestive of restructuring mentioned above

Numbers on Figure 3 are difficult to read

Line 323 – ‘still had access to whale bone’ and ‘still had the skills’... perhaps the authors here could clarify why populations may lose such skills, particularly given that ‘they do not differ from those used in the osseous industry in general’

Line 376 – As referred to, one of the greatest challenges in making interpretations relating to human-cetacean interaction is that of scavenged vs hunted – the authors claim that this is irrelevant for their interpretation relating to how ‘well placed these species must have been’, but the following sentences are not particularly convincing as to why this is the case.

Reviewer #3

(Remarks to the Author)

The article is an important contribution to the literature on hunter-gatherer-fisher populations and an example of early cetacean use during the Upper Paleolithic. The authors provide the reviewers and readers with sufficient data to contextualize their study and results. In addition, the manuscript is a new and vital data point in the growing ZooMS literature. Their use of various ZooMS methods and comparison of the results are important contributions to the literature.

I have questions regarding the Delta R applied, given the time depth and other related issues the authors discuss. Still, I recognize that calibration and radiocarbon dating of any marine organism, let alone cetaceans, can be complicated.

I did encounter minor grammatical and spelling issues in the draft, but these can be quickly remedied if accepted.

Overall, I greatly appreciate the author's contribution and look forward to seeing it published.

Reviewer #4

(Remarks to the Author)

Dear authors,

I applaud your efforts in conducting this multi-approach study of some of the oldest evidence for prehistoric whale exploitation. The article is well written and, with some technical clarification and editing, will make a good contribution to the journal. See my comments below:

Diagenetic Alteration and Collagen Quality Criteria: The range (2.9 and 3.6) of C:N ratios established by DeNiro (1985) has long been the standard for good collagen preservation. However, studies by Klinken (1999) on archaeological bones and Guiry and Szpak (2020) on modern bones have suggested that values outside the 3.0-3.3 range for mammals may result in distorted isotopic compositions due to contamination by non-collagenous carbon. I suggest removing the samples with C:N above 3.3 and reanalyzing the data set. Alternatively, you could assess collagen quality by comparing $\delta^{13}\text{C}$ and $\delta^{15}\text{N}$ versus C:N for your whale samples (see Guiry and Szpak, 2021).

Stable isotope analyses: The methodology section states that "three samples were analyzed at the Silver Lab at the University of Vienna", but does not explicitly state where the remaining samples were analyzed (apologies if I missed it). This may be important as different labs may use different pretreatments or analyses (instrumentation or data calibration), which may also cause variation in the isotope ratios (e.g., Pestle et al., 2014). Did you consider this?

Interpretation of $\delta^{13}\text{C}$ and $\delta^{15}\text{N}$ values: The authors correctly point out the significant sources of variation affecting the stable carbon and nitrogen isotope composition of their whale specimens. I agree that it is extremely challenging to carefully evaluate them; however, a more thorough assessment of the patterns of stable isotope composition of the specimens may help move this issue forward. You focus your discussion on describing the results and commenting on how they might match expected patterns of niche partitioning among cetacean species. However, you don't explicitly show the patterns of isotopic niche partitioning among living species, so their use as a reference seems a bit weak. How do patterns of the stable isotope composition recovered for the archaeological assemblage (Figure 2) contrast with those exhibited by modern representatives in the area? I am not proposing a quantitative comparison of the stable isotope composition of archaeological vs. extant assemblages (that would be problematic due to shifting baselines, differences in tissue type, etc.). Instead, I am primarily interested in whether the general distribution in the isospace (and hence, the inferred ecology) of ancient species is (or is not) consistent with that of their modern counterparts living in the Bay of Biscay. This large-scale comparison will provide independent evidence to further assess and contextualize the degree and possible impact of sources of variation on the carbon and nitrogen stable isotope composition of the ancient whales, including the issue of time averaging. I did a quick search on Google Scholar and found a large number of studies conducting stable carbon and nitrogen isotope analyses on modern species that can be used for this task. In addition, the examination of the stable isotope composition of modern representatives would likely provide clues to the question of the identity of ancient individuals assigned to the North Atlantic right/bowhead whales' group: Are the relative $\delta^{13}\text{C}$ and $\delta^{15}\text{N}$ values of individuals in this group similar with those of modern right whales inhabiting the area or bowhead whales from the Arctic?

A small note in Figure 1b: Shows that one of the bone specimens was identified by ZooMS as belonging to a seal/sea lion. No sea lions (living or extinct) have ever been identified in the North Atlantic region (Berta et al., 2018; Valenzuela-Toro & Pyenson, 2019). Therefore, it can be confidently stated that this specimen is from a seal.

Something that caught my attention is the way the raw data is presented. It is a very long PDF where the sample IDs and the data are on separate pages, making it extremely difficult to relate them. I literally had to print them out and glue them together to make sense of the numerical results. I don't know if this is due to the formatting constraints of the submission, but if it is published, I urge the authors and the editor to ensure that the raw data is presented as an Excel file or something similar.

References

- Berta, A., Churchill, M., & Boessenecker, R. W. (2018). The origin and evolutionary biology of pinnipeds: seals, sea lions, and walrus. *Annual Review of Earth and Planetary Sciences*, 46, 203-228.
- Guiry, E. J., & Szpak, P. (2020). Quality control for modern bone collagen stable carbon and nitrogen isotope measurements. *Methods in Ecology and Evolution*, 11(9), 1049-1060.
- Guiry, E. J., & Szpak, P. (2021). Improved quality control criteria for stable carbon and nitrogen isotope measurements of ancient bone collagen. *Journal of Archaeological Science*, 132, 105416.
- Pestle, W. J., Crowley, B. E., & Weirauch, M. T. (2014). Quantifying inter-laboratory variability in stable isotope analysis of ancient skeletal remains. *PLoS one*, 9(7), e102844.
- Valenzuela-Toro, A., & Pyenson, N. D. (2019). What do we know about the fossil record of pinnipeds? A historiographical investigation. *Royal Society Open Science*, 6(11), 191394.
- Van Klinken, G. J. (1999). Bone collagen quality indicators for palaeodietary and radiocarbon measurements. *Journal of Archaeological Science*, 26(6), 687-695.

Version 2:

Reviewer comments:

Reviewer #1

(Remarks to the Author)

I would like to begin by thanking the authors for their hard work and for the substantial additions, particularly in the supplementary sections of the manuscript. I have a few additional comments that need to be addressed before I can recommend the paper for publication. Some of these comments are in response to previous discussions, while others are new and arise from the updated text and supplementary material. Overall, I am pleased with how the authors have addressed all other outstanding concerns.

Section-based comments:

Abstract:

- Many of the responses to reviewer comments confirm that they are not suggesting whale exploitation was likely but are in fact discussing utilization. I would argue that exploitation and utilization are not quite the same. Please amend this throughout the abstract and manuscript, especially as using a stranded whale carcass. The exploitation is the "practice of harming or killing animals for human benefit", if you do not want the manuscript to suggest they purposefully killed the whales, this needs to be reflected better throughout the manuscript. Please rephrase appropriately.

Introduction:

- "Whales are the largest living animals on Earth and their current populations are a mere fraction of their abundance in the past (Christensen 2006)." I do not entirely agree with this sentence. Some species and populations are back to over 90% of historic population size and some at carrying capacity (humpback whales and SRW are doing particularly well), fin whales in the North Atlantic are somewhere between 50 – 70% of prewhaling size, and many we don't have historic estimates for. Please clarify "for many species", or similar.

- "Overall, ZooMS analysis of 131 cetacean specimens reveal the presence of at least six cetacean taxa in the northeastern Atlantic during the Magdalenian (Figure 1 and Supplementary Information 4): fin whale, *Balaenoptera physalus* (n = 65); sperm whale, *Physeter macrocephalus* (n = 32); gray whale, *Eschrichtius robustus* (n = 11); blue whale, *Balaenoptera musculus* (n = 2); one species of porpoise (harbor porpoise or Dall's porpoise, *Phocoenidae*, n = 1); and at least one species of Balaenid whale (*Balaenidae*), with 13 samples that can be attributed either to the North Atlantic right whale, *Eubalaena glacialis*, or to the bowhead whale, *Balaena mysticetus* (two species that are indistinguishable using ZooMS)." Following on from one of my previous points, please add "both present in the North Atlantic". SRWs and NPRWs cannot be distinguished either, but they are unlikely to be in the North Atlantic. This should be stated somewhere.

Results:

- It currently seems a bit like you are overselling your change in date of earliest "exploitation" given your error of 1500 yrs. The error suggests it might only be 500 years difference to previous findings, no? Or am I missing something? And as there were only two early samples that caused this shift, I would encourage more modest wording around this finding. Reword results etc. appropriately to capture this.

- Check font sizes of delta notations

- I would also argue you have potentially higher d15N in the NARW, as the mean of those baleens would sit in-between the min and max d15N values, thereby lower than your historic samples (if they are indeed NARWs), see comments in discussion below about why I don't fully agree with your statement suggesting they might be bowhead whales. I would hypothesize from your findings that the d15N baseline of offshore pelagic waters was much lower in the past (perhaps from high productivity) relative to coastal waters. If this was a northerly feeding ground. The fin whale and blue whale thereby would have lower d15N than the coastal NARW and gray whale – the trophic level conundrum than is no more. Of course this is just a theory, but probably quite plausible under the environmental circumstance of this time period and the bathymetry of your study region. Perhaps this could be captured somewhere?

Discussion:

- Please change exploitation to utilization here: "The exploitation of fin whale carcasses at Santa Catalina after 16 ka cal BP, and the presence of a foreshaft made of gray whale bone at Andernach with contextual dates younger than 16 ka cal BP, both show that hunter-gatherer groups still had access to whale bone at that period."

- I don't agree with the following statement: "It also suggests a higher likelihood that the *Balaenidae* samples correspond to bowhead whales rather than right whales (Figure S7.4) although this is less certain." Right whales have very broad foraging patterns globally which often overlap with bowhead whales. The more "Arctic conditions during your time period might suggest they were more likely to be Bowheads OR the Bay of Biscay may have represented a northerly feeding ground for NARWs. Please rephrase to capture this.

- "The worked objects and unworked bone fragments we analyzed are predominantly large cetaceans that mainly use offshore habitats, namely fin whale, sperm whale and blue whale". Although I partially agree with this statement it is not always the case and this should be captured here. In areas where you have deep water close to shore, like the Bay of Biscay or the St. Lawrence Estuary (there are many more examples), more "offshore" whales can be seen in large aggregations from the shore. It is not the shore that is preventing them from being close but the depth and their foraging preferences. Please rephrase to capture this, I suggest something like "The worked objects and unworked bone fragments we analyzed are predominantly large cetaceans that foraging predominantly offshore or in areas with deep water close to shore like the Bay of Biscay, namely fin whale, sperm whale and blue whale".

- I like the term "opportunistic acquisition" you have used in this section. Perhaps this could replace exploitation where appropriate?

- "Whales were likely familiar to coastal communities, and as such may well have played an important role in the Magdalenian cultural world. The Bay of Biscay's highly productive waters make it a cetacean hotspot today (Laran et al.

2017), and it was likely already the case during the Pleistocene.” I think I’ve read this sentence somewhere else in the manuscript/supp material, just check wording isn’t identical.

- “Whales were likely familiar to coastal communities, and as such may well have played an important role in the Magdalenian cultural world. The Bay of Biscay’s highly productive waters make it a cetacean hotspot today (Laran et al. 2017), and it was likely already the case during the Pleistocene. Along the Spanish coast in particular, water depth increases dramatically just a few kilometers from the coastline, bringing species that are typically found offshore, such as fin whales (McLeod et al. 2009), unusually close. These species could have been observed from high points along the coastline spouting at a distance, and irregularly closer to shore as dead or moribund individuals. The more coastal right/bowhead whales, gray whales and harbor porpoise were likely even better known. Gray whales in particular are the most coastal of the whale species (Wilson & Mittermeier 2014), spending summer at low-depth high-latitude feeding grounds, then migrate hugging the coastline to warmer lower-latitude calving areas in sheltered low-depth lagoons. Whereas in today’s climate the Bay of Biscay and the Mediterranean would more likely correspond to calving grounds, it is possible that during the cooler Upper Paleolithic climate they would have corresponded to feeding grounds (Speller et al. 2016, Rodrigues et al. 2018)—habitat reconstructions indicate that adequate shallow shelf habitat was available there during the last glaciation (Alter et al. 2015). In any case, they would have been conspicuously present at predictable seasons, and they are likely to have been part of the range of animals whose ecology would have been well known to Paleolithic hunter-gatherers.” I really like this paragraph. Perhaps it should go higher up with the paragraph that starts “The taxonomic composition of the whale-bone assemblage gives...”. The paragraph before also starts with “Finally...” and it is no longer finally (although it might be if you move this new paragraph above”.

Appendix 6:

- I thank the authors for addressing my previous concerns around species-specific DeltaR. I am not overly happy with the justification of the mixed-species deltaR that is used here, more specifically for the sperm whale and gray whale. Please include some additional words in Appendix 6 for these two species, and show the differences in the results for these two species based on deep water feeding (sperm whale, i.e. -400-800 yrs difference rather than ~150) and shallow water feeding (gray whale, i.e. -50 yrs rather than ~150). I understand that given your 1500 yr error, the results do not change, but I think it is important to acknowledge how the deltaR you have chosen isn’t overly optimal for these two species based on their known feeding habits. Also the species that are available on calib.org are mostly pelagic whales (minke, sei, humpback, fin), so I would expect the mean deltaR from these to be more suitable to the fin whale but not the gray whale or sperm whale. Please consider this explicitly somewhere.

- “Choosing the most suitable reservoir correction required careful consideration of various parameters. Foote et al. (2013) used a DeltaR = 0 to correct for the reservoir effect in Pleistocene whales. However, this no longer seems suitable since the update of the Marine20 calibration curve, as there are large fluctuations in pre-Holocene reservoir ages (Heaton et al. 2020).” Please change to Foote et al. 2013 use a DeltaR to correct for the reservoir effect in bowhead whales during the Pleistocene. It is important to note that this previous paper only corrected one species.

Appendix 7:

1. Methods for fin whale suess corrections?

2. Please change “Fin whales in our sample generally display high $\delta^{15}\text{N}$ ratios when compared with their modern counterparts (Fig. S7.2), indicating that they occupied on average a higher trophic level than today’s whales (Peterson and Fry 1987).” To “Fin whales in our sample generally display high $\delta^{15}\text{N}$ ratios when compared with their modern counterparts (Fig. S7.2), indicating that they occupied on average a higher trophic level than today’s whales (Peterson and Fry 1987), or, the $\delta^{15}\text{N}$ baseline has significantly changed over time”. I would argue that as these differences are observed across multiple species, changes in the $\delta^{15}\text{N}$ baseline is more likely given the environmental change rather than occupying a higher trophic level in the past.

Response to authors direct responses:

- Fin whales often forage at a slightly higher trophic level relative to Balaenidae, but here they have lower nitrogen isotopes, why? Is it because the fin whales were feeding at higher latitudes where baseline isotopes differ (lower nitrogen likely?). We were also puzzled by this. Following advice from the reviewers (see below) we now have included as supplementary information a comparison between the isotopic signatures of our Paleolithic samples with those of modern counterparts (Supplementary Information 7). Comparing modern fin whales (Figure S7.4) and modern bowhead and North Atlantic right whales (Figure S7.6), we found that fin whales also have lower $\delta^{15}\text{N}$ ratios than bowhead whales and a similar range to right whales. In other words, this pattern is also evident among modern whales (and indeed it is even stronger than among ancient samples). This is very interesting, given what is known about the diets of these two species, I would suggest this is more likely to be due to predominant feeding in different feeding areas at high latitudes. At least for modern fin whales, many migrate to Iceland, whilst NARWs have much more coastal migratory patterns along the coast of North America. Although the eastern NARW population is extinct, I would have expected a coastal migratory pattern too relative to the more offshore deep feeding fin whale. The depth these two organisms feed at should also be considered. Please add somewhere a statement on their differing migrations (coastal vs. open ocean) and summer feeding grounds as a likely cause of the differing $\delta^{15}\text{N}$ (due to differences in baselines rather than trophic level).

- Lines 364-365 = 437-439: Fin whales also inhabit the Mediterranean and all species forage close to the islands of the Azores. I would describe them as migratory habitat specialists rather than just offshore whales. Careful. We reworded the sentence to refer to these species as “large cetaceans that mainly use offshore habitats”. We are unsure why referring to them as “migratory habitat specialists” would have been clearer in the context of the point we are making here, which is that it is extremely unlikely that these species would have been whaled at the time. I would clarify by adding “or found foraging in deep waters close to shore” (e.g. the Bay of Biscay).

Reviewer #2

(Remarks to the Author)

Review of McGrath et al.

The authors have revised their manuscript, with an improvement in what information is retrieved from the stable isotope analyses per species as recommended by Reviewer 1.

Most importantly, they clarify a misunderstanding caused by misleading text, that the authors are not claiming that this represents any indication of active hunting of whales, and that this is supposed to represent a focus more on the significance of tool making from available materials.

However, remnants of the earlier text that causes such confusion still remains. For example, in the abstract itself: “Reconstructing the prehistoric exploitation of large cetaceans is challenging, but key to understand the history of early human coastal adaptations” (as collecting bones from coastal environments and working them does not have the same implications as adapting to life on the coast)

and

“reveals at least five species of large whales, significantly expanding the range of known taxa exploited in this period” (again, ‘exploitation’ here could be considered misleading)

These sentences could very easily be interpreted as hunting, whereas a few additional words relating this to their carcasses would clarify this, or, if keeping the potential for active hunting in their narrative, then they could add a separate short sentence clarifying that this is not the only/most likely scenario. It is important as there is a great difference in significance, which the authors should detail in the Discussion.

The newly added sentence:

“They were thus a key part of subsistence for many coastal human groups worldwide, including hunter-gatherers and Neolithic farmers, with exploitation methods that included scavenging freshly beached animals, opportunistic killing and organized whaling” retains the notion that

Amendments to the isotope interpretations, as helped by Reviewer 1 suggestions, gives much greater significance to the palaeoecology of the different species, but does not add to the interpretation of bone tool making.

The authors point out that “DNA analysis would be required to verify the species identities for these specimens”, in reference to the balaenid remains, but then why not attempt DNA analysis? (particularly given the expertise of some of the authors). Is this due to cost, expected failure rate, or sample size?

For example, as was done with Hufthammer et al. 2018 for using ZooMS supported by aDNA to show presence of grey whale in the North Atlantic waters around Norway – a reference that would support the statement:

“Previous evidence of their past presence in the North Atlantic dates either to the Holocene (10 to 0.25 ka cal BP) or to the middle of the Marine Isotope Stage 3 (MIS3), before 40 ka cal BP;...”

Given that the authors claim the porpoise was too small to radiocarbon date it would be useful to clarify this point in the text also (preferably with specimen sizes/masses if available, though not essential if not), i.e., is this more about actual sample size, or sampling damage, etc.

Now that I know the main scope is on the bone tool manufacture, rather than the implied cognitive abilities of active whale hunters, there are a few other questions more specifically about this manuscript:

What is the significance of the worked bone tools being made of remains from these particular species?

How does tool manufacture differ from other known species from the same region of the same time (and earlier)? See numerous articles referred to by Borao 2022 in JAS reports. This aspect of working bones needs to be much more detailed, or direct this as a manuscript talking about palaeoecology (still, this disjunct between the two aims leaves it somewhat unclear what the main point is – the isotope ecology of particular cetacean species at a particular time in the past OR the fact that the contemporary people were creating tools from such remains)

I suspect that it is more the latter now, based on the response to reviewers comments and amended version.

It is good to see some initial speculation on skeletal element use, though which the authors acknowledge cannot be inferred due to the workings, though I think the following paragraph on the second exploitation behaviour, of the unworked bone, could also link in here. This paragraph would in particular benefit some information with regards the species – as I do not see any taxon reference more specific than ‘whale’ in this whole paragraph, but if discussing aspects of transport, surely relating this to species (and their sizes) would be of importance.

Overall this does offer some interesting data on bone tools, through isotopes of specimens confirmed through molecular fingerprinting, supported through radiocarbon dating, but still would benefit from improvement in context with regards bone tool manufacture in relation to the current knowledge on bone industry more broadly in the Discussion.

Reviewer #4

(Remarks to the Author)

Dear authors

I am pleased to see the authors carefully considering the reviewers' comments. The revised manuscript is more thorough, and the methodologies are more appropriately explained, providing better support for the results. I especially appreciate the addition of supplemental files 6, 7, and 8.

Regarding the paleoecological component of this work, the authors have done a satisfactory job by compiling stable isotope data on modern whales to contextualize their data on fossils. As suggested, they conducted corrections for tissue type and the Suess effect to make this modern data comparable in a contemporary context. However, when performing the next step (i.e., comparing modern and fossil data), the authors overlooked the long-term temporal shifts in the carbon and nitrogen isotope compositions at the base of the food since the Pleistocene. The baseline changes can be significant, potentially compounding interpretations of patterns of moderns vs. fossils. I understand the focus of this paper is not reconstructing ancient marine isotopic baselines; however, some reflection on this issue must be considered in the text, wildly when pinpointing changes in the foraging ecology because of climate change on page 9.

I also noticed that the authors conducted interpretations based on the individual variation of the $\delta^{13}\text{C}$ OR $\delta^{15}\text{N}$ values. Technically, nothing is wrong with this, but it is important to note that this approach can lead to an incomplete view of the underpinning factors causing their variation. The $\delta^{13}\text{C}$ and $\delta^{15}\text{N}$ values of an individual are not independent, as they can be influenced by the same external (e.g., primary productivity, trophic level) and internal (e.g., metabolic routing) factors. Therefore, in addition to basing interpretations on the variation of each of these systems separately, it is also recommended that a more comprehensive interpretation be conducted that considers both. For example, as an organism's trophic level increases, not only its $\delta^{15}\text{N}$ values but also its $\delta^{13}\text{C}$ values in a predictable fashion. I encourage the authors to assess the covariation of the $\delta^{13}\text{C}$ AND $\delta^{15}\text{N}$ values and examine whether their inferences are consistent with the patterns of variation expected for both the $\delta^{13}\text{C}$ and $\delta^{15}\text{N}$ values. For instance, do the $\delta^{13}\text{C}$ values support conclusions about trophic ecology derived from $\delta^{15}\text{N}$ values? Or are deductions about benthic diving based on the $\delta^{13}\text{C}$ values of some species consistent with their corresponding $\delta^{15}\text{N}$ values?

Some other minor comments are included in the annotated manuscript and supplemental file 7. I hope these comments can be easily addressed. I appreciate the quality and importance of this study and look forward to seeing it published.

Version 3:

Reviewer comments:

Reviewer #1

(Remarks to the Author)

Thank you for addressing all of my previous comments. This manuscript was a pleasure to read, and it is evident that significant effort has greatly enhanced its quality—it's truly excellent. I have just one suggestion before I am happy to recommend its acceptance for publication. According to the policies and author guidelines of Nature Communications, all authors must make materials, data, code, and related protocols readily available to readers. I suggest uploading the raw stable isotope data as a supplementary table to facilitate its use by others. Apologies if this is already included and I simply overlooked it in the supplementary files.

Reviewer #2

(Remarks to the Author)

The manuscript now reads in a more coherent matter that yields field-specific interests, such as the application of isotope analyses and ZooMS to improving how we can interpret an archaeological assemblage – particularly with regards the isotope information.

I do still have some concerns on the overall claim of this study that tries to emphasize ancient human-seashore interactions – i.e., I understand that they may represent the oldest dated bones from whale remains but it is not clear how the results are significantly revealing about whale ecology and ancient human-seashore interactions more than already known or expected

The statement “significantly expanding the range of known taxa acquired by humans” is somewhat overstressing the facts, particularly given that later in the manuscript you state:

“Fin whales, sperm whales, blue whales and harbor porpoises are still present in the Bay of Biscay today” – so, it is unclear whether the authors are emphasizing that adding two species to this list a significant expansion of our knowledge of the ecology, or, that humans in the past were selecting for the remains of specific species to utilize? I.e., why the contemporary humans would not simply make best use of any cetacean remains that stranded upon their shores.

What the significance of this oldest whale bone use in relation to the use of terrestrial animal bones for crafting tools from remains ambiguous. I.e., it is not clear if there a particular reason why coastal populations would avoid using such remains.

Use of the phrase ‘whale utilization’ remains misleading because in this instance it equally implies active hunting of live animals – should be ‘whale carcass utilisation’ where appropriate, unless, as mentioned for last comment, there is some

reason why such human groups would not instinctively utilize such an obvious resource.

'range of whale taxa acquired' is also misleading – although true in strictest sense, it implies targeting of particular species, some level of selection, but if these are strandings then no such selection process occurred – indeed, aspects of selection are discussed in the Discussion section of the manuscript, though this should be improved with current information on strandings of these species.

With respect to “Our results show that at least one balaenid species was found in the Bay of Biscay at the time, although the imprecision of the ZooMS ID does not allow for a distinction between right and bowhead whales; DNA analysis would be required to verify the species identities for these specimens (analyses which fell outside the scope of this study)” If understanding the palaeoecology contemporary with these oldest worked cetacean bones is one of the key features of the manuscript, it is unclear why the DNA work is not done on these samples. More surprising given that you have cetacean ancient DNA specialists in the author list, and there is a column in the supplementary files that states several of the samples having subsamples sent for DNA analysis, but no clear mention of any such attempt in the manuscript, or a methods section on how this was done. It would be even more interesting given that based on the text later in the same paragraph, the species-level resolution would yield seasonal interpretation. Therefore better justification of avoiding this, and clarity on what was done, would help the reader better understand the work.

The comment “It is puzzling that bowhead/right whales (which feed mostly on copepods) appear as having lower $\delta^{13}\text{C}$ values than fin whales, but this is also the case in modern whales” – though could this be rephrased to highlight the limitations in our understanding of isotopic signatures?

As a minor point, an explanation of why “one sample where ZooMS failed to distinguish between right/bowhead and fin whales” would be important here. I.e., was this based on a poor-quality spectrum? If so, was this due to poor protein preservation of the sample?

Overall, the authors present some useful information relating to isotope analyses on a small number of cetacean species from this interesting area that will be of interest to some archaeologists and palaeoecologists.

Reviewer #4

(Remarks to the Author)

Dear authors

I have read the new version of this manuscript. I am pleased you have thoroughly addressed most of the reviewers' feedback, including mine. I have no further comments or suggestions, and I look forward to seeing it published.

Manuscript NCOMMS-23-62988, “Oldest whale bone tools reveal past whale ecology and ancient human-seashore interactions”: detailed response to reviewers

Black lettering = reviewers’ comments

Blue lettering = authors’ responses and comments.

All line numbers below are given as “xxx = yyy”, where “xxx” corresponds to line numbers in the initial submission, and “yyy” corresponds to line numbers in the revised manuscript.

Reviewer #1 (Remarks to the Author):

This study taxonomically identifies and radiocarbon dates whale artefacts and ecofacts from sites near to the Bay of Biscay. The marine reservoir correction that the authors implemented date the samples somewhere between 16 - 20 kya. The radiocarbon results are rather striking, providing evidence for some of the earliest known utilisation of whale faunal remains in Europe at this scale. However, there are some important implications of using a blanket reservoir correction for different whale species that need to be considered and the associated caveats discussed before this manuscript should be accepted for publication. Whilst additional methodological information on the stable isotope analysis and improved interpretation of the stable isotope results is also necessary. Despite the major revisions required, I believe the interesting findings from this study deserve consideration for publication, as they contribute to an ever-expanding pool of evidence of early interactions between humans and marine taxa, with implications on our understanding of how humans and coastal habitats have co-existed for millennia far beyond what many researchers have imagined to date.

Radiocarbon dating considerations:

- Differences in the foraging strategies of the different whale species and the implications of this on marine reservoir corrections MUST be considered and discussed, especially as this could bring dates more in-line with previous evidence from this region.
 - o Sperm whales forage in deep waters and existing deltaR values can vary between -168 and 504 +/- 60 years. See Beck et al. 2022. <https://doi.org/10.1017/RDC.2022.79>
 - o Bowhead whales forage in polar/sub-polar waters close to the sea surface, the deltaR for Bowheads has been estimated at 24 +/-58 years. See Pienkowski et al. 2022. <https://doi.org/10.1111/bor.12606> Similar foraging behaviour is observed for Eubalaena.
 - o Fin whales forage in water depths of 100 – 400ms and migrate vast distances. They are known to forage predominantly in polar and sub-polar waters, but some individuals are known to also forage at lower latitudes, see Buss et al. 2022. <https://doi.org/10.1007/s00227-022-04131-x>
 - o Grey whales forage in shallow sedimentary environments and migrate vast distances. Please consider the implications of this on the reservoir corrections and the Portlandia effect. See Dyke et al. 2013 <https://doi.org/10.1111/j.1502-3885.2012.00256.x>

We acknowledge the concerns raised by the reviewer regarding the reservoir correction of radiocarbon ages of the different whale species, and we agree that these issues need to be addressed in the paper to better clarify our choice of DeltaR. We thank the reviewer for the suggested references.

We have now created a new section in Supplementary Information 6 that discusses in detail our choice of DeltaR. It reads:

“Dating of marine animals needs to take into account the marine reservoir effect: given that oceans are depleted in ^{14}C in relation to the atmosphere, conventional radiocarbon dates for marine organisms appear older than for coeval terrestrial individuals (Heaton et al. 2020). Some studies use sample pairing between coeval terrestrial and marine taxa to aid interpretations of reservoir ages at archeological sites (e.g., Dury et al. 2022). This was not possible in our case, because our samples are all from Pleistocene cave and rockshelter sites with complex stratigraphy (in which each excavated “layer” is the result of an accumulation of multiple occupation episodes) and most of our samples are from ancient excavations with poor stratigraphic resolution. We thus relied on the correction for the reservoir effect implemented for marine samples in OxCal 4.4, which is based on the Marine20 calibration curve (Bronk Ramsey 2009).

In addition, dating of marine organisms can be further refined by taking into account the fact that the marine reservoir effect varies across oceanic masses, being for example stronger (i.e. leading to seemingly older organisms) at higher latitudes and in deeper waters (Heaton et al. 2023). This deviation from the standard (globally averaged) marine reservoir effect is expressed as a DeltaR term. In the case of our whale specimens, however, it is not the coastal waters adjacent to the archeological sites that matter, but the water mass (or masses) where whales fed. The whale species identified in this dataset are very diverse in terms of their feeding ground locations and feeding depths. For example, previous studies estimated DeltaR values: between -168 and 504 \pm 60 years for sperm whales (that feed in deep waters but can be found in either high latitudes or sub-tropical seas; (Beck et al. 2022); at 24 \pm 58 years for bowhead whales (that forage in polar/sub-polar waters close to the sea surface; Pienkowski et al. 2022); and about 350 years for a fin whale (a species that forages at depths up to 400 meters and migrates vast distances; Birkenmajer and Olsson 1998). Intra-specific variability in DeltaR values renders it difficult to obtain meaningful values per species. We have not found published estimates of DeltaR for gray whales, but the foraging strategy of this species (which feeds on invertebrates found in shallow sedimentary environments and migrates over very long distances) renders it particularly susceptible to intraspecific variation in DeltaR, especially as it will be dependent on the type of sediment (molluscs feeding on calcareous sediments appear over 2000 years older than suspension-feeding ones (England et al. 2013).

In addition to these difficulties, DeltaR values published in the literature from modern (Holocene) whales need to be approached with caution when applied to Pleistocene samples. Indeed, the spatial variation in the reservoir effect of water masses likely varied over time, in response to changes in ocean circulation. The foraging grounds of whales are also likely to have been different, given that in the Late Pleistocene the study region was substantially cooler. For these reasons, we have opted for not attempting to apply a species-specific DeltaR based on information from modern whales, and instead use a single DeltaR as suitable as possible to the period and region of our samples.

Choosing the most suitable reservoir correction required careful consideration of various parameters. Foote et al. (2013) used a DeltaR = 0 to correct for the reservoir effect in Pleistocene whales. However, this no longer seems suitable since the update of the Marine20 calibration curve, as there are large fluctuations in pre-Holocene reservoir ages (Heaton et al. 2020). Monge Soares et al. (2016) published a DeltaR = -117 ± 70 yr for the Late Pleistocene based on three samples pairs (shells and bones/charred wood) from the Cantabrian coast. Mangerud et al. (2006) showed that there is little difference between the DeltaR from whales and mollusks found at the same location. However, since the update of the Marine20 calibration curve, the DeltaR from Monge Soares et al. needs updating as well. Monge Soares et al. used three paired ^{14}C ages for the calculation of their Late Pleistocene DeltaR (terrestrial ^{14}C ages: $15\,656 \pm 75$, $19\,710 \pm 120$ and $23\,040 \pm 50$ yr BP). Considering the ages of our

samples, we decided it would be most suitable to use the two youngest paired ^{14}C ages for an updated DeltaR, because the third sample pair was considerably older. These two paired Late Pleistocene ^{14}C dates from Monge Soares et al. (2016) were used in the online application (Reimer & Reimer 2017) to calculate updated DeltaR values. These new DeltaR values are statistically the same (0.037 ($\chi^2:0.05 = 3.84$)). The weighted mean of these DeltaR and standard deviation are calculated according to: <http://calib.org/marine/AverageDeltaR.html>, which gives a new Delta R of -448 ± 39 yr. The calibration was done in OxCal 4.4 (Bronk Ramsey 2009).

Additionally, using all North Atlantic whales (various species) in the Marine Reservoir Correction Database we get a DeltaR of -168 ± 53 , compared with the DeltaR of -117 ± 70 from Soares for the Holocene. This means that Soares' data actually slightly overestimate the reservoir age, making samples slightly younger. This may not be the case for the Late Pleistocene samples but it does strengthen our idea that the material used for paired dating in this region by Soares is not that far off from whales themselves.

Finally, uncertainty regarding the adequate DeltaR for our samples should not affect our overall results, which are presented in broad "time slices" of 1.5 millennium (Figure 2). It does however mean that caution is needed when interpreting individual radiocarbon dating results. These can be improved in the future as more information becomes available regarding suitable corrections for the marine reservoir effect."

o Harbor porpoises forage locally in rocky coastal environments, although they seasonally also forage further offshore, I would estimate that this species is least likely to be affected by the deltaR choice used here.

Note that there was only one sample identified as porpoise, and the sample size was too small for radiocarbon dating: this specimen was not dated.

- Many studies use sample pairing between terrestrial and marine taxa to aid interpretations of reservoir ages at archaeological sites. See Dury et al. 2021. <https://doi.org/10.1177/09596836211041> - it is not clear to me why other chronological evidence (e.g. terrestrial fauna or shells from the same sites) were not used for comparison. Please explain why only whales were used, it seems an odd choice given all the challenges with using marine mammals for marine reservoir corrections and dating, especially for migratory species that feed in multiple bodies of waters at varied depths.

This is a good point, and we now discuss explicitly in Supplementary Information 6 why we were unable to use paired samples of terrestrial and marine taxa:

"Some studies use sample pairing between coeval terrestrial and marine taxa to aid interpretations of reservoir ages at archeological sites (e.g., Dury et al. 2022). This was not possible in our case, because our samples are all from Pleistocene cave and rockshelter sites with complex stratigraphy (in which each excavated "layer" is the result of an accumulation of multiple occupation episodes) and most of our samples are from ancient excavations with poor stratigraphic resolution. We thus relied on the correction for the reservoir effect implemented for marine samples in OxCal 4.4, which is based on the Marine20 calibration curve (Bronk Ramsey 2009)."

- Known deltaR corrections for marine taxa foraging in polar waters relative to lower latitudes can vary by up to approximately 800 – 1200 years due to the upwelling of older water in these regions. This will impact the surface waters less than the deeper waters (e.g. a fin whale

foraging in polar waters will look older than a bowhead whale). Given these considerations, the results of your findings may differ by over a millennia for a fin whale foraging at higher latitudes. However, we cannot guarantee that fin whales always forage in the same body of water given their migratory foraging behaviours and differences between individuals of the same population (e.g. evidence of foraging at lower latitudes (Buss et al. 2022) and during winter (Silva et al. 2019). Although this may impact how definitive the dates you have provided here are, I do not think these considerations will impact the overall result from the manuscript, but the uncertainty around the MRE given the taxonomic groups you have studied MUST be incorporated in the text.

As mentioned above, we now discuss extensively in Supplementary Information 6 how DeltaR values can vary across species, warning explicitly the reader not to take them as definitive. In agreement with the reviewer, we do not think these uncertainties affect the overall results of our manuscript.

Stable isotope considerations:

- The reason for including the SIA results is currently not clear to the reader. Are all these samples from the same time period? Or a few millennia apart. Where these samples dated?

The stable isotope data originates from the same material that was subjected to radiocarbon dating, which means that the data spreads the same chronological time span as the radiocarbon samples (see reply below for details). Accordingly, these results give us insights regarding the foraging behavior of the whales and paleoenvironmental context in the late Pleistocene.

We have now clarified this in the manuscript (lines 183-188): “Thirty-seven of the worked objects and 31 of the unworked bone fragments were also sampled for radiocarbon dating using the compact AMS ECHOMICADAS, and for carbon and nitrogen stable isotope analysis ($\delta^{13}\text{C}$ and $\delta^{15}\text{N}$). This work enabled us to assess the range of whale taxa exploited by the Magdalenian foragers and the chronology of exploitation. The results of the stable isotope analysis, placed in the context of patterns of isotopic niche partitioning among contemporary whales, shed light on the whales’ relative foraging behavior and past ocean ecology.”

- The carbon isotope ratios suggest the balaenidae, fin whales and gray whales were likely feeding in different bodies of water (if from the same time period), please see comments on radiocarbon considerations above. The relationship between the radiocarbon dates and the SIA carbon data should be considered. The Balaenidae carbon values may potentially indicate younger samples (if these have been dated), is that something you observe? I would try and link the interpretation from the SIA data and the radiocarbon dates more in the discussion.

This is a very interesting point from the reviewer, so we looked into this and found that the ages of the different whale groups broadly overlap:

The balaenidae (right/bowhead whales) are dated from 16-16.5ka to 17.1-17.8ka cal BP

The fin whales are dated from 13.2-13.5ka to 19.5-20ka cal BP

The gray whales are dated from 16.3-16.9ka to 17.4-18ka cal BP.

We plotted the conventional ^{14}C age against the $\delta^{13}\text{C}$ ratio, finding no correlation. We have added this additional result to Supplementary Information 7 and the following text to the results section of the main text: “We found no evidence that these patterns of stable isotopic signatures across taxa are driven by differences in the radiocarbon age of samples: there was

no significant relationship between conventional ^{14}C age and the $\delta^{13}\text{C}$ (Fig. S7.3 in Supplementary Information 7).”

- Please be careful with wording, relative foraging preferences can be inferred from stable isotopes but exact diet cannot (e.g. the prey chosen) be inferred without other known ecological data/information. Please amend throughout the text.

Agreed, and we have now reviewed the wording accordingly.

- Fin whales often forage at a slightly higher trophic level relative to Balaenidae, but here they have lower nitrogen isotopes, why? Is it because the fin whales were feeding at higher latitudes where baseline isotopes differ (lower nitrogen likely?).

We were also puzzled by this. Following advice from the reviewers (see below) we now have included as supplementary information a comparison between the isotopic signatures of our Paleolithic samples with those of modern counterparts (Supplementary Information 7). Comparing modern fin whales (Figure S7.4) and modern bowhead and North Atlantic right whales (Figure S7.6), we found that fin whales also have lower $\delta^{15}\text{N}$ ratios than bowhead whales and a similar range to right whales. In other words, this pattern is also evident among modern whales (and indeed it is even stronger than among ancient samples).

- Please include additional information on SIA methodology (see comments in attached manuscript).

We have now developed the methods section to provide more information on the stable isotope methodology.

Introduction considerations:

You state that understanding the why and how of this Magdalenian whale exploitation is thus of crucial importance. However, I do not feel your approach here sheds light on the why and the how, but does provide important evidence on when and which taxa may have been utilized by humans in this region. I believe the introduction could use some tightening up around the when and which taxa, and why this is an extremely unique and important finding as the main

aim of this research. Rather than precluding towards the why and how, without providing much evidence for either of these questions.

We removed the reference to “the why and how”. The importance of precisely dating and identifying the whale species (the “when and which taxa”) is stressed in the following paragraph.

Reviewer #1 (Remarks in the attached file, by the order of appearance in the original manuscript):

I find the title slightly confusing. What is a whale-bone industry? And is it the ecology of the ocean in the past or the ecology of the whale? Or the humans? Alternative title suggestion: Ancient human-seashore interactions reveal oldest known utilisation of whales

“Industry” is a common term in archeology to refer to a set of techniques, tools and activities used to transform a specific raw material, and the resulting material evidence (objects, waste products, etc.). This word is more specific than “utilization”, and it was chosen here because there are earlier cases of the opportunistic utilization of stranded whales (e.g. in the African Earlier Stone Age), but only as a source of food and not to make tools. But since this term seems unclear, we changed it, and also replaced “past ocean ecology” with “past whale ecology”.

Lines 114-115 = 121-122, “their current populations are a mere fraction of their abundance in the distant past”: Do you mean the recent past? Populations were rather large for many taxa up until the 20th century...

Changed for “in the past”. We did not mean “recent past” as for some species exploitation and depletion started centuries ago (including: bowhead, right and gray whales).

Lines 115-116 = 123-124, “whales represented an enormous potential source of food and other resources”: Well, not all of them, as many inhabit offshore environments in the high seas so would have been challenging to target... you mention this later on. Please capture this here.

We restructured this passage and toned down the formulation as “enormous potential source”.

We maintain however that even whales that inhabit offshore environments were an important source of resources, as they can be found stranded on the shore. And indeed, fin whales and blue whales are among the species we identified as having been used to produce worked objects (Figure 1a), some of which valuable enough to be transported inland for hundreds of kilometers.

Note that at this early point in the text we are discussing the utilization of whale resources in general, not the specific possibility of whale hunting.

Lines 116-117 = 122-127, “Long before industrial whaling depleted the populations of most species”: This doesn’t currently link with the previous sentences, when and what was industrial whaling?

The word “industrial whaling” was indeed ambiguous as it can apply to different activities and periods. For example: the fin/blue whaling industry only started with the development of steam power at the end of the 19th century; sperm whaling was a global-scale industry in the

18th and 19th centuries; Basques had established a right whaling industry in medieval times; gray whales may have been industrially whaled in Europe shores even earlier.

As these are early introductory sentences in the paper (simply to make the point that whales were once abundant and important resources), we do not think it makes sense to go into this level of detail. We thus restructured this passage, replacing “industrial whaling” with “intensive whaling”.

Line 161-163 [removed] “Dietary data and feeding ecology of these whales is lacking as well”: Ref? Are you suggesting dietary data is missing on all North Atlantic whale species? There is lots of information on whale diet for some species. I would rephrase this sentence to capture that diet is unknown for some species but not others and provide citations.

We meant: for this particular geographic and chronological context. We opted to cut this sentence to avoid ambiguity.

Line 173 = 183 “A subset of these worked objects and unworked bone fragments were also sampled...”: State sample size. The dated objects should come first. How many were worked? How many were unworked? And from which sites. The WHEN this happened seems to be the most important result from this paper. I would encourage you to tailor the manuscript around this.

We added sample sizes, distinguishing between worked objects and unworked bone fragments. The sites list, and number of objects per site, have been added in the Results/Radiocarbon section, although this was present in the Supplementary information and shown on fig. 3.

Line 187 = 199: Complete parentheses “Of the 173 bone specimens (83 worked objects and 90 bone fragments)” + How can you be sure the undated samples were Magdalenian?

We completed the parentheses. The undated objects can be considered Magdalenian because of their typology, their stratigraphic context, and the characteristics of the archeological assemblages they were found in. It was briefly mentioned in the introduction but we added a sentence to make this more explicit (lines 169-171).

Lines 189-190 = 201, “Of the 83 worked objects visually attributed to whale bone”: Delete “visually attributed to whale bone”, repeat from intro/methods

This has been removed.

Lines 193-195 = 203: I find these lines hard to follow, which samples are these now? How is this more precise?

We removed this sentence since R2 also found it ambiguous.

Lines 195-196 = 207, “Of the 90 unworked bone fragments from Santa Catalina”: Either remove the location here, or include the locations of the worked bones above.

We removed ‘from Santa Catalina’.

Line 201 = 213: “Overall, the 130 ZooMS results reveal the presence of at least six cetacean species” replace with “Overall, the 130 cetacean specimens reveal the presence of at least six species”

We have updated to “Overall, the ZooMS analysis of the 131 cetacean specimens reveal...”

Lines 211-212 = 223-224: This confuses me. The gray whale and North Atlantic right whale although extirpated were previously known to inhabit this region (see Frasier et al./Alter etc). The fin whale, blue whale, gray whale and Balaenidae have been identified from this region using more recent archaeological specimens previously (see van den Hurk 2023). I don't think they have been identified for the first time BUT for the first time at this point in time. The Bay of Biscay is a cetacean hotspot due its deep coastal waters and high productivity. Please be careful not to overstate your findings. Moreover, it would be nice to see a short section in the discussion on the differences in species assemblages (e.g. only one balaenidae vs dominated by balaenidae) between the two time periods, and why this may differ so much between time periods.

This comment made us realize that our previous wording was indeed ambiguous. We are reporting here on new results showing that these species were accessible as sources of resource to Magdalenian coastal hunter-gatherers, not on novel results showing these species were already present in the North Atlantic at this point in time. As reviewer 1 notes, the latter would have been trivial, as all of these species are native to the North Atlantic, and thus expected to have been present in this Ocean basin much earlier than the Magdalenian period. For example, and as we mention later in the manuscript, there are bone records of gray whales in the North Atlantic going back to the Pleistocene (Alter et al. 2015). We have revised the text accordingly.

We have now expanded the discussion to compare more explicitly the cetacean communities found today in the Bay of Biscay with those that were found in the Magdalenian period (lines 354-379).

Lines 213-214 = 354-355 [moved to the Discussion], “the range of cetaceans that were accessible to hunter-gatherers”: May have been...

Done.

Lines 214-217 = 354-379 [moved to the Discussion]: I completely agree, the Bay of Biscay is still a cetacean hotspot today, I'm not sure this sentence should come here. I would prefer this near the start of the introduction.

We agree this sentence was out of place in the results. This point is now made in the discussion (lines 354-379) where we contrast the cetacean communities found today in the Bay of Biscay with those that were found in the Magdalenian period, also noting that the Bay of Biscay remains an area of high diversity and abundance of cetaceans today.

Line 219 = 271, replace the title with “Interspecific variation in foraging inferred from stable isotope analysis”

We simplified the title to “Stable isotope analysis”.

Lines 221-222 = 275-276, “49 of which yielded results”: Why? See comments on QC checks and missing information in methods.

In total, 8 samples failed collagen extraction, 2 samples failed IRMS analysis and for 7 samples the atomic CN ratio was too high or the N peak was below 700 mV. We have now clarified this in the results.

Lines 223 = 397: replace “dietary” with “foraging”; the distinction between foraging preferences and diet is important. Whales can feed on the same prey (e.g. antarctic krill) at two different depths and have completely different isotope signatures, despite having the same diet. Please amend throughout the text. Studies previously using historical samples of whales to show isotopic differences between species should be cited. E.g. Buss et al. 2022. Garcia-vernet et al. 2021

Agreed, we replaced dietary with foraging. We now cite Buss et al. 2023 and Garcia-Vernet et al. 2021.

Lines 225-226 = 278 etc. [rewritten]: True but the small variation in isotopes due to interskeletal variation in turnover is not very relevant to your study. I would remove this.

Agreed, we removed this.

Line 229 = 278, “a distinction between species”: This should be taxa given the lack of distinction between Eubalaena and bowhead whales.

Agreed, we changed this to taxa.

Line 230 = 397 etc, “differences in prey stable isotope composition”: Not true. This should be differences in foraging preferences. Many isotope studies on marine predator niche partitioning show that predators feeding on the same prey in different locations including depth, distance to shore, time of year, etc. have different isotopic signature. Please consider this throughout the text.

We have rephrased this sentence accordingly (it is now in the Discussion).

Line 232 = [moved to SI_7], “to feed largely on krill, at least in this area”: remove “at least” + Which type of krill? Krill is a generic term for a few different taxa. Is this just Euphausia? What about amphipods? Which studies show this?

We have moved the detail regarding the diet of each whale species into Supplementary Information 7, where we add more detail regarding the prey species and supporting references.

Lines 231-232 = [moved to SI_7]: Do we even know what blue whales feed on in the Bay of Biscay/Mediterranean? Do they feed here? Or just migrate past?

We do not know if the single blue whale in our dataset was only migrating past or feeding in the area, but we compared its isotopic signature with that of blue whales migrating through the Azores and found that they overlap (Fig. S7.4 in Supplementary Information 7).

Lines 233-235 = [moved to SI_7]: Or, the lower nitrogen and carbon values of blue and fin whales actually signify their polar and sub-polar feeding habits whilst they are at higher

latitude feeding grounds. Please consider the foraging behaviour and migrations of the whales.

We now discuss the feeding strategy of each species in Supplementary Information 7.

Line 239 = 226, Radiocarbon dating: The radiocarbon dating results should come before the isotope results, not only to reflect the methods but because it is important we are considering animals from the same time period when considering their isotopic niche. If this is not the case, this alongside the caveats associated with this issue, need to be clear to the reader.

The fact that the Discussion starts with radiocarbon dating is suggestive of restructuring mentioned above

In the Results section, the radiocarbon dating results have been moved before the isotope results. It is indeed more consistent with the organization of the Discussion section.

Line 270, general comment on the stable isotope section: The stable isotope data seems lost in the manuscript. Which time periods are these from? Comparing the isotopic niche of different species from different time periods seems odd and the results do not add context to the manuscript story and are not referred to again after. I would include them for two reasons, firstly because they nicely support the ZooMS results (i.e. sperm, balaenidae and fin whales group where you would expect them too and I would use them to discuss the foraging preferences of the different species slightly more in relation to marine reservoir corrections in the discussion. The foraging behaviours of these species are extremely important to the largest caveat of your study, which correction to use. As you state, sperm whales feed in much deeper waters, deeper waters are older waters, and therefore the same correction should not be used for sperm whales vs. more shallow feeding baleen whales. Moreover, coastal shallow feeding species (e.g. grey whale and harbour porpoise) are much more likely to feed in waters close to the atmospheric more likely to resemble local marine reservoirs (for this species I would be more confident in your findings). The migratory Eubalaena and fin whales feed in multiple water bodies at different latitudes, so there is much less certainty from the conclusions from these species. This leads me to my final point on this, the deep waters of the Bay of Biscay are productive and classified of up-welling, up-welling of old deep waters is likely. The animals feeding here would likely have mixtures of older and newer waters and reservoir corrections would need to be considered with caution. What dates do terrestrial evidence from the same sites point to? How do these relate to the whales? This is something that seems to be missing from the text that needs to be included.

We have now provided more detail regarding the stable isotope methods, results and implications. This includes Figure S7.3 (in Supplementary Information 7) which clarifies the time periods associated with the stable isotope data.

As mentioned above we discuss in Supplementary Information 6 why we opted for not attempting to apply a species-specific ΔR based on information from modern whales (and instead use a single ΔR as suitable as possible to the period and region of our samples) and also why we were unable to use paired samples of terrestrial and marine taxa to inform our results.

Lines 282-284 = 306-308: Please clarify, “if our reservoir corrections are correct”. If I have read Figure 3 correctly, the data that push this back are all from Fin whales. Fin whales migrate vast distances including to feed in polar and sub-polar waters during Summer. The Marine Reservoir correction states “Marine20 is intended for calibration of marine radiocarbon samples from non-polar regions; it is not suitable for calibration in polar regions

where variability in sea ice extent, ocean upwelling and air-sea gas exchange may have caused larger changes to concentrations of marine radiocarbon.” The Bay of Biscay may not have been much further South of the ice sheet 20 ka, however, if the corrections are not suitable for high latitude feeding fin whales, it does beg the question of whether your data show evidence of whether whales were being hunted 20 ka, or whether the corrections are inaccurate. It would be really nice to see some sensitive around deltaR for the different species, what is the maximum and minimum deltaR for each species, how would this change your results. This would be really nice to strengthen your findings. Moreover, were the 13 fin whale samples from this early group worked or unworked? The Bay of Biscay is known for rough seas and a cetacean hotspot. I would not be surprised to find 13 stranded whales along this coast during this time period, so the wording through the manuscript has to be more careful on this topic.

Firstly, we need to point out that the reviewer misunderstood one aspect of our interpretation, which is that we discuss evidence of whale utilization as evidenced through bone working, and these were most likely acquired through scavenging of stranded whales. We do not assume there was whale hunting (this is now clearer in the text – see comments below).

We appreciate the reviewer’s concern regarding the use of the Marine20 curve for samples in this region at the time. The Marine20 curve is indeed intended for the calibration of non-polar samples and the Bay of Biscay may actually have been a polar environment at the time. We added a section discussing this in Supplementary Information 6. It reads as follows:

“Heaton et al. (2023) propose a specific calibration of marine samples from polar regions, essentially calculating a range between a minimum and maximum DeltaR. The minimum is based on the Marine20 curve and the maximum is based on the latitude of the sample and the Large Scale Geostrophic Ocean General Circulation Model (LSG OGCM), which is suggested to use for samples from outside ~ 40°S–40°N during glacial periods. Our samples are found between of 43-44 °N, although the whales could of course have been feeding at higher latitudes. However, we do not have the exact location.

Linking back to our manuscript, stating that our results provide evidence of whale bone working from 20ka BP, which is based on two fin whales. If the absolute max DeltaR would be applied, these 2 fin whales would be between 2000-1500-1000 years younger (Fig. 3 in Heaton et al. 2023, although there is a huge amount variation between 71 °N, 88 °N and 53 degrees °N, and again we don’t know where these animals were feeding). It would be more realistic to assume sea ice was most likely covering a large part of the North Atlantic waters, extending much further south than it does today. This also means that many whales were not feeding that far north due to sea ice coverage, making the 71 and 88 °N unlikely, except for beluga’s and bowheads. Additionally, we see with modern fin whales that they feed in various locations, have summer and winter feeding grounds, and can also adapt their feeding strategy to minimise interspecific competition, suggesting that the maximum value would not be a realistic assumption. All these assumptions combined suggest that the maximum DeltaR was possibly ~800 yr or so, although it remains extremely uncertain since all of this is largely based on assumptions. However, the main point here is that this does not change the story of our manuscript because the age brackets we use in our manuscript are 1500 years.”

Additionally, the following text was added to the manuscript line 282-284: “...and push back the chronology of its inception by ca. 2 millennia, (with the caveat of radiocarbon dating uncertainty; Supplementary Information 6).”

Regarding the last point (“were the 13 fin whale samples from this early group worked or unworked?”): all the pieces that we dated are worked objects, except the assemblage from Santa Catalina (site number 8 on the figure), so the 2 earliest occurrences (not 13) on fig.3 are

indeed worked objects: they are referred to as “artifacts” in the text. We added a sentence to the caption of fig.3 to make this clearer. Furthermore, the whale bones from Santa Catalina are not from a stranding site, this is not a natural accumulation, they were found among the faunal remains of a Magdalenian occupation level and show traces of anthropic processing. We added the following sentence in the introduction to clarify the status of the Santa Catalina assemblage: “We also analyze 90 unworked bone fragments from the single assemblage of Santa Catalina cave (Biscay), also ascribed to whale on a visual basis, found among the faunal remains of the site’s Upper Magdalenian occupation and showing traces of anthropic processing (notably percussion notches).”

Lines 285-287 = 313-316: The relationship between the worked artefacts and the bones that could be from strandings needs to be clarified throughout the manuscript. Please recreate figure 3 with two panels (one for worked artefacts and the other for bone fragments that could potentially be from stranding). Do the worked artefacts also show the same pattern?

As stated in the introduction, and at several points in the manuscript, all the unworked bones in our study come from a single site: Santa Catalina. If we created a panel for the unworked bones only, there would be a single dot on the map: we do not find it relevant. The whale bones from Santa Catalina are not from a stranding site, this is not a natural accumulation, they were found among the faunal remains of a Magdalenian occupation level and show traces of anthropic processing. We added the following sentence in the introduction (lines 178-181) to clarify the status of the Santa Catalina assemblage: “We also analyze 90 unworked bone fragments from the single assemblage of Santa Catalina cave (Biscay), also ascribed to whale on a visual basis, found among the faunal remains of the site’s Upper Magdalenian occupation and showing traces of anthropic processing (notably percussion notches).”

Figure 3: “worked and unworked cetacean bone remains”: Please separate these out into two panels.

(See reply above.)

Lines 326-329 = 344-346: ... Or a question of the marine reservoir correction being incorrect. Please consider whale migration and feeding in polar and sub-polar waters for part of the year, and the implications this would have on your results. See my other comments.

Yes, the complexity of the reservoir correction remains present throughout our manuscript, hence we added a new statement in the calibration section.

Lines 337-341 = 369-379: Given where the ice sheet was 20 ka, I would assume bowhead whale are just as likely during the time. I am also very surprised that you only identified one Balaenidae - please discuss the contrast with the van den Hurk et al. 2023 paper and the sites from the same region.

The previous wording in our manuscript could have been ambiguous creating a misunderstanding. We have at least one Balaenid species (right whale and/or bowhead whale), but these are represented by 13 Balaenidae individuals. We have updated the text to clarify this point.

Like Hurk et al. (2023), who also used ZooMS to identify their bone specimens, we were unable to distinguish whether our specimens correspond to right whale or bowhead whale. We now make it even clearer in the discussion that both species are plausible (lines 369-379).

Lines 359-360 = 431-432: (about the scarcity of porpoise) Smaller bones also preserve less well in archaeological record?

A significant preservation bias in disfavor of smaller cetacean bones is unlikely given our archeological contexts (in almost all cases: karstic caves and rockshelters with good preservation of osseous tissues). We added a sentence to clarify that (lines 426-429). It reads: “With the exception of the object from Andernach, all our specimens come from karstic environments (cave and rockshelter sites) with good preservation of osseous material, including small elements. The proportions of the different whale taxa in our sample are thus unlikely to be affected by taphonomic factors such as differential preservation of bone tissues.”

Lines 360-362 = 432-434: What about preservation at Santa Catalina. Fin whales are the second largest animal to ever live on earth and their bones would likely preserve longer, perhaps other cetaceans from this site were bias against because of this.

A significant preservation bias in disfavor of smaller cetacean bones is unlikely given our archeological contexts (in almost all cases: karstic caves and rockshelters with good preservation of osseous tissues). We added a sentence to clarify that (lines 426-429). It reads: “With the exception of the object from Andernach, all our specimens come from karstic environments (cave and rockshelter sites) with good preservation of osseous material, including small elements. The proportions of the different whale taxa in our sample are thus unlikely to be affected by taphonomic factors such as differential preservation of bone tissues.”

Lines 364-365 = 437-439: Fin whales also inhabit the Mediterranean and all species forage close to the islands of the Azores. I would describe them as migratory habitat specialists rather than just offshore whales. Careful.

We reworded the sentence to refer to these species as “large cetaceans that mainly use offshore habitats”. We are unsure why referring to them as “migratory habitat specialists” would have been clearer in the context of the point we are making here, which is that it is extremely unlikely that these species would have been whaled at the time.

Lines 364-368 = 436-451: The point of this paragraph is not clear to me. I would remove or re-work with the following paragraph that makes similar points but with supportive evidence.

The point in this paragraph is to discuss how the whale resources were acquired by the hunter-gatherers, and to stress that the taxonomic composition of the assemblages argues in favor of passive acquisition (strandings/drift whales vs active hunting). We added a sentence to clarify this.

Line 366 = 439, “The active hunting of these species (...) is extremely implausible”: Please rephrase: If reservoir corrections are OK, we have not found technological evidence of humans during this having the capacity for hunting large offshore whales, so we assume active hunting is unlikely.

We rephrased the sentence, but we did not include the mention of the reservoir correction, which we felt was off the point. Whatever incertitude we may have on the reservoir correction, human groups with the technological capacity to hunt large offshore whales are millennia younger, and evidence of this would not be found at Magdalenian sites.

Lines 371-372 = 442-443, “and possibly harbor porpoises”: Why are harbor porpoises separated here. They often feed close to rocky shorelines with occasional seasonal offshore foraging. I would just include them with the other two.

We rephrased to include porpoises with the other two species.

Lines 372-373 = 443-445: Even earlier evidence coming out of Labrador/Newfoundland. See Buss et al. 2023 and refs within.

We could not find this information in Buss et al. 2023 (<https://doi.org/10.1371/journal.pone.0295604>). Most of the references in this article are for historical periods (the last two millennia); and for earlier evidence of active hunting, the authors cite Seersholm et al. 2016, which we also cited. We did add a reference to Buss et al. 2023 for its general perspective on the utilization of whales by humans.

Line 379 = 509-513: I don't agree with this statement for your study site. The Bay of Biscay is a cetacean hotspot because of its deep productive waters. It is one of the few locations that you can observe some of the species that you refer to as “high sea species”. Fin whales can be observed in depths of 100-400ms, and there would have been higher abundances of most species during this time. Please reconsider the bathymetry of the study site and the uniqueness of the Bay of Biscay having deep waters very close to shore. I would reconsider your suggestions that humans would have only encountered more shallow water species from the shore. I'm sure many of these species would have been observed from the shore in such productive waters. On a clear day from a hill of 30 to 100 meters you would be able to see the horizon 12 - 22 miles away.

We have rephrased this text to acknowledge the possibility that all species, including fin whales, could have been seen from the shore, with coastal whales nonetheless likely to be better known.

Lines 382-389 = 513-519: I think this is a really important point. The likelihood of the Bay of Biscay being a feeding ground is quite likely given the bathymetry, ice sheet location and climate during this time. Right whales only seek out coastal shallow waters to calve during winter. Whilst gray whales do need shallower waters to feed (~30m).

Agreed!

Line 391 = 453: This is the first time the worked artefacts are discussed as evidence of whaling. Why not in the results?

As explained in this paragraph, the worked artifacts are not discussed here as evidence of whaling. The worked artifacts are taken as evidence of the use of whale bones as raw material for the manufacture of implements, and the ZooMS results on these artifacts are discussed in this perspective.

Lines 402-404 = 464-467: How would the preservation of dense sperm whale jaw bone differ from the other more porous species?

A significant preservation bias in disfavor of smaller cetacean bones is unlikely given our archeological contexts (in almost all cases: karstic caves and rockshelters with good

preservation of osseous tissues). We added a sentence to clarify that (lines 426-429). It reads: “With the exception of the object from Andernach, all our specimens come from karstic environments (cave and rockshelter sites) with good preservation of osseous material, including small elements. The proportions of the different whale taxa in our sample are thus unlikely to be affected by taphonomic factors such as differential preservation of bone tissues.”

Line 458 = 536-537, “both previously unsuspected”: But why? As a historical marine ecologist, I was not surprised by the species you identified or to the Bay of Biscay being a cetacean hotspot, it is extremely productive with deep waters close to shore and is still a hotspot today. Moreover, it will have been and still in a migration route for many of the larger whale species.

We replaced “unsuspected” with “undocumented for this region at this time period”.

Line 465 = 544, Sample selection, photogrammetry and sampling: Restructure based on comments below and importance of methods to manuscript.

We restructured this part based on the other comments from this reviewer (see previous sections).

Lines 467-471 = 546-549: Could easily condense these sentences.

Done.

Lines 476-477 = 553-555: I don’t understand this sentence. Weren’t all the objects previously identified as Cetacea? Why are these 41 mentioned here?

As is explained in the sentence, 41 corresponds to the number of bones that had been identified as whale bones at Santa Catalina (and published as such by Castaños, 2014) before our study began. As a part of this study, the collection was reassessed and the sample was increased to 90 potential fragments of whale bone. We added the indication “before this study” to clarify this.

Lines 483-485 = 560-562: After photogrammetry. This includes some results.

We note in the following lines that photogrammetry was done before any sampling.

Lines 487-493 = 564-571: Where are the photogrammetry results in the main text, evidence of early exploitation based on the artefacts should be mentioned somewhere in the results. What was the photogrammetry used for? I presume the 3D models were used to “preserve” a copy of the specimens before destructive sampling, but why 3D model all the specimens and not only the ones that were sampled? What was this information used for in your study? The first time this is mentioned as evidence of early exploitation is in the discussion.

The photogrammetry was only used for conservation reasons, as a way to preserve a digital copy of the objects before sampling. The photogrammetry data were not used as evidence of early exploitation, this is why they are not mentioned in the results. Only the objects intended for sampling were subjected to photogrammetry. We added a sentence to make this clearer (line 565-566). It reads: “This was done for conservation reasons, in order to preserve a digital copy of the specimens before they were morphologically altered by the sampling.”

Lines 493-498 = 573-576: How much did you sample? Why?

We added the mean masses for the different categories of samples (sampling for ZooMS, sampling for 14C dating on worked objects, sampling for 14C dating on unworked bone fragments: lines 576-579). The sampling strategy was adapted case by case—depending on whether the specimen was intended for ZooMS and 14C or only for ZooMS; on the degree of bone preservation, allowing us to take a solid fragment or only bone powder; on the size and morphology of the specimen, heritage significance and museum instructions, that permitted or prevented certain sample sizes. For this reason, it is difficult to give anything else than object-by-object explanations for sample masses.

Lines 501-510 = 581-591: I find the ZooMS methods section slightly confusing. How many samples did you try the minimal methods for? Which ones? Did they all fail completely or just get poor spectra? I'm not sure it is even necessary to include the minimal method in your methods but if you are going to include it, perhaps just a short sentence maximum at the start of your ZooMS methods stating that you tried it and it didn't work and then put the methods you actually used all in one place, so the reader doesn't have to jump back and forth between sections for the actual destructive method with a list of samples you tried for both methods and their results clearly in the supplementary.

We feel it's worthwhile to note the lack of success with the non-invasive method. Considering the antiquity of the samples, a non-invasive sampling method is more ethical than a destructive sampling method, and other researchers may be interested in the relative success of both methods. We have condensed and rearranged the methods section so that the information flows more logically between the different methods.

Line 511 = 588-589, "the majority of the modified bone objects": The majority sounds different to the 83 and 41 specimens you listed above? Please clarify.

We added the precise numbers to clarify.

Line 512 = 590, "the unmodified bone samples": Repeat from previous sentence?

This sentence applies to all our ZooMS samples, not just the unmodified ones. We changed this.

Lines 524-525 = 602-604: What was the minimum number of spectra peak used to identify each species?

The minimum number of peaks used for an identification depends on the taxonomic level of certainty for each sample, and the presence of species-specific (or diagnostic) peaks. For example, specimens assigned to the 'Cetacean' category might be identified based on the presence of m/z 1079 and a combination of non-diagnostic peaks. Whereas for other specimens, species could be assigned species-diagnostic peaks even if not all markers were present. We have included an additional Supplementary Table (Supplementary Information 4) with the observable peaks listed for all samples.

Line 528 = 616, "subsamples of bone and bone powder": How much?

We have added the range of mg used for analysis.

Lines 535-574 = 623-649: These two sections are extremely lengthy and could easily be shortened. Suggestion to restructure:

- During initial digests, organic glue contaminants (bovine) were detected for some samples. Therefore, we re-digested X number of samples as follows ..., which significantly improved spectra results (see supplementary).

- XX and YY samples were run on the MALDI-TOF-MS using the following settings ..., from single and double digests, respectively.

Section condensed and re-organized as noted above.

Lines 535-552 = 623-649: Similar to above. This could easily be condensed into one section on “ZooMS” and the double extraction could easily come before the MALDI-TOF info, so no repetition is required.

Section condensed and re-organized as noted above.

Lines 555-561 = 623-649: Based on this, it is not clear to me why you tried the minimal method in the first place?

Considering the antiquity of the samples, and the importance of maintaining the morphology, the museums preferred to conduct a non-invasive sampling method ahead of destructive sampling attempts.

Lines 561-563 = 623-649: Repetitive/discussion - remove from methods.

Section condensed and re-organized as noted above.

Figure 4: move to supplementary.

Figure 4 was moved to supplementary information (SI_5).

Line 590 = 656, “Bone collagen extraction protocol”: This should come in its own section as collagen extractions were for both radiocarbon dating and stable isotope analysis.

Agreed, this has been changed.

Line 597-598 = 656-666, “submerged in 0.1M NaOH for 20 min (if discoloration appeared, new NaOH was added for another 20 min)”: These are very old samples, I’m surprised all humics were removed within 40 mins maximum of NAOH. Is this correct?

In addition to the NaOH step, the samples were also treated with XAD resin, which removed any further humic acids, if still present after the NaOH wash.

Line 600 = 668, “at 90°C until dissolution”: This is very hot and likely to start degrading the collagen, is this correct?

Potentially yes, but as long as collagen is in solution afterwards and filtered, this is not an issue. We obtained normally looking collagen (fluffy, white, as expected (not for all samples of course)). The most important thing is that gelatinization is done below 100 degrees and above 65 degrees.

Between lines 601 and 602 = 670-672, add a title “Radiocarbon dating” (see comment on “Bone collagen extraction protocol” section above).

Agreed, this has been changed.

Line 602 = 673-676, “The next steps follow the XAD resin approach, which was developed by Stafford et al.”: to do what?

FTIR-ATR was performed to get a better understanding of the collagen preservation in these samples. However, this analysis also revealed that some of the bone samples had anomalous peaks in FTIR-ATR spectra, most likely indicating the presence of glue contamination. XAD resin was used to ensure this type of contamination was eliminated, as the classical ABA treatment may not always do this. The following text was added to the manuscript: “As anomalous peaks were observed in FTIR-ATR spectra of several samples, most likely indicating the presence of glue or consolidant, the XAD resin approach was applied to eliminate this type of contamination. This approach was developed by Stafford et al. (1982, 1987, 1988), is also described in van der Sluis et al. (2023b) and consist of the following steps:...”

Line 628 = 700, “Measurement parameters such as ^{12}C current and ^{13}CH current were checked”: for what?

The current values are generally checked (i.e. assessed), which is automatically done in the software for the ^{12}C current. For normal samples ($>500\mu\text{gC}$) the high energy ^{12}C current should be $> 10\mu\text{A}$ at least. For the ^{13}CH current, it is not systematically checked but it is systematically taken into account with the applied ^{13}CH -correction. when there is a problem with it, it is very quickly visible because of the error bar which becomes larger. In other words, the ^{12}C and ^{13}CH currents are parameters that are “monitored” during ^{14}C measurement. The following text is added to the manuscript: “Measurement parameters such as ^{12}C current and ^{13}CH current were monitored during ^{14}C measurement.”

Lines 636-637 = 707-715: Should this be a new sentence?

Done.

Lines 658-666 = 728-736: What other QC checks were done on the data aside from CN ratios? Were samples run in triplicate? These are very old samples. What international standards were used? Using only one standard doesn’t allow for linearity corrections. I feel like some details are missing here

In most cases, we were very lucky to have enough collagen to perform both radiocarbon dating and stable isotope analysis, as we were dealing with small samples that were also in some cases contaminated from consolidation, hence the XAD treatment. We were able to do a few duplicate measurements (5 samples in total). If we compare the two datasets using a Wilcoxon signed rank test (non-parametric test for small sample sizes) in R, it unfortunately gives back a message (*cannot compute exact p-value with zeroes*) as there are a few ties in the dataset precluding this analysis, which is a good sign. The maximum difference is 0.3%. Even though we cannot test this statistically, the values appear very similar. We do not report these values in the manuscript.

Identifier	$\delta^{15}\text{N}$ (‰)	$\delta^{13}\text{C}$ (‰)
21042	16.5	-12.4
21042.duplo	16.5	-12.5
20074	11.55	-14.27
20074 duplo	11.42	-14.34
20124	5.95	-20.46
20124duplo	5.61	-20.45
21056	2.7	-18.5
21056 duplo	2.8	-18.5
20124	5.7	-20.8
20124 duplo	5.7	-20.5

Moreover, 3 primary standards (from IAEA) were used to calibrate the Alanine for $\delta^{15}\text{N}$ air: IAEA USGS-25 (ammonium sulfate), IAEA N1 (ammonium sulfate) and IAEA N2 (ammonium sulfate). And 1 primary standard (from IAEA) was used to calibrate the Alanine for $\delta^{13}\text{C}$ V-PDB: IAEA 600 (caffeine).

The following text was added: “Three primary standards were used to calibrate the Alanine for $\delta^{15}\text{N}$ air: IAEA USGS-25 (ammonium sulfate), IAEA N1 (ammonium sulfate) and IAEA N2 (ammonium sulfate). And 1 primary standard (from IAEA) was used to calibrate the Alanine for $\delta^{13}\text{C}$ V-PDB: IAEA 600 (caffeine). Analytical precision is $\pm 0.2\text{‰}$ for both $\delta^{13}\text{C}$ and $\delta^{15}\text{N}$ values. Five samples were run in duplicate and the difference fell between 0-0.3‰ for $\delta^{13}\text{C}$ ratios, and 0-0.34‰ for $\delta^{15}\text{N}$ ratios.”

Line 660 = 728, “were weighed into tin capsules”: size?

These were large (157 μl) universal soft tin containers for archeological samples (OD = 5mm, H = 8mm, PN=240-06400 from Thermo Scientific). We added “(5x8 mm)” to the text.

Lines 661-662 = 729-730, “Isotopic values of all samples were measured...”: Is this correct? Or was alanine used for the proportion of C & N within the sample rather than the isotopic ratios?

Yes, both the $\delta^{15}\text{N}$, $\delta^{13}\text{C}$ and [%N], [%C] measured values were normalized with the Alanines of the run.

Line 662 = 730, “the laboratory standard alanine”: company?

This alanine was bought at Fisher Scientific (100g, Laboratory reagent grade L-alanine, chromatographically homogenous).

Lines 664-665 = 746-747, “Acceptable stable isotope ratios have an atomic C:N ratio that falls between 2.9-3.6 (DeNiro 1985)”: Please include citation Guiry and Szpak 2020 Methods in Eco and Evo.

It would be unnecessary to mention this paper, as we do not use the more stringent criteria for C:N ratios, since we are dealing with ancient rather than modern material, for which the tighter C:N ratio range would be required (please see comment above).

Reviewer #2 (Remarks to the Author):

The authors present a manuscript looking at reconstructing prehistoric exploitation of large cetaceans using ZooMS and isotope analyses of worked bone material.

However, overall it was left unclear as to whether this study is investigating palaeoecology OR human seashore interactions – the latter is much more nuanced, and not well achieved in the presented work... ‘Past ocean ecology’ to many people would mean a much broader topic than the cetaceans discussed here...

The ‘multiproxy’ analysis is not targeted at answering one question, but seems to be a collective of different methods, answering different questions, with the only thing in common being that they are contemporary...

The reference to “past ocean ecology” has been removed from the title (see above). Regarding the other remarks, we believe that our manuscript contributes to understanding ancient human-seashore interactions by illuminating which whale resources were present, when and how they were accessed and exploited, etc., which implies a better understanding of whale ecology at that period.

Commenting on the manuscript more specifically:

The abstract suggests that the finding of five species of large whales ‘significantly’ expands the range of known species, but makes little effort to clarify what species and why this is potentially interesting. Likewise, the brief statement on stable isotope results that ‘reflect species-specific preferences’ is particularly uninformative.

The abstract was kept concise in order to comply with the journal’s limit of 150 words. We agree that giving more information would be useful, but adding the species list, elaborate upon their interest, or give details re. the stable isotopes results would take us beyond that words limit, and we can see no information that could be deleted from the present version in order to “make space” for something else.

[Lines 280-285 = 303-307?] The one clear statement relates to the radiocarbon dates, suggesting that the study involves the oldest known evidence of whalebone working, but lacks clarification on how we can have confidence that it is the activity that is being dated, rather than the potential for re-worked older material.

We added two sentences to clarify this: “The whale bones used for tool manufacture do not necessarily come from fresh carcasses, but they are unlikely to have been more than a few decades old, given the hazards of bone preservation in the open air on the seashore (Christensen 2016)—the low density of cetacean bones (Gray et al. 2007) suggesting that they will be particularly subject to density-mediated attrition (Lyman 2021). Accordingly, at our level of chronological resolution, the date of the material is essentially the same as the date of the bone working activity.”

Introduction:

Line 116-118 = 122-127 – needs citation for whales being key part of subsistence of prehistoric coastal populations, and how this is evidenced – if this statement stems from the sources mentioned in subsequent sentences, then restructure the paragraph accordingly

We restructured this passage to ensure that the references to these points are clearly presented.

Line 119 = 125-126 – It seems important to clarify how archaeologists can distinguish between the exploitation methods briefly mentioned – for example, scavenged vs hunted, and the extent of organization of the latter.

This point is addressed in the articles cited. Charpentier et al. 2022, notably, is a review paper with a section devoted to the identification of ancient whaling techniques. The issue of whaling in our specific archeological context (the final Pleistocene in western Europe) is also addressed in the Discussion section. We felt that developing this point in the introduction would lengthen it without necessity.

Line 121 = 127 – This is before moving on to the issue of site loss by nature of many being coastal

Agreed; no text modification implied.

Line 128 = 135 – unclear how the unworked whale bones have contributed to the evidence of the beginning of whale exploitation – perhaps clarify that this relates to fragmentation patterns with human agency implied, if this is the point being made

The bones are unworked (i.e., not used as raw material for tool manufacturing) but they were brought to the site and processed by humans. We added that information.

Unclear what distinguishes the first two paragraphs in terms of their message, structured awkwardly in that the first goes into details to introduce whale exploitation, and the second more widely at coastal exploitation. Likewise, the start of the third paragraph is awkward and seems as though would be better as a continuation of the previous paragraph

It is a matter of personal preference to either start with broad statements before going into details, or start by stressing the main focus of the work (in this case whales) and then explain why it illuminates broader issues. We prefer to keep the structure as it is, but we put the third paragraph in continuation with the second as requested. The end of the first paragraph was modified after another comment from Reviewer 1 (see above).

Line 161 [removed] – awkward/incomplete sentence (Dietary data and feeding ecology of these whales...)

This sentence was removed in the revised version.

Line 173 = 181-183 – Unclear why neither of the citations for ZooMS in its first instance refers to the first publication of the method, or its first application to the study of cetaceans, nor is ZooMS defined

We have defined ZooMS, and updated the publications to the first applications of ZooMS in archaeology.

Line 175 = 185-186 – statement “This work enabled us to assess the range of whale species...” should be toned down somewhat, given that there are clear biases against the presented range reflecting the ‘true’, full range of whale species present – other than what could be expected for known palaeobiogeographic distributions of such species

We agree that the presented range might not reflect the ‘true’, full range of whale species present. For this reason, we wrote that “This work enabled us to assess the range of whale species exploited by the Magdalenian foragers” (and not “the range of species present”). We added the restriction (“the range of taxa identified here might not reflect the full range of species present in the Bay of Biscay at that period”) in the following sentence.

Line 190 = 202 – ‘while 8 were identified as large terrestrial mammals’ is an interesting point that should be more detailed – i.e., what might be causing these misidentifications

We added a sentence to explain this (lines 204-207). It reads: “The visual misidentification of 8 objects made of bone from large terrestrial mammals was due to their thoroughly porous aspect, normally a diagnostic feature of whale bones, but also present in some anatomical elements of certain terrestrial species (in this case mammoth, rhinoceros, reindeer and equids), and that can be misleading when dealing with small, fragmented objects.”

Line 191 = 203 – the fact that 90% of ‘macroscopic visual attribution’ to the identification as a ‘whale’ is interesting from a scientific perspective, but not particularly informative to the wider story of past ecology and/or human seashore interaction, whichever of these the study aims to address

We removed this sentence since R1 also found it ambiguous.

Line 193 = 203 – The different levels of initial morphological identification were not particularly clear on first reading, so this sentence about positively classified and likely made of, etc., was difficult to follow – should be re-written or better introduced

We removed this sentence since R1 also found it ambiguous.

Line 196 = 208 – It at first seems odd that you have more correct ‘identifications’ with the worked material, than you get with the naturally fragmented material

We note in the following sentence and in the methods section that visual selection of unworked material was more conservative and inclusive, to maximize the number of cetacean bones sampled.

Line 208 onwards = 220 onwards – some of this reads as Discussion rather than Results

The last part of this paragraph was moved to the Discussion.

Line 211 = 223-224 – would be useful to state when the first known appearances of these newly identified taxa, fin, grey, blue, right, and bowhead whales, are to the North Atlantic

This comment and a related one by reviewer 1 made us realize that our previous wording was indeed ambiguous. We are reporting here on new results showing that these species were accessible as sources of resource to Magdalenian coastal hunter-gatherers, not on novel results showing these species were already present in the North Atlantic at this point in time. As reviewer 1 notes, the latter would have been trivial, as all of these species are native to the North Atlantic, and thus expected to have been present in this Ocean basin much earlier than the Magdalenian period. For example, and as we mention later in the manuscript, there are bone records of gray whales in the North Atlantic going back to the Pleistocene (Alter et al. 2015). We have revised the text accordingly.

We have now expanded the discussion to compare more explicitly the cetacean communities found today in the Bay of Biscay with those that were found in the Magdalenian period (lines 354-379).

Line 219 = 271 – this whole section on stable isotope analyses seems to not add anything to the story of ancient human seashore interactions, though perhaps this whole study is more of interest to palaeoecology – if so, would be interesting to clarify this

Following the advice of reviewer 2 below we have restructured this section to focus here on the results obtained, leaving the discussion for later. We clarify in the introduction (line 186-188) that “The results of the stable isotope analysis, placed in the context of patterns of isotopic niche partitioning among contemporary whales, shed light on the whales’ relative foraging behavior and past ocean ecology.”

Line 225-227 = 278 etc. [rewritten] – This sentence also reads more like it belongs in the Discussion, but it is unclear if this information is even appropriate – if so, it would benefit a re-structure and re-write – the high number of citations within this paragraph make it clear how out of place it is in the Results

Agreed, we have rewritten this section to focus on the results, with the interpretation presented in the Discussion.

Line 245 – so here it is made clearer that there is chronological support linking the stratigraphy with the radiocarbon dates, which could come across better earlier on

We added a sentence in the introduction, lines 168-170, to make this link more explicit (see our reply to a preceding comment above).

Figure 2 – Although it is clear how these isotope ratios differ between species, as would be expected given differences in diet, it remains unclear what the message is with this study – i.e., if this is to look at what ‘past ocean ecology’ was, it would be more interesting to make comparison with values from extant animals

We agree, and have now included an extended section (Supplementary Information 7) where we make this comparison.

Line 280 = 303 – so if this is more a confirmation of early evidence of cetacean bone working, it becomes more important to give this study novelty by discussing the significance of the new taxa found in this region

As explained in the introduction and in the discussion, the fact that foragers from the Magdalenian period worked whale bone is not a novelty. The novelty is the construction of a more precise chronological framework thanks to direct ¹⁴C-dating—pushing back the inception of the phenomenon by 2 millennia, pinpointing a “peak” period of ca. 1.5 millennium when most of the objects are produced (and followed by their near disappearance), spatializing the dated objects to show a progressive diffusion towards the Pyrenees (fig.3), etc. This is more than confirmation. The significance of the newly identified taxa is discussed in the following paragraphs of this section.

The fact that the Discussion starts with radiocarbon dating is suggestive of restructuring mentioned above

In the Results section, the radiocarbon dating results have been moved before the isotope results. It is indeed more consistent with the organization of the Discussion section.

Numbers on Figure 3 are difficult to read

This has been changed.

Line 323 = 344 – ‘still had access to whale bone’ and ‘still had the skills’... perhaps the authors here could clarify why populations may lose such skills, particularly given that ‘they do not differ from those used in the osseous industry in general’

Since this hypothesis was unlikely anyway, and apparently confusing, we removed this part of the sentence.

Line 376 = 448-451 – As referred to, one of the greatest challenges in making interpretations relating to human-cetacean interaction is that of scavenged vs hunted – the authors claim that this is irrelevant for their interpretation relating to how ‘well placed these species must have been’, but the following sentences are not particularly convincing as to why this is the case.

We revised this section to make it more explicit what our position is: we consider it unlikely that whales were actively hunted, because the bone assemblage is dominated by species that are not coastal, and because there is no archaeological evidence that hunter-gatherers at the time already had the necessary technology (lines 445-451).

Reviewer #3 (Remarks to the Author):

The article is an important contribution to the literature on hunter-gatherer-fisher populations and an example of early cetacean use during the Upper Paleolithic. The authors provide the reviewers and readers with sufficient data to contextualize their study and results. In addition, the manuscript is a new and vital data point in the growing ZooMS literature. Their use of various ZooMS methods and comparison of the results are important contributions to the literature.

I have questions regarding the Delta R applied, given the time depth and other related issues the authors discuss. Still, I recognize that calibration and radiocarbon dating of any marine organism, let alone cetaceans, can be complicated.

As mentioned above, following comments from reviewer 1, we now discuss extensively in Supplementary Information 6 how we derived the DeltaR value we used, acknowledging the difficulties associated with radiocarbon dating of cetaceans.

I did encounter minor grammatical and spelling issues in the draft, but these can be quickly remedied if accepted.

Overall, I greatly appreciate the author's contribution and look forward to seeing it published.

Reviewer #4 (Remarks to the Author):

Dear authors,

I applaud your efforts in conducting this multi-approach study of some of the oldest evidence for prehistoric whale exploitation. The article is well written and, with some technical clarification and editing, will make a good contribution to the journal. See my comments below:

Diagenetic Alteration and Collagen Quality Criteria: The range (2.9 and 3.6) of C:N ratios established by DeNiro (1985) has long been the standard for good collagen preservation. However, studies by Klinken (1999) on archaeological bones and Guiry and Szpak (2020) on modern bones have suggested that values outside the 3.0-3.3 range for mammals may result in distorted isotopic compositions due to contamination by non-collagenous carbon. I suggest removing the samples with C:N above 3.3 and reanalyzing the data set. Alternatively, you could assess collagen quality by comparing $\delta^{13}\text{C}$ and $\delta^{15}\text{N}$ versus C:N for your whale samples (see Guiry and Szpak, 2021).

We acknowledge the concerns raised by the reviewer regarding the followed atomic CN ratio range followed, and we thank the reviewer for the suggested references.

Following advice from the reviewers (see below) we now have included as supplementary information a comparison between the isotopic signatures of our Paleolithic samples with those of modern counterparts (Supplementary Information 7), where was also discuss the differences in atomic CN ratios. It reads:

Collagen quality control

As diagenetic alteration can affect stable isotope values as well as radiocarbon ages, only results from good quality collagen with an atomic CN ratio between 2.9 and 3.6 ought to be used (DeNiro 1985). However, studies by van Klinken (1999) on archeological bones and Guiry and Szpak (2020) on modern bones have suggested that values outside the 3.0-3.3 range for mammals may result in distorted isotopic compositions due to contamination by non-collagenous carbon. We compared the values for whale species that agree with different atomic CN ratio ranges (2.9-3.3 and 3.3-3.6) (Fig. S7.1). In these figures there is considerable overlap between samples with an atomic CN ratio up to 3.3 and 3.6, only the gray whale samples show lower carbon stable isotope values (circa 1‰), although this could also be due to the small sample size. We believe most of the variation visible here is due to individual variation between the whales, rather than diagenetic alteration. Whales are exceptionally large (marine) mammals, thus resulting in a larger variation in stable isotope ratios in their bone collagen due to their longevity and slow bone turnover time. Samples with atomic CN ratios between 2.9-3.6 were included in the discussion further below.

Figure S7.1. $\delta^{13}\text{C}$ and $\delta^{15}\text{N}$ ratios of Paleolithic whales (this study), showing the differences between the different atomic CN ratios per species.

Additionally, stable isotope analysis was done on collagen, while ^{14}C dating was performed on collagen after XAD resin treatment, which would have removed any, if present, remaining contamination. When examining the ^{14}C ages against atomic CN ratios from a single site (Santa Catalina) (Fig. S7.2), there is no evidence of a correlation that might indicate contamination.

Figure S7.2. ^{14}C ages plotted against the atomic CN ratio of material from the site of Santa Catalina. A correlation between the two might indicate contamination, which is not the case.

Stable isotope analyses: The methodology section states that "three samples were analyzed at the Silver Lab at the University of Vienna", but does not explicitly state where the remaining samples were analyzed (apologies if I missed it). This may be important as different labs may use different pretreatments or analyses (instrumentation or data calibration), which may also cause variation in the isotope ratios (e.g., Pestle et al., 2014). Did you consider this?

We had previously included this information under the bone collagen section, but have now restructured the methods, adding more details about the samples analyzed in Vienna. All stable isotope related information is now available under the section "Isotope Ratio Mass Spectrometry", which comprises the following text:

“Bone collagen samples (320-380 μg) were weighed into tin capsules and analyzed with a Thermo Scientific EA Flash 2000 coupled to a Delta V Advantage isotopic mass spectrometer at the SSMIM (Service de Spectrométrie de Masse Isotopique du Muséum; Paris). Isotopic values of all samples were measured relative to the laboratory standard alanine, which has a reproducibility of 0.3 wt% for N and 0.6 wt% for C. $\delta^{13}\text{C}$ and $\delta^{15}\text{N}$ values are reported relative to the VPDB and AIR, respectively. Three primary standards were used to calibrate the Alanine for $\delta^{15}\text{N}$ air: IAEA USGS-25 (ammonium sulfate), IAEA N1 (ammonium sulfate) and IAEA N2 (ammonium sulfate). And 1 primary standard (from IAEA) was used to calibrate the Alanine for $\delta^{13}\text{C}$ V-PDB: IAEA 600 (caffeine). Analytical precision is $\pm 0.2\text{‰}$ for both $\delta^{13}\text{C}$ and $\delta^{15}\text{N}$ values.

Subsamples (0.2-0.3 mg) of extracted bone collagen from six samples (Hum4, Hum5, Hum8, Hum9, Hum14 and Hum18) were taken for the analysis of isotope ratios of carbon and nitrogen by elemental analyzer-isotope ratio mass spectrometry (EA-IRMS; Thermo Scientific EA-Isolink with a Flash 2000 coupled to a Delta V Advantage isotope ratio mass spectrometer) in the Silver Laboratory (Large-Instrument Facility for Advanced Isotope Research) at the Center of Microbiology and Environmental Systems Science of the University of Vienna. Stable isotope values of all samples are measured relative to the laboratory standard alanine, which has a reproducibility of 0.07 wt% for N and 0.1 wt% for C. $\delta^{13}\text{C}$ and $\delta^{15}\text{N}$ values are reported relative to the VPDB and AIR standard, respectively, and are measured with an analytical precision of $\pm 0.1\text{‰}$ for both $\delta^{13}\text{C}$ and $\delta^{15}\text{N}$ values.

Acceptable stable isotope ratios have an atomic C:N ratio that falls between 2.9-3.6 (DeNiro 1985).”

Interpretation of $\delta^{13}\text{C}$ and $\delta^{15}\text{N}$ values: The authors correctly point out the significant sources of variation affecting the stable carbon and nitrogen isotope composition of their whale specimens. I agree that it is extremely challenging to carefully evaluate them; however, a more thorough assessment of the patterns of stable isotope composition of the specimens may help move this issue forward. You focus your discussion on describing the results and commenting on how they might match expected patterns of niche partitioning among cetacean species. However, you don't explicitly show the patterns of isotopic niche partitioning among living species, so their use as a reference seems a bit weak. How do patterns of the stable isotope composition recovered for the archaeological assemblage (Figure 2) contrast with those exhibited by modern representatives in the area? I am not proposing a quantitative comparison of the stable isotope composition of archaeological vs. extant assemblages (that would be problematic due to shifting baselines, differences in tissue type, etc.). Instead, I am primarily interested in whether the general distribution in the isospace (and hence, the inferred ecology) of ancient species is (or is not) consistent with that of their modern counterparts living in the Bay of Biscay. This large-scale comparison will provide independent evidence to further assess and contextualize the degree and possible impact of sources of variation on the carbon and nitrogen stable isotope composition of the ancient whales, including the issue of time averaging. I did a quick search on Google Scholar and found a large number of studies conducting stable carbon and nitrogen isotope analyses on modern species that can be used for this task. In addition, the examination of the stable isotope composition of modern representatives would likely provide clues to the question of the identity of ancient individuals assigned to the North Atlantic right/bowhead whales' group: Are the relative $\delta^{13}\text{C}$ and $\delta^{15}\text{N}$ values of individuals in this group similar with those of modern right whales inhabiting the area or bowhead whales from the Arctic?

We thank the reviewer for this suggestion that we have followed by now including (in Supplementary Information 7) a detailed comparison between the stable isotopic signatures of

ancient whales with those of their modern counterparts. This new analysis enriches the discussion substantially. Unfortunately, it does not allow for a clear-cut answer to the question of ancient individuals assigned to right/bowhead whales, as there is some overlap between the modern isotopic signatures of the two species, although it does suggest bowhead whales are the most likely.

A small note in Figure 1b: Shows that one of the bone specimens was identified by ZooMS as belonging to a seal/sea lion. No sea lions (living or extinct) have ever been identified in the North Atlantic region (Berta et al., 2018; Valenzuela-Toro & Pyenson, 2019). Therefore, it can be confidently stated that this specimen is from a seal.

This is a good point. We have updated Figure 1b to remove 'sea lion', and noted in the text that one seal was identified in addition to large land mammals.

Something that caught my attention is the way the raw data is presented. It is a very long PDF where the sample IDs and the data are on separate pages, making it extremely difficult to relate them. I literally had to print them out and glue them together to make sense of the numerical results. I don't know if this is due to the formatting constraints of the submission, but if it is published, I urge the authors and the editor to ensure that the raw data is presented as an Excel file or something similar.

It was indeed presented as an Excel file, but the submission system apparently converted it into a pdf. We agree that the Excel-type table format should be used in the final publication.

References

- Berta, A., Churchill, M., & Boessenecker, R. W. (2018). The origin and evolutionary biology of pinnipeds: seals, sea lions, and walruses. *Annual Review of Earth and Planetary Sciences*, 46, 203-228.
- Guiry, E. J., & Szpak, P. (2020). Quality control for modern bone collagen stable carbon and nitrogen isotope measurements. *Methods in Ecology and Evolution*, 11(9), 1049-1060.
- Guiry, E. J., & Szpak, P. (2021). Improved quality control criteria for stable carbon and nitrogen isotope measurements of ancient bone collagen. *Journal of Archaeological Science*, 132, 105416.
- Pestle, W. J., Crowley, B. E., & Weirauch, M. T. (2014). Quantifying inter-laboratory variability in stable isotope analysis of ancient skeletal remains. *PLoS one*, 9(7), e102844.
- Valenzuela-Toro, A., & Pyenson, N. D. (2019). What do we know about the fossil record of pinnipeds? A historiographical investigation. *Royal Society Open Science*, 6(11), 191394.
- Van Klinken, G. J. (1999). Bone collagen quality indicators for palaeodietary and radiocarbon measurements. *Journal of Archaeological Science*, 26(6), 687-695.

Manuscript NCOMMS-23-62988, “Oldest whale bone tools reveal past whale ecology and ancient human-seashore interactions”: detailed response to reviewers

Black lettering = reviewers’ requests organized by theme.

Blue lettering = authors’ responses and comments.

Reviewer #1

I would like to begin by thanking the authors for their hard work and for the substantial additions, particularly in the supplementary sections of the manuscript. I have a few additional comments that need to be addressed before I can recommend the paper for publication. Some of these comments are in response to previous discussions, while others are new and arise from the updated text and supplementary material. Overall, I am pleased with how the authors have addressed all other outstanding concerns.

Abstract:

- Many of the responses to reviewer comments confirm that they are not suggesting whale exploitation was likely but are in fact discussing utilization. I would argue that exploitation and utilization are not quite the same. Please amend this throughout the abstract and manuscript, especially as using a stranded whale carcass. The exploitation is the “practice of harming or killing animals for human benefit”, if you do not want the manuscript to suggest they purposefully killed the whales, this needs to be reflected better throughout the manuscript. Please rephrase appropriately.

There was indeed a misunderstanding regarding our use of the word “exploitation”. As it is usually employed in prehistoric archeology, the expression “the exploitation of animal resources” applies to any organized use of animal resources, whatever the acquisition method, and that includes collecting and scavenging (as per <https://doi.org/10.4000/palethnologie.5137>), thus does not necessarily imply active hunting. In our manuscript, it was defined as such in the first lines of the introduction (“exploitation methods that included scavenging freshly beached animals, opportunistic killing and organized whaling”). We changed the word for “utilization”, “acquisition”, or similar expressions, throughout the manuscript.

Introduction:

- “Whales are the largest living animals on Earth and their current populations are a mere fraction of their abundance in the past (Christensen 2006).” I do not entirely agree with this sentence. Some species and populations are back to over 90% of historic population size and some at carrying capacity (humpback whales and SRW are doing particularly well), fin whales in the North Atlantic are somewhere between 50 – 70% of prewhaling size, and many we don’t have historic estimates for. Please clarify “for many species”, or similar.

We changed for “the current populations of many species” line 131.

- “Overall, ZooMS analysis of 131 cetacean specimens reveal the presence of at least six cetacean taxa in the northeastern Atlantic during the Magdalenian (Figure 1 and

Supplementary Information 4): fin whale, Balaenoptera physalus (n = 65); sperm whale, Physeter macrocephalus (n = 32); gray whale, Eschrichtius robustus (n = 11); blue whale, Balaenoptera musculus (n = 2); one species of porpoise (harbor porpoise or Dall's porpoise, Phocoenidae, n = 1); and at least one species of Balaenid whale (Balaenidae), with 13 samples that can be attributed either to the North Atlantic right whale, Eubalaena glacialis, or to the bowhead whale, Balaena mysticetus (two species that are indistinguishable using ZooMS)." Following on from one of my previous points, please add "both present in the North Atlantic". SRWs and NPRWs cannot be distinguished either, but they are unlikely to be in the North Atlantic. This should be stated somewhere.

We added "both present in the North Atlantic" line 229-230.

Results:

- It currently seems a bit like you are overselling your change in date of earliest "exploitation" given your error of 1500 yrs. The error suggests it might only be 500 years difference to previous findings, no? Or am I missing something? And as there were only two early samples that caused this shift, I would encourage more modest wording around this finding. Reword results etc. appropriately to capture this.

We agree that "push back the chronology of its inception by ca. 2 millennia" (our previous wording) can sound like overselling, since there are uncertainties with the reservoir effect and it is based on two samples only. The dates for these two samples are around 19.5 ka cal BP with our calibration, while the previous contextual dates (on terrestrial material) were around 18-17.5 ka cal BP. Even if our calibration is off by a few centuries, there is at least a one millennium difference. We reworded in this perspective (line 322): "at least one millennium".

- Check font sizes of delta notations

It is font size 10, superscript letters. We will follow the journal's instructions if this needs to be changed.

- I would also argue you have potentially higher d15N in the NARW, as the mean of those baleens would sit in-between the min and max d15N vales, thereby lower than your historic samples (if they are indeed NARWs), see comments in discussion below about why I don't fully agree with your statement suggesting they might be bowhead whales. I would hypothesis from your findings that the d15N baseline of offshore pelagic waters was much lower in the past (perhaps from high productivity) relative to coastal waters. If this was a northerly feeding ground. The fin whale and blue whale thereby would have lower d15N than the coastal NARW and gray whale – the trophic level conundrum than is no more. Of course, this is just a theory, but probably quite plausible under the environmental circumstance of this time period and the bathymetry of your study region. Perhaps this could be captured somewhere?

The following text was added lines 451-452 (also in relation to the comment further down):" It also suggests a higher likelihood that the Balaenidae samples correspond to bowhead whales rather than right whales (Figure S7.4) although this is less certain. *While the Arctic conditions in the region would favor the bowhead hypothesis, the region could similarly have functioned as a northerly feeding ground for right whales.*"

Discussion:

- Please change exploitation to utilization here: *“The exploitation of fin whale carcasses at Santa Catalina after 16 ka cal BP, and the presence of a foreshaft made of gray whale bone at Andernach with contextual dates younger than 16 ka cal BP, both show that hunter-gatherer groups still had access to whale bone at that period.”*

We changed to “acquisition”.

- I don’t agree with the following statement: *“It also suggests a higher likelihood that the Balaenidae samples correspond to bowhead whales rather than right whales (Figure S7.4) although this is less certain.”* Right whales have very broad foraging patternings globally which often overlap with bowhead whales. The more Arctic conditions during your time period might suggest they were more likely to be Bowheads OR the Bay of Biscay may have represented a northerly feeding ground for NARWs. Please rephrase to capture this.

The following text was added lines 451-452 (also in relation to the comment above):” It also suggests a higher likelihood that the Balaenidae samples correspond to bowhead whales rather than right whales (Figure S7.4) although this is less certain. *While the Arctic conditions in the region would favor the bowhead hypothesis, the region could similarly have functioned as a northerly feeding ground for right whales.”*

- *“The worked objects and unworked bone fragments we analyzed are predominantly large cetaceans that mainly use offshore habitats, namely fin whale, sperm whale and blue whale”.* Although I partially agree with this statement it is not always the case and this should be captured here. In areas where you have deep water close to shore, like the Bay of Biscay or the St. Lawrence Estuary (there are many more examples), more “offshore” whales can be seen in large aggregations from the shore. It is not the shore that is preventing them from being close but the depth and their foraging preferences. Please rephrase to capture this, I suggest something like *“The worked objects and unworked bone fragments we analyzed are predominantly large cetaceans that foraging predominantly offshore or in areas with deep water close to shore like the Bay of Biscay, namely fin whale, sperm whale and blue whale”.*

We changed (lines 468-469) to: *“The worked objects and unworked bone fragments we analyzed are mainly large cetaceans that predominantly forage offshore or in areas with deep water close to shore like the Bay of Biscay—namely fin whale, sperm whale and blue whale.”*

- I like the term “opportunistic acquisition” you have used in this section. Perhaps this could replace exploitation where appropriate?

We changed “exploitation” to “utilization” or “acquisition” throughout the manuscript (except when precisely referring to active capture of whales by humans).

- *“Whales were likely familiar to coastal communities, and as such may well have played an important role in the Magdalenian cultural world. The Bay of Biscay’s highly productive waters make it a cetacean hotspot today (Laran et al. 2017), and it was likely already the case during the Pleistocene.”* I think I’ve read this sentence somewhere else in the manuscript/supp material, just check wording isn’t identical.

We could not find this sentence anywhere else in the manuscript.

- “Whales were likely familiar to coastal communities, and as such may well have played an important role in the Magdalenian cultural world. The Bay of Biscay’s highly productive waters make it a cetacean hotspot today (Laran et al. 2017), and it was likely already the case during the Pleistocene. Along the Spanish coast in particular, water depth increases dramatically just a few kilometers from the coastline, bringing species that are typically found offshore, such as fin whales (McLeod et al. 2009), unusually close. These species could have been observed from high points along the coastline spouting at a distance, and irregularly closer to shore as dead or moribund individuals. The more coastal right/bowhead whales, gray whales and harbor porpoise were likely even better known. Gray whales in particular are the most coastal of the whale species (Wilson & Mittermeier 2014), spending summer at low-depth high-latitude feeding grounds, then migrate hugging the coastline to warmer lower-latitude calving areas in sheltered low-depth lagoons. Whereas in today’s climate the Bay of Biscay and the Mediterranean would more likely correspond to calving grounds, it is possible that during the cooler Upper Paleolithic climate they would have corresponded to feeding grounds (Speller et al. 2016, Rodrigues et al. 2018)—habitat reconstructions indicate that adequate shallow shelf habitat was available there during the last glaciation (Alter et al. 2015). In any case, they would have been conspicuously present at predictable seasons, and they are likely to have been part of the range of animals whose ecology would have been well known to Paleolithic hunter-gatherers.” I really like this paragraph. Perhaps it should go higher up with the paragraph that starts “The taxonomic composition of the whale-bone assemblage gives...”. The paragraph before also starts with “Finally...” and it is no longer finally (although it might be if you move this new paragraph above).

It is indeed better this way: we moved this paragraph as suggested.

Appendix 6:

- I thank the authors for addressing my previous concerns around species-specific DeltaR. I am not overly happy with the justification of the mixed-species deltaR that is used here, more specifically for the sperm whale and gray whale. Please include some additional words in Appendix 6 for these two species, and show the differences in the results for these two species based on deep water feeding (sperm whale, i.e. -400-800 yrs difference rather than ~150) and shallow water feeding (gray whale, i.e. -50 yrs rather than ~150). I understand that given your 1500 yr error, the results do not change, but I think it is important to acknowledge how the deltaR you have chosen isn’t overly optimal for these two species based on their known feeding habits. Also the species that are available on calib.org are mostly pelagic whales (minke, sei, humpback, fin), so I would expect the mean deltaR from these to be more suitable to the fin whale but not the gray whale or sperm whale. Please consider this explicitly somewhere.

We thank the reviewer for agreeing to accept the DeltaR and we agree to include a statement to clarify that this value is not optimal for sperm and gray whales. The following text is added to Appendix 6 in the DeltaR section: “This may not be the case for the Late Pleistocene samples but it does strengthen our idea that the material used for paired dating in this region by Soares is not that far off from whales themselves. However, the DeltaR used here is not optimal for two species in this dataset, namely sperm whales due to their large variation in \$\Delta R\$ (between \$-168\$ and \$504 (\pm 60)\$ \$^{14}C\$ yr (Beck et al. 2022)) and gray whales due to feeding in shallow waters where they might ingest calcareous sediment with different \$^{14}C\$ ages (England et al. 2013).”

- “Choosing the most suitable reservoir correction required careful consideration of various parameters. Foote et al. (2013) used a $\Delta R = 0$ to correct for the reservoir effect in Pleistocene whales. However, this no longer seems suitable since the update of the Marine20 calibration curve, as there are large fluctuations in pre-Holocene reservoir ages (Heaton et al. 2020).” Please change to “Foote et al. 2013 use a ΔR to correct to correct for the reservoir effect in bowhead whales during the Pleistocene”. It is important to note that this previous paper only corrected one species.

We changed the text as suggested.

Appendix 7:

1. Methods for fin whale suess corrections?

The specifics of the Suess corrections are present in the excel file ‘SI_8_PaleoCet_material_SI_comparison’ under the tab ‘Suess correction’. All information obtained from the SuessR package is included here. The value from column ‘G’ (net. cor) was used to correct the $\delta^{13}\text{C}$ values from the literature, including the fin whales.

2. Please change “Fin whales in our sample generally display high $\delta^{15}\text{N}$ ratios when compared with their modern counterparts (Fig. S7.2), indicating that they occupied on average a higher trophic level than today’s whales (Peterson and Fry 1987).” To “Fin whales in our sample generally display high $\delta^{15}\text{N}$ ratios when compared with their modern counterparts (Fig. S7.2), indicating that they occupied on average a higher trophic level than today’s whales (Peterson and Fry 1987), or, the $\delta^{15}\text{N}$ baseline has significantly changed over time”. I would argue that as these differences are observed across multiple species, changes in the $\delta^{15}\text{N}$ baseline is more likely given the environmental change rather than occupying a higher trophic level in the past.

We changed the text as suggested.

Response to authors direct responses:

- Fin whales often forage at a slightly higher trophic level relative to Balaenidae, but here they have lower nitrogen isotopes, why? Is it because the fin whales were feeding at higher latitudes where baseline isotopes differ (lower nitrogen likely?).

We were also puzzled by this. Following advice from the reviewers (see below) we now have included as supplementary information a comparison between the isotopic signatures of our Paleolithic samples with those of modern counterparts (Supplementary Information 7). Comparing modern fin whales (Figure S7.4) and modern bowhead and North Atlantic right whales (Figure S7.6), we found that fin whales also have lower $\delta^{15}\text{N}$ ratios than bowhead whales and a similar range to right whales. In other words, this pattern is also evident among modern whales (and indeed it is even stronger than among ancient samples). This is very interesting, given what is known about the diets of these two species, I would suggest this is more likely to be due to predominant feeding in different feeding areas at high latitudes. At least for modern fin whales, many migrate to Iceland, whilst NARWs have much more coastal migratory patterns along the coast of North America. Although the eastern NARW population is extinct, I would have expected a coastal migratory pattern too relative to the more offshore deep feeding fin whale. The depth these two organisms feed at should also be considered. Please add somewhere a statement on their differing migrations (coastal vs. open ocean) and

summer feeding grounds as a likely cause of the differing $\delta^{15}\text{N}$ (due to differences in baselines rather than trophic level).

The following text is added to Supplementary Information 7 page 6: “Despite the partial overlap between the modern samples of both species precluding a more certain species identification of the Paleolithic bone samples, they ancient sample appear more likely to be bowheads than right whales. *Although fin whales often are known to forage at a slightly higher trophic level relative to Balaenidae, when comparing modern fin whales (Figure S7.4) and modern bowhead and North Atlantic right whales (Figure S7.6), we found that fin whales also have lower $\delta^{15}\text{N}$ ratios than bowhead whales and a similar range to right whales. This pattern is not only evident among ancient whales samples but also in modern whales and can probably be explained by their different migration styles (coastal vs open ocean) and summer feeding grounds characterized by different baseline $\delta^{15}\text{N}$ values.*

- Lines 364-365 = 437-439: *Fin whales also inhabit the Mediterranean and all species forage close to the islands of the Azores. I would describe them as migratory habitat specialists rather than just offshore whales. Careful. We reworded the sentence to refer to these species as “large cetaceans that mainly use offshore habitats”. We are unsure why referring to them as “migratory habitat specialists” would have been clearer in the context of the point we are making here, which is that it is extremely unlikely that these species would have been whaled at the time. I would clarify by adding “or found foraging in deep waters close to shore” (e.g. the Bay of Biscay).*

We changed to (lines 468-469): “The worked objects and unworked bone fragments we analyzed are mainly large cetaceans that predominantly forage offshore or in areas with deep water close to shore like the Bay of Biscay—namely fin whale, sperm whale and blue whale.”

Reviewer #2

The authors have revised their manuscript, with an improvement in what information is retrieved from the stable isotope analyses per species as recommended by Reviewer 1.

Most importantly, they clarify a misunderstanding caused by misleading text, that the authors are not claiming that this represents any indication of active hunting of whales, and that this is supposed to represent a focus more on the significance of tool making from available materials.

However, remnants of the earlier text that causes such confusion still remains. For example, in the abstract itself:

“Reconstructing the prehistoric exploitation of large cetaceans is challenging, but key to understand the history of early human coastal adaptations” (as collecting bones from coastal environments and working them does not have the same implications as adapting to life on the coast)

and

“reveals at least five species of large whales, significantly expanding the range of known taxa exploited in this period” (again, ‘exploitation’ here could be considered misleading)

These sentences could very easily be interpreted as hunting, whereas a few additional words relating this to their carcasses would clarify this, or, if keeping the potential for active hunting in their narrative, then they could add a separate short sentence clarifying that this is not the

only/most likely scenario. It is important as there is a great difference in significance, which the authors should detail in the Discussion.

There was indeed a misunderstanding regarding our use of the word “exploitation”. As it is usually employed in prehistoric archeology, the expression “the exploitation of animal resources” applies to any organized use of animal resources, whatever the acquisition method, and that includes collecting and scavenging (as per <https://doi.org/10.4000/palethnologie.5137>), thus does not necessarily imply active hunting. In our manuscript, it was defined as such in the first lines of the introduction (“exploitation methods that included scavenging freshly beached animals, opportunistic killing and organized whaling”). We changed the word for “utilization” “acquisition” or similar expressions, throughout the manuscript.

The newly added sentence:

“They were thus a key part of subsistence for many coastal human groups worldwide, including hunter-gatherers and Neolithic farmers, with exploitation methods that included scavenging freshly beached animals, opportunistic killing and organized whaling” retains the notion that

Amendments to the isotope interpretations, as helped by Reviewer 1 suggestions, gives much greater significance to the palaeoecology of the different species, but does not add to the interpretation of bone tool making.

The interpretation of using whale bone to manufacture tools has been discussed in previous articles that reported the initial (only visual, without taxonomic ID, and badly dated) identification of whale bone among Magdalenian assemblages (Pétillon 2008, 2013, Langley and Street 2013, Lefebvre et al. 2021). We added a paragraph (lines 348-355) to refer to this discussion: “Most of the objects made of whale bone are weapon elements (projectile points and foreshafts) typologically similar to the antler points that make up a large part of the Magdalenian hunting equipment (e.g., Pétillon 2016b). From what can be observed on the finished objects, and on the few pieces of manufacturing waste documented (e.g., Lucas et al. 2023), the manufacturing techniques used to work whale bone into objects do not differ from those used for working antler. The choice of using whale bone to make one part of the weapon tips might be linked to the dimensions of the raw material, that make it possible to manufacture very long implements, and perhaps the specific mechanical properties of whale bone as compared to antler and to terrestrial bone (for details refer to Pétillon 2008, 2013, Langley and Street 2013, Lefebvre et al. 2021).”

The authors point out that “DNA analysis would be required to verify the species identities for these specimens”, in reference to the balaenid remains, but then why not attempt DNA analysis? (particularly given the expertise of some of the authors). Is this due to cost, expected failure rate, or sample size?

Destructive testing applications for this project initially focused on ZooMS for taxonomic screening, radiocarbon dating, and SIA; DNA analysis was not within the original scope of the project. We are currently pursuing mtDNA genome analyses to verify the species ID, but this falls outside the scope of this study. We added line 396: “DNA analysis would be required to verify the species identities for these specimens (*analyses which fell outside the scope of this study*)”.

For example, as was done with Hufthammer et al. 2018 for using ZooMS supported by aDNA to show presence of grey whale in the North Atlantic waters around Norway – a reference that would support the statement:

“Previous evidence of their past presence in the North Atlantic dates either to the Holocene (10 to 0.25 ka cal BP) or to the middle of the Marine Isotope Stage 3 (MIS3), before 40 ka cal BP;...”

We added this reference (line 412).

Given that the authors claim the porpoise was too small to radiocarbon date it would be useful to clarify this point in the text also (preferably with specimen sizes/masses if available, though not essential if not), i.e., is this more about actual sample size, or sampling damage, etc.

We added a sentence in the methods section (lines 593-595): “The single object identified as made of porpoise bone through ZooMS analysis was a thin fragment of point, and permission was not granted to sample material sufficient for both ZooMS and 14C dating.”.

Now that I know the main scope is on the bone tool manufacture, rather than the implied cognitive abilities of active whale hunters, there are a few other questions more specifically about this manuscript:

What is the significance of the worked bone tools being made of remains from these particular species?

How does tool manufacture differ from other known species from the same region of the same time (and earlier)? See numerous articles referred to by Borao 2022 in JAS reports. This aspect of working bones needs to be much more detailed, or direct this as a manuscript talking about palaeoecology (still, this disjunct between the two aims leaves it somewhat unclear what the main point is – the isotope ecology of particular cetacean species at a particular time in the past OR the fact that the contemporary people were creating tools from such remains)

I suspect that it is more the latter now, based on the response to reviewers comments and amended version.

The interpretation of using whale bone to manufacture tools has been discussed in previous articles that reported the initial (only visual, without taxonomic ID, and badly dated) identification of whale bone among Magdalenian assemblages (Pétillon 2008, 2013, Langley and Street 2013, Lefebvre et al. 2021). We added a paragraph (lines 348-355) to refer to this discussion: “Most of the objects made of whale bone are weapon elements (projectile points and foreshafts) typologically similar to the antler points that make up a large part of the Magdalenian hunting equipment (e.g., Pétillon 2016b). From what can be observed on the finished objects, and on the few pieces of manufacturing waste documented (e.g., Lucas et al. 2023), the manufacturing techniques used to work whale bone into objects do not differ from those used for working antler. The choice of using whale bone to make one part of the weapon tips might be linked to the dimensions of the raw material, that make it possible to manufacture very long implements, and perhaps the specific mechanical properties of whale bone as compared to antler and to terrestrial bone (for details refer to Pétillon 2008, 2013, Langley and Street 2013, Lefebvre et al. 2021).”

It is good to see some initial speculation on skeletal element use, though which the authors acknowledge cannot be inferred due to the workings, though I think the following paragraph on the second exploitation behaviour, of the unworked bone, could also link in here. This

paragraph would in particular benefit some information with regards the species – as I do not see any taxon reference more specific than ‘whale’ in this whole paragraph, but if discussing aspects of transport, surely relating this to species (and their sizes) would be of importance.

We changed a sentence to add the mention of the species (lines 522-524): “The transportation of these elements requires explanation, as the species involved (fin whale and right/bowhead whale) are large, and the site was by then 4-5 km from the coast and 70 m up a steep cliff (Berganza & Arribas 2014a).” References to the large size of the bone elements were also added to lines 525 and 537.

Overall this does offer some interesting data on bone tools, through isotopes of specimens confirmed through molecular fingerprinting, supported through radiocarbon dating, but still would benefit from improvement in context with regards bone tool manufacture in relation to the current knowledge on bone industry more broadly in the Discussion.

The interpretation of using whale bone to manufacture tools has been discussed in previous articles that reported the initial (only visual, without taxonomic ID, and badly dated) identification of whale bone among Magdalenian assemblages (Pétillon 2008, 2013, Langley and Street 2013, Lefebvre et al. 2021). We added a paragraph (lines 348-355) to refer to this discussion: “Most of the objects made of whale bone are weapon elements (projectile points and foreshafts) typologically similar to the antler points that make up a large part of the Magdalenian hunting equipment (e.g., Pétillon 2016b). From what can be observed on the finished objects, and on the few pieces of manufacturing waste documented (e.g., Lucas et al. 2023), the manufacturing techniques used to work whale bone into objects do not differ from those used for working antler. The choice of using whale bone to make one part of the weapon tips might be linked to the dimensions of the raw material, that make it possible to manufacture very long implements, and perhaps the specific mechanical properties of whale bone as compared to antler and to terrestrial bone (for details refer to Pétillon 2008, 2013, Langley and Street 2013, Lefebvre et al. 2021).”

Reviewer #4

I am pleased to see the authors carefully considering the reviewers' comments. The revised manuscript is more thorough, and the methodologies are more appropriately explained, providing better support for the results. I especially appreciate the addition of supplemental files 6, 7, and 8.

Regarding the paleoecological component of this work, the authors have done a satisfactory job by compiling stable isotope data on modern whales to contextualize their data on fossils. As suggested, they conducted corrections for tissue type and the Suess effect to make this modern data comparable in a contemporary context. However, when performing the next step (i.e., comparing modern and fossil data), the authors overlooked the long-term temporal shifts in the carbon and nitrogen isotope compositions at the base of the food since the Pleistocene. The baseline changes can be significant, potentially compounding interpretations of patterns of moderns vs. fossils. I understand the focus of this paper is not reconstructing ancient marine isotopic baselines; however, some reflection on this issue must be considered in the text, wildly when pinpointing changes in the foraging ecology because of climate change on page 9.

I also noticed that the authors conducted interpretations based on the individual variation of the $\delta^{13}\text{C}$ OR $\delta^{15}\text{N}$ values. Technically, nothing is wrong with this, but it is important to note that this approach can lead to an incomplete view of the underpinning factors causing their variation. The $\delta^{13}\text{C}$ and $\delta^{15}\text{N}$ values of an individual are not independent, as they can be influenced by the same external (e.g., primary productivity, trophic level) and internal (e.g., metabolic routing) factors. Therefore, in addition to basing interpretations on the variation of each of these systems separately, it is also recommended that a more comprehensive interpretation be conducted that considers both. For example, as an organism's trophic level increases, not only its $\delta^{15}\text{N}$ values but also its $\delta^{13}\text{C}$ values in a predictable fashion. I encourage the authors to assess the covariation of the $\delta^{13}\text{C}$ AND $\delta^{15}\text{N}$ values and examine whether their inferences are consistent with the patterns of variation expected for both the $\delta^{13}\text{C}$ and $\delta^{15}\text{N}$ values. For instance, do the $\delta^{13}\text{C}$ values support conclusions about trophic ecology derived from $\delta^{15}\text{N}$ values? Or are deductions about benthic diving based on the $\delta^{13}\text{C}$ values of some species consistent with their corresponding $\delta^{15}\text{N}$ values?

We agree with the reviewer that stable isotope research usually allows a more holistic picture to be made by looking at how the $\delta^{13}\text{C}$ and $\delta^{15}\text{N}$ ratios co-vary. However, in these situations the stable isotope values of the food source(s) are then usually also known or measured. Understandably, we lack this information here for the ancient samples. In addition to comparing the ancient whale samples with modern samples in Supplementary Information 7, the co-variation between both isotopic values is also discussed when possible, as is feeding in benthic environments for gray whales.

Some other minor comments are included in the annotated manuscript and supplemental file 7. I hope these comments can be easily addressed. I appreciate the quality and importance of this study and look forward to seeing it published.

Comments in the annotated manuscript:

Lines 122: instead of “ $\delta^{13}\text{C}$ and $\delta^{15}\text{N}$ stable isotope ratios”, write “The $\delta^{13}\text{C}$ and $\delta^{15}\text{N}$ values”.

Done.

Lines 197-198: instead of “*shed light on the whales’ relative foraging behavior and past ocean ecology*”, write “Contributing to reconstructing whales’ relative foraging behavior and past ocean ecology”.

Done.

Line 219: instead of “...*although also including one seal. The higher error rate for Santa Catalina...*”, write “... but also one seal. The higher error rate for the unworked bone fragments from Santa Catalina...”

Done.

Line 227: Balaenidae => do not italicize the family name.

Done.

Lines 273: instead of “*all but one yielded a result*”, write “all but one yielded reliable results”.

Done.

Figure 1: do not italicize Phocidae.

Done.

Lines 287-289: “*In total, 8 samples failed collagen extraction, 2 samples failed IRMS analysis and for 7 samples the atomic CN ratio was too high or the N peak was below 700 mV.*” => How high? Please specify the range. Refer to Supplementary file 8.

Samples with atomic CN ratios that fell outside the acceptable range or produced N peaks below 700 mV were not included in Supplementary file 8, as this file deals specifically with samples suitable for interpretation and comparison with modern samples. These values are, however, reported in ‘SI_1_PaleoCet_material_revised’, column AQ shows the atomic CN ratios, while column AR shows remarks, including ‘N peak < 700 mV’. The following text is added to the manuscript (italics): “*In total, 8 samples failed collagen extraction, 2 samples failed IRMS analysis and for 7 samples the atomic CN ratio was too high (ranging between 3.77 and 6.28) or the N peak was below 700 mV (Supplementary Information 1).*”

Lines 291-295: “*fin whale samples are clustered at low $\delta^{15}\text{N}$ values and (with one exception) intermediate values of $\delta^{13}\text{C}$; the single blue whale sample is nested among the fin whales; right/bowhead whale samples show low $\delta^{13}\text{C}$ ratios and intermediate values of $\delta^{15}\text{N}$; gray whales display high $\delta^{13}\text{C}$ ratios and intermediate values of $\delta^{15}\text{N}$; and sperm whales show elevated values both of $\delta^{15}\text{N}$ values and $\delta^{13}\text{C}$.*” => I recommend adding the average and range of $\delta^{13}\text{C}$ and $\delta^{15}\text{N}$ values for each species/group in the text or a table.

Agreed, we added averages, standard deviations and ranges for each species to the SI 8 file under the tab ‘Paleocet material’ and this information is also included in a table (Table 1) in the main text:

	$\delta^{13}\text{C}$ (‰)	$\delta^{15}\text{N}$ (‰)
fin whales		
average	-14.1	11.3
stdev.	0.76	0.72
min	-16.2	9.1
max	-11.6	13.0
Gray whales		
average	-12.8	12.9
stdev.	0.51	0.93
min	-13.3	11.9
max	-12.3	14.1
Right/bowhead whale		
average	-16.0	12.9
stdev.	0.55	0.60
min	-16.8	12.2
max	-15.0	14.1
sperm whales		

average	-13.2	16.8
stdev.	0.51	0.47
min	-14.0	16.1
max	-12.4	17.5

Line 296-297: “*strongly suggesting it corresponds to a fin whale*” => “might have corresponded”.

Done.

Line 299: instead of “*and the $\delta^{13}C$* ”, write “and the $\delta^{13}C$ values”.

Done.

Line 301: instead of “*Paleolithic samples*”, write “Paleolithic whale samples”.

Done.

Line 301-305: “*modern counterparts*” => Specify whether those modern counterparts inhabited (or were collected) the same region where the fossil remains were found.

The following text was added: “A comparison between the isotopic signatures of Paleolithic samples with modern counterparts shows in most cases an overlap (at least partial) between the two. *The large majority of modern samples used for this comparison originate from the same oceanic basin (Atlantic) as our samples, except for gray whale (Pacific) and one study with sperm whales (Pacific) (Supplementary Information 8).*”

Figure 3: Perhaps it is just me, but the color difference between bowhead/right whales and sperm whales is too subtle. I suggest adding more contrast or using a different symbol. Also, you could add points depicting each species' average and standard.

We changed the symbol of the right/bowhead whales to a filled square but kept the same colors, which are consistent across figures. However, we preferred to not include the averages and standard deviations to this figure, as it would clutter the graph. Instead, we added this information in table form to the main text (Table 1).

Lines 401-403: instead of “*The sea-ice loving bowhead whales could have been a more likely species, and stable isotope analysis also points in this direction (Supplementary Information 7). It is not impossible that both species could have been present, in different seasons.*”, write: “The sea-ice bound bowhead whales could have been a more likely species, with stable isotope also providing some support also in this direction (Supplementary Information 7, figure S7.6).”

Done.

Line 408: instead of “*even if the circumstances*”, write “despite the circumstances”.

Done.

Line 422: “*Stable isotope signatures reflect the feeding strategies of animals*” => I suggest adding some precision: The stable isotope composition of marine vertebrates is controlled by their diet and habitat preferences.

Done.

Line 424: “*onto past ocean ecology*” => “*onto whales’ ecology*” would be more accurate.

Done.

Lines 424-427: “*In general, the isotopic signatures of Paleolithic whales overlap those of their modern counterparts, suggesting broadly similar feeding strategies (Supplementary Information 7-8).*” => This sentence should be rephrased as differences in the stable isotope composition among fossil and modern representatives exist (e.g., Figure S7.4)

We modified the sentence as follows (italics): “*In general, the stable isotopic signatures of Paleolithic whales overlap to some extent with those of their modern counterparts, suggesting broadly similar feeding strategies, while existing differences can be ascribed to variation in stable isotope baseline values, feeding ground locations and possibly trophic level (Supplementary Information 7-8).*”

Lines 427-428: “*As in modern whales, ancient fin whales and blue whale samples have relatively low nitrogen stable isotope ratios*” => Low compared to what?

We modified the sentence as follows (italics): “*As in modern whales, ancient fin whales and blue whale samples have relatively low nitrogen stable isotope ratios in comparison with the other whale samples, consistent with these species’ reliance on krill.*”

Lines 427-432: “*As in modern whales, ancient fin whales and blue whale samples have relatively low nitrogen stable isotope ratios, consistent with these species’ reliance on krill (Borrell et al. 2012; Blevins et al. 2022), whereas gray whales display higher $\delta^{13}\text{C}$ ratios, characteristic for their feeding behavior in benthic environments (Caraveo-Patiño et al. 2007), while sperm whales (the only toothed whale species in this dataset) show the most elevated $\delta^{15}\text{N}$ values, reflecting a diet at a higher trophic level including large squid (Ruiz-Cooley et al. 2004) (Figure 3).*” => Although independent analysis/interpretation of the variation in $\delta^{13}\text{C}$ and $\delta^{15}\text{N}$ values can be informative, examining their joint variation can provide a more complete picture of the underlying factors. In other words, the $\delta^{13}\text{C}$ and $\delta^{15}\text{N}$ values should be “read” and interpreted together. For example, there is a stepwise enrichment of trophic levels of $\sim 1\text{‰}$ for $\delta^{13}\text{C}$ and 3‰ for $\delta^{15}\text{N}$ in marine predators. Therefore, if patterns of $\delta^{13}\text{C}$ values are assessed alone (without paying attention to patterns in their corresponding $\delta^{15}\text{N}$ values), they could easily be misinterpreted for changes in feeding habitat (more coastal) when it is not. Consequently, I suggest reconsidering these interpretations based on the pooled analysis of variations of the $\delta^{13}\text{C}$ and $\delta^{15}\text{N}$ values. Following this rationale, do the $\delta^{13}\text{C}$ values of sperm whales support the interpretation that they consume higher trophic-level prey?

We have looked at both isotopes in tandem but examining this step-wise enrichment is hard to do if we don’t have stable isotope values of the consumed prey. Therefore we compare the whales’ stable isotope results in relation to each other and those of modern species.

Line 436: instead of “higher $\delta^{13}\text{C}$ ratios”, write “higher of $\delta^{13}\text{C}$ values”.

Done.

Lines 433-434: “*It is puzzling that bowhead/right whales (which feed mostly on copepods) appear as having higher $\delta^{13}\text{C}$ ratios than fin whales*” => I see the opposite in Figure 3. Bowhead/right whales have the lowest $\delta^{13}\text{C}$ values (~16 per mil) among fossil whales. The same occurs for modern representatives (~-18 per mil vs.~-16 per mil)

This should be “lower $\delta^{13}\text{C}$ values” instead of “higher $\delta^{13}\text{C}$ ratios”! We corrected that.

Line 434-436: “*the overlap (...) is typically partial*” => What do you mean by that?

We modified the sentence as follows: “This said, *in most cases* the overlap in stable isotope signatures between modern and ancient whale samples is *only partial*”

Lines 436-439: “*These differences may reflect a shift in whale feeding preferences (e.g., if modern whales now target prey at lower trophic levels or warmer waters), or environmental change, or a combination. Indeed, previous studies found rapid changes in whale feeding behavior (and in corresponding stable isotope signatures) as a result of environmental changes: for example, Jory et al. (2021) found that a reduction in biomass of arctic krill coincided with a dietary niche widening in fin whales in Canadian waters, with 60% of individuals changing their foraging strategy from specialist to generalist feeders in order to reduce intraspecific competition. Given the magnitude of the environmental change from the Late Pleistocene to the present, the changes in whale foraging strategies may well have been even stronger.*” => What about temporal changes in the carbon and nitrogen isotope compositions at the base of the food web? Might these differences be explained instead by large-scale temporal changes in the isoscape since the Pleistocene? For instance, see Reade et al. (2023). <https://doi.org/10.1371/journal.pone.0268607>

The sentence was modified as follows: “These differences may reflect a shift in whale feeding preferences (e.g., if modern whales now target prey at lower trophic levels or warmer waters), *large scale temporal changes in the stable isotope baseline (Reade et al. 2023), environmental change, or a combination.*”

Lines 457-458: “*The proportions of the different whale taxa in our sample are thus unlikely to be affected by taphonomic factors such as differential preservation of bone tissues.*” => What do you mean by proportions? Abundance?

We changed the sentence as follows: “*The relative abundance of the different whale taxa in our sample is thus unlikely to be affected by taphonomic factors...*”

Lines 462: instead of “*the scarcity of the harbor porpoise*”, write “the scarcity of the harbor porpoise remains”.

Done.

Lines 473-474: “*species whose ecology brings them substantially closer to the coast, namely gray whales, right/bowhead whales and harbor porpoises*” => add reference.

We added a reference to Wilson & Mittermeier 2014.

Line 611: “*for radiocarbon dating*” => Only for radiocarbon dating?

Yes, we added that information: “For radiocarbon dating *only*”.

Line 619: “*a non-invasive collagen sampling method*” => I would rather say a “minimally invasive” collagen sampling method. (Same for lines 624 and 628.)

Done.

Line 623: “*see results below*” => above?

Yes, we corrected that.

Line 643: instead of “*both qualities*”, write “both bone conditions”.

Done.

Line 646: add a comma “as a result,”

Done.

Line 650, “*Subsamples of bone or bone powder, ranging from 20 to 70 mg*” => You stated above, “On average, 75.7 mg of bone was extracted from each specimen for ZooMS”. Please clarify.

There were sometimes several subsamples per specimen. We changed the sentence as follows: “Subsamples of bone or bone powder, *sometimes several per specimen*, and ranging from 20 to 70 mg”.

Line 763-764: “*a Delta V Advantage isotopic mass spectrometer*” in what institution?

We added that information: “a Delta V Advantage isotopic mass spectrometer *in the Service de Spectrométrie de Masse Isotopique (SSMIM) in the Muséum national d’Histoire naturelle, Paris, France.*”

Line 766: instead of “ $\delta^{13}C$ and $\delta^{15}N$ values”, write “The $\delta^{13}C$ and $\delta^{15}N$ values”

Done.

Comments on Supplementary information 7:

Page 3, “*We used the SuessR package (Clark et al. 2020) to calculate the correction needed based on the region and year of sample collection.*”: refer to Supplementary information 8.

We added that reference.

Page 3, “*dilapidation*”: The correct word is delipidation or defatting. Also correct the corresponding table in the Supplementary Information 8.

Done.

Page 5, “*The ancient blue whale overlaps with the higher end of the isotopic ranges of modern whales (Fig. S7.4), compatible with similar feeding strategies*”: I don’t see an overlap, which makes me wonder how you conclude that they might have similar feeding strategies.

We changed this sentence for: “*The ancient blue whale shows higher $\delta^{13}\text{C}$ and $\delta^{15}\text{N}$ values than its modern counterparts, and isotopically appears to fall in between the modern and ancient fin whales (Fig. S7.4).*”

Page 5, “*and the Pacific Oceans (37 skin samples from Guerra et al. 2020)*”: By including the Pacific, you are combining data from two regions with different $\delta^{13}\text{C}$ and $\delta^{15}\text{N}$ isoscapes, which you do not take into account. I suggest only including data for North Atlantic whales if possible.

We removed the data from Guerra et al. 2020 from the graph and from the text, in both SI7 and SI8.

Page 7, “*ancient gray whale from the Pacific*”: The addition of data from animals from the Pacific is justified here.

Agreed!

Manuscript NCOMMS-23-62988, “Late Paleolithic whale bone tools reveal human and whale ecology in the Bay of Biscay”: detailed response to reviewers

Black lettering = reviewers’ requests.

Blue lettering = authors’ responses and comments.

Reviewer #2:

The manuscript now reads in a more coherent matter that yields field-specific interests, such as the application of isotope analyses and ZooMS to improving how we can interpret an archaeological assemblage – particularly with regards the isotope information.

I do still have some concerns on the overall claim of this study that tries to emphasize ancient human-seashore interactions – i.e., I understand that they may represent the oldest dated bones from whale remains but it is not clear how the results are significantly revealing about whale ecology and ancient human-seashore interactions more that already known or expected.

The statement “significantly expanding the range of known taxa acquired by humans” is somewhat overstretching the facts, particularly given that later in the manuscript you state:

“Fin whales, sperm whales, blue whales and harbor porpoises are still present in the Bay of Biscay today” – so, it is unclear whether the authors are emphasizing that adding two species to this list a significant expansion of our knowledge of the ecology, or, that humans in the past were selecting for the remains of specific species to utilise? I.e., why the contemporary humans would not simply make best use of any cetacean remains that stranded upon their shores.

What the significance of this oldest whale bone use in relation to the use of terrestrial animal bones for crafting tools from remains ambiguous. I.e., it is not clear if there a particular reason why coastal populations would avoid using such remains.

The rise of the sea level in the late Pleistocene submerged the Paleolithic coastline, and drowned or destroyed the sites that could have documented the seashore occupation and use by humans, including their use of whale products. Before this study, we could indeed hypothesize that a number of cetacean species were present in the Bay of Biscay at that period; that some of these species were similar to those observed in the same area today despite the strong environmental differences; that strandings did occur; that people used the products from these strandings; and that these uses included bone tool-making. But there was no direct analytical proof of this, no precise notion of the range of cetacean taxa involved and of their ecology at that period, and no dates for this hypothetical phenomenon. We provide results for all these aspects, including e.g. the first indirect evidence for the use of other, unpreserved whale products such as baleen (through the first formal identification of mysticetes in that context). Our radiocarbon dates show that, unexpectedly and rather counter-intuitively, the practice of whale bone working does not seem to be a “background noise” throughout the Upper Paleolithic, but appears rather limited in time, with a peak phase followed by an almost complete disappearance, while the whale bone resource itself is still present. Among the whale taxa identified is the Gray whale, previously thought to be absent

from the North Atlantic in MIS2 (Alter et al. 2012, 2015). We believe these results, and others in the paper, are beyond what was known or expected for this context.

Use of the phrase ‘whale utilization’ remains misleading because in this instance it equally implies active hunting of live animals – should be ‘whale carcass utilisation’ where appropriate, unless, as mentioned for last comment, there is some reason why such human groups would not instinctively utilize such an obvious resource.

Regarding how humans were accessing the whale bones, we have revised the text and added qualifiers (“whale products”) to remove any remaining ambiguity. Note that in some cases, we do refer to whale acquisition in the general sense, including active hunting (e.g. in the introduction, where we say “reconstructing the beginning of whale utilization”: this is about the beginning of a process that did culminated with whaling, not just to the “beginning of whale carcass/remains utilization”). Edits were done on lines 82, 112, 114, 126, and 156. There are not many changes because in most cases we already referred to “whale bones”, “carcasses” and “products”. Also note that we explicitly address acquisition in the discussion (lines 408-413) where we spell out that the evidence points to these bones not coming from hunting: “However, there is no evidence that European Pleistocene hunter-gatherers had the necessary technologies for hunting these species, such as seafaring (Philippe 2018), or multibarbed points that could have been used as harpoons heads (barbed points appear in the local archeological record only after 16 ka cal BP: Pétillon 2016). Overall, then, the archaeological evidence points towards an opportunistic acquisition of whale resources in the Bay of Biscay during the Magdalenian period”.

‘range of whale taxa acquired’ is also misleading – although true in strictest sense, it implies targeting of particular species, some level of selection, but if these are strandings then no such selection process occurred – indeed, aspects of selection are discussed in the Discussion section of the manuscript, though this should be improved with current information on strandings of these species.

The expression “range of whale taxa acquired” is not used in the last version of the manuscript. The abstract mentioned the “range of known taxa utilized”, and we changed it to “range of known taxa whose products were utilized” to avoid any remaining ambiguity (see comment above).

With respect to “Our results show that at least one balaenid species was found in the Bay of Biscay at the time, although the imprecision of the ZooMS ID does not allow for a distinction between right and bowhead whales; DNA analysis would be required to verify the species identities for these specimens (analyses which fell outside the scope of this study)”

If understanding the palaeoecology contemporary with these oldest worked cetacean bones is one of the key features of the manuscript, it is unclear why the DNA work is not done on these samples. More surprising given that you have cetacean ancient DNA specialists in the author list, and there is a column in the supplementary files that states several of the samples having subsamples sent for DNA analysis, but no clear mention of any such attempt in the manuscript, or a methods section on how this was done. It would be even more interesting given that based on the text later in the same paragraph, the species-level resolution would yield seasonal interpretation. Therefore better justification of avoiding this, and clarity on what was done, would help the reader better understand the work.

The results presented in this manuscript are the output of several projects that did not include funding or sampling requests for aDNA analysis. It was outside the scope of the study. The ZooMS results on the “Right/Bowhead” whales led us, after the end of the projects and after due authorization, to subsample some of the specimens for future aDNA analysis. Research on these is in progress, but waiting for the results (if any) would have delayed publication for several years and lengthened the manuscript beyond journal recommendations. We believe the results in their current form are of interest for the readership of the journal, and we hope we will be able to follow up with aDNA results in the future.

The comment “It is puzzling that bowhead/right whales (which feed mostly on copepods) appear as having lower $\delta^{13}\text{C}$ values than fin whales, but this is also the case in modern whales” – though could this be rephrased to highlight the limitations in our understanding of isotopic signatures?

We followed the reviewer’s proposal and rephrased this sentence to read (lines 365-368):

“The fact that bowhead/right whales (which feed mostly on copepods) have higher $\delta^{15}\text{N}$ values than fin whales (again, also found in modern whales) is less straightforward to explain, but likely explained by feeding in different water masses, highlighting the complexity of interpreting isotopic signatures.”

We took the opportunity to correct a mistake that we had introduced in this sentence in the last revision: we had corrected “higher $\delta^{13}\text{C}$ ” (which is not true; see Figure 3) to “lower $\delta^{13}\text{C}$ ” which is correct but was not the point being discussed here (and further discussed in Supplementary Information): what is surprising is that bowhead/right whales have higher values of $\delta^{15}\text{N}$ when they feed at lower trophic levels.

As a minor point, an explanation of why “one sample where ZooMS failed to distinguish between right/bowhead and fin whales” would be important here. I.e., was this based on a poor-quality spectrum? If so, was this due to poor protein preservation of the sample?

This sample generated spectra of relatively poor quality suggesting the collagen was not well preserved, and in particular was missing the *P2_a2 292* collagen marker which distinguishes between fin and bowhead/right whales.

Line 241-244 of the manuscript was updated to: “The one sample where ZooMS failed to distinguish between right/bowhead and fin whales—due to the overall poor quality of the spectra and specifically the absence of a clear peptide marker at *P2_a2 292*—is nested within the fin whale samples, strongly suggesting it might have corresponded to a fin whale (Figure 3)”.

Overall, the authors present some useful information relating to isotope analyses on a small number of cetacean species from this interesting area that will be of interest to some archaeologists and palaeoecologists.